# Overview and update of the SPARC Data Initiative: Comparison of stratospheric composition measurements from satellite limb sounders

Michaela I. Hegglin[1], Susann Tegtmeier[2], John Anderson[3], Adam E. Bourassa[2], Samuel Brohede[4], Doug Degenstein[2], Lucien Froidevaux[5], Bernd Funke[6], John Gille[7], Yasuko Kasai[8], Erkki Kyrölä[9], Jerry Lumpe[10], Donal Murtagh[11], Jessica L. Neu[5], Kristell Pérot[11], Ellis Remsberg[12], Alexey Rozanov[13], Matthew Toohey[2], Joachim Urban[11, deceased], Thomas von Clarmann[14], Kaley A. Walker[15], Hsiang-Jiu Wang[16], Carlo Arosio[13], Robert Damadeo[12], Ryan Fuller[5], Gretchen Lingenfelser[12, retired], Christopher McLinden[17], Diane Pendlebury[17], Chris Roth[2], Niall J. Ryan[15], Christopher Sioris[17], Lesley Smith[18], and Katja Weigel[13]

[1]Department of Meteorology, University of Reading, Reading, United Kingdom
[2]Institute of Space and Atmospheric Studies, University of Saskatchewan, Saskatoon, Canada
[3]Atmospheric Science, Hampton University, Hampton, VA, United States
[4]FluxSense AB, Gothenburg, Sweden
[5]Jet Propulsion Laboratory, California Institute of Technology, Pasadena, CA, United States
[6]Instituto de Astrofísica de Andalucía, CSIC, Granada, Spain
[7]National Center for Atmospheric Research, Boulder, CO, United States
[8]National Institute of Information and Communications Technology, Tokyo, Japan
[9]Earth observation, Finnish Meteorological Institute, Helsinki, Finland
[10]Computational Physics, Inc., Boulder, CO, United States
[11]Department of Space, Earth, and Environment , Chalmers University of Technology, Gothenburg Sweden
[12]NASA Langley Research Center, Hampton, VA, United States
[13]Institute of Environmental Physics (IUP), University of Bremen, Bremen, Germany
[14]Karlsruhe Institute of Technology, IMK, Karlsruhe, Germany
[15]Department of Physics, University of Toronto, Toronto, Canada
[16]School of Earth and Atmospheric Sciences, Georgia Institute of Technology, Atlanta, GA, United States
[17]Environment and Climate Change Canada, Toronto, Canada
[18]Cooperative Institute for Research in Environmental Sciences, University of Colorado, Boulder, CO, United States

**Correspondence:** M. I. Hegglin (m.i.hegglin@reading.ac.uk)

**Abstract.**

The SPARC Data Initiative (SPARC, 2017) performed the first comprehensive assessment of currently available stratospheric composition measurements obtained from an international suite of space-based limb sounders. The initiative's main objectives were (1) to assess the state of data availability, (2) to compile timeseries of vertically resolved, zonal monthly mean trace gas and aerosol fields, and (3) to perform a detailed inter-comparison of these timeseries, summarising useful information and highlighting differences among datasets. The datasets extend over the region from the upper troposphere to the lower mesosphere (300-0.1 hPa) and are provided on a common latitude-pressure grid. They cover 26 different atmospheric constituents including the stratospheric trace gases of primary interest, $O_3$ and $H_2O$, major long-lived trace gases ($SF_6$, $N_2O$, HF, $CCl_3F$, $CCl_2F_2$,

NO$_y$), trace gases with intermediate lifetimes (HCl, CH$_4$, CO, HNO$_3$), and shorter-lived trace gases important to stratospheric chemistry including nitrogen-containing species (NO, NO$_2$, NO$_x$, N$_2$O$_5$, HNO$_4$), halogens (BrO, ClO, ClONO$_2$, HOCl), and other minor species (OH, HO$_2$, CH$_2$O, CH$_3$CN), and aerosol. This overview of the SPARC Data Initiative introduces the updated versions of the SPARC Data Initiative timeseries for the extended time period 1979-2018 and provides information on

the satellite instruments included in the assessment: LIMS, SAGE I/II/III, HALOE, UARS-MLS, POAM II/III, OSIRIS, SMR, MIPAS, GOMOS, SCIAMACHY, ACE-FTS, ACE-MAESTRO, Aura-MLS, HIRDLS, SMILES, and OMPS-LP. It describes the Data Initiative's top-down climatological validation approach to comparing stratospheric composition measurements based on zonal monthly mean fields, which provides upper bounds to relative inter-instrument biases and an assessment of how well the instruments are able to capture geophysical features of the stratosphere. An update to previously published evaluations of

ozone and water vapour monthly mean timeseries is provided. In addition, example trace gas evaluations of methane (CH$_4$), carbon monoxide (CO), a set of nitrogen species (NO, NO$_2$, and HNO$_3$), the reactive nitrogen family (NO$_y$), and hydroperoxyl (HO$_2$) are presented. The results highlight the quality, strengths and weaknesses, and representativeness of the different datasets. As an intended summary, the current state of our knowledge of stratospheric composition and variability is provided based on the overall consistency between the datasets. As such, the SPARC Data Initiative data and evaluations can serve as

an atlas or reference of stratospheric composition and variability during the 'golden age' of atmospheric limb sounding. The updated SPARC Data Initiative zonal monthly mean timeseries for each instrument are publicly available and accessible via the Zenodo data archive (Hegglin et al., 2020, doi:10.5281/zenodo.4265393).

*Copyright statement.* TEXT

# 1 Introduction

The past four decades starting in the late 1970s represent a 'golden age' of stratospheric composition measurements from satellite limb sounders, which capture the spatio-temporal structure of stratospheric composition with a vertical resolution of approximately 1 to 5 km. These limb observations have been used extensively to monitor the state of the stratospheric ozone layer that protects human and ecosystem health (e.g., Randel et al., 1999; Harris et al., 2015; WMO, 2011, 2014, 2018) and to study the processes leading to anthropogenic ozone depletion (e.g., Manney et al., 1994; Dessler et al., 1995; Santee et al.,

2008). Such research provided the crucial science basis that underpinned actions taken under the Montreal Protocol and its Amendments for the protection of the ozone layer, which is considered to be the most successful international treaty on an environmental issue to date. Limb observations, and merged products thereof, are also becoming increasingly important for the detection and attribution of climate change and potential feedback mechanisms, including the role of stratospheric water vapour and aerosol trends and variability in radiative forcing of climate (e.g., Solomon et al., 2010, 2011; Gilford et al., 2016;

IPCC, 2014; Schmidt et al., 2018). More generally, limb observations are used for the study of stratospheric dynamics and transport (e.g., Gray and Pyle, 1986; Solomon et al., 1986; Holton and Choi, 1988; Funke et al., 2005a; Manney et al., 2009),

empirical studies of stratospheric climate and variability (e.g., Randel et al., 2006, 2010, 2011; Manney et al., 2008; Hegglin et al., 2009; Bourassa et al., 2010; Stiller et al., 2012; Gille et al., 2014), data merging and trend evaluation activities (e.g., Randel and Wu, 1998; Hegglin et al., 2014; Shepherd et al., 2014; Froidevaux et al., 2015; Harris et al., 2015; Davis et al., 2016; Arosio et al., 2019; SPARC LOTUS report, 2019), with merged datasets also being used as forcing databases in climate

models (e.g., Cionni et al. (2011) for ozone; Thomason et al. (2018) for aerosol), and for the validation of the representation of transport and chemistry in numerical models (e.g., Eyring et al., 2006; Gettelman et al., 2010; Hegglin et al., 2010; Strahan et al., 2011; Kolonjari et al., 2018; Froidevaux et al., 2019).

The validity of any data and trend analysis, however, strongly depends on the understanding of the observational uncertainty and overall quality of the datasets used, which hitherto was deemed unsatisfactory (SPARC CCMVal, 2010). Uncertainty and

bias estimates are particularly important to inform chemical data assimilation systems (Inness et al., 2013; Errera et al., 2016) and to develop observational metrics for the evaluation of model performance (Douglass et al., 1999; Waugh and Eyring, 2009). In response to this need, the Stratosphere-troposphere Processes and their Role in Climate (SPARC) core project of the World Climate Research Programme (WCRP) initiated the SPARC Data Initiative with the aim to coordinate a comprehensive assessment of available vertically-resolved chemical trace gas and aerosol observations obtained from an international suite

of satellite limb sounders. The SPARC Data Initiative's main objectives were (1) to assess the availability of datasets, (2) to compile timeseries of vertically resolved, zonal monthly mean trace gas and aerosol fields, and (3) to perform a detailed inter-comparison of these timeseries, summarizing useful information and highlighting differences among datasets. The SPARC Data Initiative thereby complements other SPARC activities that have focused on the assessment of stratospheric ozone (e.g., Harris et al., 2015; SPARC, 2019), water vapour (SPARC, 2000; Khosrawi et al., 2018; Lossow et al., 2019), and aerosol (SPARC,

2006; Kremser et al., 2016). The provision of error estimates for atmospheric temperature and composition measurements from space following a unified methodological approach, which was highlighted by the SPARC Data Initiative (SPARC, 2017) to be a missing component of its analysis, is now the focus of the SPARC Towards Unified Error Reporting (TUNER) Initiative (von Clarmann et al., 2019). A first application of the SPARC Data Initiative zonal monthly mean timeseries is the evaluation of stratospheric ozone and water vapor in global reanalyses (Davis et al., 2017) as part of the SPARC Reanalysis Intercomparison

Project (S-RIP) (Fujiwara et al., 2017). SPARC Data Initiative gridded datasets have also been contributed to the annual State of the Climate reports in the Bulletin of the American Meteorological Society (Blunden and Arndt, 2018; 2019).

Here we present an update of the SPARC Data Initiative (SPARC, 2017), which focused on composition measurements from 1979-2010, extending its evaluation of the gridded data timeseries out to the end of 2018 (see Figure 1). The update features gridded datasets based on more recent retrieval versions and adds the observations of OMPS-LP (on SUOMI NPP) and SAGE

III/ISS to the original list of satellite limb sounders presented in SPARC (2017) (LIMS, SAGE I/II/III, HALOE, UARS-MLS, POAM II/III, OSIRIS, SMR, MIPAS, GOMOS, SCIAMACHY, ACE-FTS, ACE-MAESTRO, Aura-MLS, HIRDLS, and SMILES; see Section 2 for the full meaning of these acronyms). The gridded datasets include the stratospheric trace gases of primary interest, $O_3$ and $H_2O$, major long-lived trace gases ($SF_6$, $N_2O$, HF, $CCl_3F$, $CCl_2F_2$, $NO_y$), trace gases with intermediate lifetimes (HCl, $CH_4$, CO, $HNO_3$), and shorter-lived trace gases important to stratospheric chemistry including

nitrogen-containing species (NO, $NO_2$, $NO_x$, $N_2O_5$, $HNO_4$), halogens (BrO, ClO, $ClONO_2$, HOCl), and other minor species

(OH, HO$_2$, CH$_2$O, CH$_3$CN), and aerosol. The observations considered have been compiled on a common latitude-pressure grid, covering the region from the upper troposphere to the lower mesosphere (300-0.1 hPa) with a latitudinal resolution of 5°. A summary of the available trace gas and aerosol gridded datasets from each instrument is given in Figure 2. Almost half of these are based on newer data versions than those used in SPARC (2017) (highlighted in Figure 2 and with details provided in Tables 1 and 2). The data are published via Zenodo (doi:10.5281/zenodo.4265393). Note that early data versions of chemical trace gases (i.e., research products) are not included (except for the SAGE III/ISS H$_2$O product) and many more species could be made available. Also, there are a handful of early satellite limb sounders such as SAMS on Nimbus 7 (Jones et al., 1986), ISAMS (Taylor et al., 1993) and CLAES (Roche et al., 1993) on UARS, ATMOS (Gunson et al., 1996) and MAS (Hartmann et al., 1996) on the ATLAS Space Shuttle missions, and ILAS on ADEOS (Sasano et al., 1999) that could not be evaluated in this assessment due to a lack of resources and generally shorter timeseries than those from other datasets.

The paper is organised as follows. Section 2 provides information on the participating satellite instruments, which vary in terms of measurement method, geographical coverage, spatial and temporal sampling and resolution, time period, and retrieval algorithm. The methodology used to create and compare the trace gas and aerosol imeseries is described in Section 3. The SPARC Data Initiative introduced a top-down climatological validation approach to the evaluation of stratospheric composition measurements (Hegglin et al., 2013; Tegtmeier et al., 2013; SPARC, 2017), based on the comparison of gridded trace gas and aerosol datasets. This top-down approach complements (but does not replace) the more traditional validation approach that uses coincident profile measurements and sometimes focuses on bottom-up error budgets to characterise measurement uncertainty. The top-down climatological validation approach has the advantages that it is consistent between all instruments, avoids sensitivity to arbitrary coincidence criteria, and generally produces larger sample sizes, which minimises the random part of the measurement error (or in other words cancels any kind of random fluctuations). The information gained from the SPARC Data Initiative approach thereby allows us to obtain upper bounds of systematic biases between instruments by reducing the noise from single measurements through averaging. Importantly, it enables assessing the latitude-dependence of these systematic biases. This work also provides unique information on how well the different instruments are capable of capturing distinct chemical and geophysical features in stratospheric composition, with the consistency among the instruments constraining our current knowledge of the state of the stratosphere.

Section 4 includes example trace gas evaluations of the longer-lived trace gases ozone (Section 4.1), water vapour (Section 4.2), and methane (CH$_4$; Section 4.3); and the medium- to shorter-lived trace gases carbon monoxide (CO; Section 4.4), nitrogen-containing species (NO, NO$_2$, NO$_x$, HNO$_3$, and NO$_y$; Section 4.5), and also hydroperoxyl (HO$_2$; Section 4.6). These evaluations all use updated versions of the datasets used in SPARC (2017), with differences to the old versions highlighted. A summary and conclusions of the updated and evaluated SPARC Data Initiative data, including an overview of our knowledge of the mean state of atmospheric trace gas distributions, are given in Sections 5 and 6. Note that, due to the complicating factor that aerosol extinction measurements are wavelength-dependent, the aerosol evaluations are based on a modified comparison approach, which will be presented in a follow-on publication. In addition to this paper, a special issue in the Journal of Geophysical Research (JGR) - Atmospheres on the SPARC Data Initiative has presented the evaluations of water vapour (Hegglin et al., 2013), ozone (Tegtmeier et al., 2013), the comparison of ozone from limb sounders with the nadir-viewing

Aura-TES instrument (Neu et al., 2014), an assessment of the impact of instrument-specific sampling patterns on measurement bias (Toohey et al., 2013), the dependence of the standard error of the mean on the sample size for profiles obtained with a non-random sampling pattern (Toohey and von Clarmann, 2013), and a single instrument study on SMILES observations (Kreyling et al., 2013). The reader is also referred to the WCRP SPARC Data Initiative Report (SPARC, 2017) which offers the complete assessment of all the different original atmospheric trace gas observations and aerosol, and is accessible via dx.doi.org/10.3929/ethz-a-010863911.

## 2 Satellite Instruments

The SPARC Data Initiative (SPARC, 2017) originally evaluated observations from 18 different satellite limb sounders and additionally, the nadir sounder Aura-TES. The latter instrument was used for comparisons in the upper troposphere and lower stratosphere (UTLS) only, focusing on the comparability between limb (with high vertical resolution measurements) and nadir sounders (with high horizontal resolution measurements) applying observation operators (Neu et al., 2014). In this update, TES is no longer included, but the instruments SAGE III on the ISS (hereafter SAGE III/ISS) and OMPS-LP on SUOMI NPP are added for evaluations including trace gas datasets between 2011 and 2018.

The instruments considered here all use passive remote sensing techniques, which are based on the detection of natural radiation emitted from the Sun or stars, or from the atmosphere itself (unlike active sounders such as LIDARs). The different instruments can be classified according to their observation geometry (limb emission, solar or stellar occultation, limb scattering or nadir emission) and the wavelengths they are measuring at, as compiled in Table 3. In the following, we provide a short description of each instrument, with the most important instrument characteristics summarized in Tables 4 and 5, and the representative sampling patterns provided in Figure 3. Note that the vertical range observed can depend on the retrieved species. Further information on the instrument and retrieval algorithms can be found in the SPARC Data Initiative report (SPARC, 2017).

### 2.1 LIMS on Nimbus 7

The Limb Infrared Monitor of the Stratosphere (LIMS) instrument was launched onboard the Nimbus 7 satellite in October 1978 (Gille and Russell, 1984). The spacecraft occupied a Sun-synchronous orbit, crossing the ascending node at $\sim$1:00 pm local time (LT) and the descending node at $\sim$11:00 pm LT, taking observations from 64°S to 84°N latitude. LIMS used broadband radiometry to observe infrared limb emission, with two radiometer channels for sensing temperature (atmospheric $CO_2$) centred near 15 $\mu$m, and further four channels for sensing trace gases: 6-7 $\mu$m for $H_2O$ and $NO_2$, 9-10 $\mu$m for $O_3$, and 11-12 $\mu$m for $HNO_3$ (Remsberg et al., 2004). LIMS obtained radiance profiles at every $\sim$0.8 degree latitude along its orbital, tangent-point tracks, yielding $\sim$260 profiles per orbit with $\sim$14 orbits per day. LIMS operated successfully from launch through its end date in May 1979, when there was final depletion of the cryogen gas supply for cooling its detectors.

## 2.2 SAGE I on AEM-2, SAGE II on ERBS, SAGE III on Meteor-3M, and SAGE III on the ISS

The Stratospheric Aerosol and Gas Experiment (SAGE) series of instruments consists of five instruments including the Stratospheric Aerosol Measurement (SAM II) on Nimbus 7 that span the period from 1978 through 2005 (McCormick, 1989) and after a pause continuing from 2017 to present. Note that SAM II has not been included in the SPARC Data Initiative evaluations.

The Stratospheric Aerosol and Gas Experiment I (SAGE I) was launched onboard the Applications Explorer Mission-B (AEM-B) satellite in February 1979 (McCormick et al., 1979). The spacecraft was in a ∼600 km orbit with an inclination of 560° that allowed for solar occultation measurements from 79°S to 79°N. The SAGE I instrument had four spectral channels centered at wavelengths of 1000, 600, 450, and 385 nm for measurements of aerosol extinction, $O_3$ and $NO_2$ concentration profiles. SAGE I made 15 sunrise and 15 sunset measurements per day that each covered a narrow latitude band and were

separated by ∼24° in longitude. It took ∼1.5 month for SAGE I sampling location to shift from one latitude extreme to the other. While there were sunset measurements during the entire 34-month lifetime of SAGE I, there were only 6 month of sunrise observations due to a spacecraft power problem early in the mission. SAGE I ceased operation in November 1981 due to a power system failure.

The Stratospheric Aerosol and Gas Experiment II (SAGE II) was launched onboard the Earth Radiation Budget Satellite

(ERBS) in October 1984 (Mauldin III et al., 1985; McCormick et al. 1989). The spacecraft occupied a 57° inclined orbit at an altitude of ∼610 km that allowed for observations from 80°S to 80°N. The SAGE II instrument was a broadband spectrometer that operated in the spectral range of ∼375-1030 nm for aerosol and trace gas observations (Mauldin et al., 1985). SAGE II measured 15 sunrise and 15 sunset measurements each day that covered a narrow latitude band and are separated by ∼24° in longitude. After late 2000, an azimuthal pointing problem resulted in the instrument operating at half-duty cycle. The ERBS

mission was decommissioned in October 2005.

The Stratospheric Aerosol and Gas Experiment III (SAGE III/M3M) was launched onboard the Russian Meteor-3M (M3M) spacecraft in December 2001 (Mauldin et al., 1998). The spacecraft was placed on a Sun-synchronous orbit, with an altitude of ∼1020 km, inclination of 99.50°, and equatorial crossing time (ascending node) at 9:15 am local time (LT). The SAGE III/M3M provided both solar and lunar measurements, with satellite sunrise events at 60 to 30°S and satellite sunset events at

45 to 80°N. Lunar events varied from pole to pole. The SAGE III instrument used a grating spectrometer that operated in the spectral range of ∼295-1025 nm and a single photodiode near 1550 nm for aerosol and trace gas observations (Mauldin et al., 1998). The M3M spacecraft ceased functioning in January 2006.

The Stratospheric Aerosol and Gas Experiment III on the International Space Station (SAGE III/ISS) is the second instrument from the SAGE III project (Mauldin et al., 1998). It was launched on the SpaceX Falcon 9 spacecraft in February 2017. Unlike

the first SAGE III instrument on the Meteor 3M spacecraft (SAGE III/M3M), SAGE III/ISS is in a mid-inclination orbit (51.6°). The solar observations can provide near global (70°S-70°N) measurements on a monthly basis with sampling similar to that of the SAGE II measurements. The SAGE III/ISS uses a grating spectrometer operating between ∼280 and ∼1035 nm as well as a single photodiode covering 1542 nm ±15 nm to retrieve aerosol and other trace gases (SAGE III ATBD, 2002). It can provide vertical profiles of $O_3$, $H_2O$, $NO_2$ and aerosol extinctions at multiple wavelengths through the solar occultation

technique. The lunar occultation measurements can augment the sampling of solar observations with measurements of $O_3$, $NO_2$, as well as $NO_3$ and $ClO_2$. The sampling pattern and resulting monthly and annual sampling density of SAGE III/ISS is shown in Figure 4, equivalent to what is shown for the other instruments in Chapter 2 of the SPARC Data Initiative report (SPARC, 2017).

## 2.3 HALOE on UARS

The Halogen Occultation Experiment (HALOE) was launched onboard the Upper Atmosphere Research Satellite (UARS) in September 1991 (Russell et al., 1993). The spacecraft occupied a 57° inclined orbit at an altitude of ∼585 km that allowed for observations from 80°S to 80°N. The HALOE instrument used a combination of broadband radiometry and gas filter correlation techniques to observe several trace gas species in the spectral range of ∼2.4-10.4 $\mu$m. HALOE measured 15 sunrise and 15 sunset events per day and achieved near-global coverage in approximately a month. The daily measurement spacing was equal in longitude and varied seasonally in latitude. The UARS mission was decommissioned in December 2005.

## 2.4 MLS on UARS

The Upper Atmosphere Research Satellite Microwave Limb Sounder (UARS-MLS) was also launched onboard UARS in September 1991 (Barath et al., 1993, Waters et al., 1993, 1999). The spacecraft's orbit (see Section 2.3) allowed for MLS observations in two sets of latitude bands, alternating roughly every 36 days (as governed by spacecraft yaw maneuvers) between mostly northern hemisphere and mostly southern hemisphere latitudes, with full coverage of low latitudes at all times. UARS-MLS performed microwave thermal emission measurements using antenna scans of the Earth's limb and three radiometers to detect spectral line and continuum signals (at 1.45, 1.63, and 4.76 mm wavelengths) and to retrieve profiles of upper atmospheric temperature, trace gases, as well as upper tropospheric (UT) $H_2O$ and cloud ice water content. UARS-MLS provided more than 1300 profiles (per species) along the sub-orbital track every day, during both daytime and nighttime. The UARS-MLS measurements became increasingly sparse after 1994 in order to preserve the antenna scanning mechanism and as a result of UARS battery power limitations. The last UARS-MLS profiles were obtained in 2001, before UARS was officially decommissioned in December 2005.

## 2.5 POAM II/III on SPOT-3/4

The Polar Ozone and Aerosol Measurement II (POAM II) was launched onboard the SPOT-3 spacecraft in September 1993 (Glaccum et al., 1996). The spacecraft occupied a Sun-synchronous orbit, crossing the descending node at 10:30 am LT, that allowed for observations in two latitude bands at 88 to 62°S and 65 to 71°N. The POAM II instrument used broadband radiometry to observe trace gases and aerosols in the spectral range of ∼350-1070 nm. POAM II used the solar occultation technique and made 14 measurements per day in each hemisphere, equally spaced in longitude around a circle of approximately constant latitude. Satellite sunrise measurements were made in the Northern Hemisphere (55°N-71°N) and sunsets in the Southern Hemisphere (63°S-88°S). The latitude coverage changes slowly with season and is exactly periodic from year to

year. The SPOT-3 spacecraft ceased functioning in November 1997. POAM II produced a three-year data set (1993-1996) of polar stratospheric $O_3$, $H_2O$, $NO_2$, and aerosols.

The Polar Ozone and Aerosol Measurement III (POAM III) was launched onboard the SPOT-4 spacecraft in March 1998 (Lucke et al., 1999). The spacecraft occupied a Sun-synchronous orbit, crossing the descending node at 10:30 am LT, that allowed for observations in two latitude bands at 88 to 62°S and 65 to 71°N. The POAM III instrument used broadband radiometry to observe aerosol and trace gases in the spectral range of ∼345-1030 nm (Lucke et al., 1999). POAM III has exactly the same sampling pattern as POAM II. The POAM III instrument ceased functioning in December 2005. POAM III produced an eight-year data set (1998-2005) of polar stratospheric $O_3$, $H_2O$, $NO_2$, $O_2$ (total density), and aerosols.

## 2.6 OSIRIS on Odin

The Optical Spectrograph and InfraRed Imaging System (OSIRIS) was launched onboard the Odin satellite in February 2001 (Murtagh et al., 2002; Llewellyn et al., 2004). The spacecraft occupies a 97.8° inclined, Sun-synchronous orbit, crossing the ascending node near 6:00 pm LT that allows for near-global observations between 82°S and 82°N. The OSIRIS spectrograph has a single line of sight that vertically scans the Earth's limb measuring the spectral radiance of scattered sunlight in the spectral range of 290-810 nm. OSIRIS provides approximately 500 profiles of aerosol and trace gases per day along the orbital track during daytime. Tropical latitudes are sampled throughout the year, but due to the seasonally changing solar illumination conditions at the tangent point of the observation, the coverage of mid and high latitudes is limited to the sunlit hemisphere in the summer and winter, with near global coverage for about one month around each equinox. The latitude coverage changes slowly with the degradation of the orbit but follows essentially the same pattern in each year. OSIRIS reached 20 years in orbit in February 2021 and continues operation at the time of writing.

## 2.7 SMR on Odin

The Sub-Millimetre Radiometer (SMR) was also launched onboard the Odin satellite in February 2001 (Murtagh et al., 2002). See Section 2.6 for details on the satellite's orbit. The SMR has a single line of sight that vertically scans the Earth's limb measuring the thermal emission of the atmosphere in the 0.55 mm wavelength region. SMR provides approximately 900 profiles of trace gases per day along the orbital track. Not all gases can be measured simultaneously but rather the tuning of the instrument is varied on a daily basis to optimise the various science goals. Thus, while some species such as $O_3$ are measured on a close to daily bases others are only measured a few times per month. SMR reached 20 years in orbit in February 2021 and continues operation at the time of writing.

## 2.8 GOMOS on Envisat

The Global Ozone Monitoring by Occultation of Stars (GOMOS) instrument was launched onboard the Envisat spacecraft in March 2002 (Bertaux et al., 2010). The spacecraft occupied a 98.55° inclined, Sun-synchronous polar orbit, crossing the descending node at 10:00 am LT that allowed GOMOS global night-time observations. The GOMOS instrument used a grating

spectrometer to observe trace gases $O_3$, $NO_2$, $NO_3$, and aerosols in the spectral range of 248-690 nm. GOMOS used the stellar occultation method and made 100-200 nighttime occultations per day. The latitude coverage of GOMOS was global, except for the summer-time polar regions. The Envisat spacecraft ceased functioning in April 2002.

## 2.9 MIPAS on Envisat

The Michelson Interferometer for Passive Atmospheric Sounding (MIPAS) was also launched onboard Envisat in March 2002 (Fischer et al., 2008). See Section 2.8 for details on the satellite's orbit that allowed MIPAS to attain global daytime and nighttime limb emission measurements. MIPAS was a Fourier transform spectrometer that operated in the spectral range of 4.3-15 $\mu$m wavelength region for trace gas, temperature, and aerosol observations (Fischer et al., 2008). From 2002-2004, MIPAS recorded one limb scan of spectra each 510 km and provided about 1000 vertical profiles per day. From 2005-2012, the along-track horizontal spacing was 410 km, however at slightly degraded spectral while improved vertical resolution. In this paper data produced with the processor developed and operated by the Institut für Meteorologie und Klimaforschung (IMK) in cooperation with the Instituto de Astrofísica de Andalucía are used (von Clarmann et al., 2009). Several other MIPAS retrieval products are available (see Lossow et al., 2019), however, were not contributed to the SPARC Data Initiative in the required format. Note, the IMK-processor also provides more species than these other processors.

## 2.10 SCIAMACHY on Envisat

The Scanning Imaging Absorption spectroMeter for Atmospheric CHartographY (SCIAMACHY) (Burrows et al., 1995; Bovensmann et al., 1999) was also launched onboard Envisat in March 2002. See Section 2.8 for details on the satellite's orbit that allowed SCIAMACHY to attain observations between 85°S and 85°N (65° in the winter hemisphere). The SCIAMACHY instrument was an eight-channel passive imaging grating spectrometer that observed aerosol and trace gases in the spectral range of ~214-2386 nm. SCIAMACHY used the limb scattering, nadir backscattering, and solar occultation techniques, although only the results from limb scattering measurements are used in this study. In the limb scattering mode SCIAMACHY made over 1000 measurements per day. Cross-track scans, each consisting of four measurements, are equally spaced in latitude and longitude. The latitude coverage changes slowly with season and is exactly periodic from year to year. Measurements performed within the South Atlantic Anomaly were rejected. From the measurements in the limb viewing geometry, SCIA-MACHY produced an almost ten-years data set (2002-2012) of stratospheric $O_3$, $H_2O$, $NO_2$, BrO and aerosols.

## 2.11 ACE-FTS on SciSat-1

The Atmospheric Chemistry Experiment-Fourier Transform Spectrometer (ACE-FTS) was launched onboard the SciSat-1 spacecraft in August 2003 (Bernath, 2006). The spacecraft occupies a drifting orbit at an inclination of 74° that allows for observations from to 85°S and 85°N. The ACE-FTS instrument is a high-resolution (0.02 cm$^{-1}$) FTS measuring the full spectral range between 750 and 4400 cm$^{-1}$ to measure the chemical composition of the atmosphere (Bernath et al., 2005). The ACE-FTS uses the solar occultation technique to measure approximately 15 sunrise and 15 sunset occultations per day

and achieves global latitude coverage over a period of three months (i.e., one season). The latitude coverage is almost exactly periodic from year to year. At the time of writing, measurements from the ACE-FTS are ongoing.

## 2.12   ACE-MAESTRO on SciSat-1

The Measurement of Aerosol Extinction in the Stratosphere and Troposphere Retrieved by Occultation (MAESTRO) was
launched together with the ACE-FTS onboard the SciSat-1 spacecraft in August 2003 (McElroy et a., 2007). See Section 2.11 for details on the satellite's orbit. The MAESTRO instrument consists of a dual grating spectrometer to observe trace gases and aerosols in the spectral range of ∼280-1030 nm. MAESTRO uses the solar occultation technique and makes 15 sunrise and 15 sunset measurements per day, equally spaced in longitude. The two ACE instruments take simultaneous measurements of the same air mass using a common Sun-tracking mirror that is located within the ACE-FTS. At the time of writing, measurements
from MAESTRO are ongoing.

## 2.13   MLS on Aura

The Aura Microwave Limb Sounder (Aura-MLS) was launched onboard the Aura satellite in July 2004 (Waters et al., 1999; 2006). The spacecraft occupied a 98° inclined near-polar, Sun-synchronous orbit, with a 1:45 pm LT ascending node equator crossing time that allows for observations from about 80°S to 80°N on a daily basis. Aura-MLS, similar to its UARS prede-
cessor version (see Section 2.4), performs microwave thermal emission measurements using antenna scans of the Earth's limb and five radiometers to detect spectral line and continuum signals (at 0.47, 1.25, 1.58, and 2.54 mm wavelengths, along with measurements at 0.12 mm of OH) and to retrieve profiles of upper atmospheric temperature and many trace gases, as well as cloud ice water content. Aura-MLS provides about 3500 profiles (per species) along the sub-orbital track every day, during both daytime and nighttime. At the time of writing, measurements from two of the Aura instruments (MLS and the Ozone
Monitoring Instrument, OMI) are ongoing.

## 2.14   HIRDLS on Aura

The High Resolution Dynamics Limb Sounder (HIRDLS) instrument was also launched onboard Aura in July 2004 (Gille and Barnett, 1992). See Section 2.13 for details on the satellite's orbit. Unfortunately during launch, a plastic film became detached and blocked the path between the scan mirror and the aperture of HIRDLS (Gille et al., 2008), reducing coverage to latitudes
from about 63°S to 80°N on a daily basis, with observing times at 3 pm and 12 am. HIRDLS was a limb-scanning infrared radiometer and observed temperature, 10 trace gases and aerosols in 21 broad spectral channels at wavelengths between 6.12 to 17.76 $\mu$m. HIRDLS observed approximately 6400 profiles each day, with profiles spaced approximately every 100 km along the orbit track. HIRDLS stopped acquiring data on 17 March 2008 due to a chopper failure. Useful HIRDLS data began in January 2005, and ended at the end of December 2007.

## 2.15 SMILES on the ISS

The Superconducting Submillimeter-Wave Limb Emission Sounder (SMILES) was installed on the International Space Station (ISS) in September 2009 (Kikuchi et al., 2010). As mentioned in Section 2.2, the ISS is in a circular, mid-inclination orbit (at $51.6°$). With the SMILES antenna mounted so that its field-of-view is $45°$ to the left of the orbital plane, the observed latitude region was increased to between $38°$S and $65°$N. Three times during the observation period, in late November, middle of February, and beginning of April, the ISS turned $180°$ along its yaw axis, so that the field-of-view deflection was pointing southward, resulting in inverse hemispheric observation ranges ($65°$S-$38°$N). Three times SMILES was the demonstration of ultrasensitive sub-mm limb emission observations with a 4 K-cooled receiver system. 1630 observation points were obtained per day. The non-Sun-synchronous orbit of the ISS allowed the instrument to observe the diurnal variation of minor short-lived species. The instrument was in operation between October 2009 and April 2010.

## 2.16 OMPS-LP on Suomi NPP

The advanced Ozone Mapping and Profiler Suite (OMPS) was launched onboard the Suomi National Polar-orbiting Partnership (NPP) spacecraft in 2002 (Jaross et al. 2014). The spacecraft occupies a Sun-synchronous orbit with a 1:30 pm ascending node equator crossing time, that allows for observations between $81.5°$S and $81.5°$N (limited to $60°$ in the winter hemisphere). OMPS consists of three spectrometers: a downward-looking nadir mapper, nadir profiler and limb profiler. Only the measurements from the latter instrument (OMPS-LP) are used in this study. The OMPS-LP instrument is equipped with a 2D imaging prism spectrometer and observes aerosol and trace gases in the spectral range of $\sim$280-1000 nm through 3 entrance slits separated horizontally by $4.25°$ (about 250 km). OMPS-LP uses the limb scattering technique and makes about 2500 measurements per day (with each of 3 entrance slits), equally spaced in latitude and longitude. The latitude coverage changes slowly with season and is exactly periodic from year to year (see Figures 3 and 4). Measurements affected by the South Atlantic Anomaly are rejected. The Suomi-NPP spacecraft is still in operation at the time of writing. OMPS-LP produces a data set (2012-present) of stratospheric $O_3$ and aerosols.

It should be noted that the two OMPS-LP ozone datasets used in the SPARC Data Initiative are based on different retrieval algorithms, IUP-OMPS (Arosio et al., 2018) and USask-OMPS (Zawada et al., 2018). The main difference between these two products is that USask is retrieved using a 2D tomographic algorithm and IUP uses a standard 1D algorithm. Furthermore, the spectral information and associated tangent height ranges are used differently. NASA also produces a stratospheric ozone product from OMPS-LP (Rault and Loughman, 2013) which is not included in the SPARC Data Initiative.

## 3 Gridded Dataset Construction and Evaluation Methodology

In the following, a short summary of the method used to compile the SPARC Data Initiative zonal monthly mean timeseries is provided. More detailed information on the instrument-specific data preparation and handling can be found in the *SPARC Data Initiative* report (SPARC, 2017, chapter 3, pages 30-36).

### 3.1 Gridded Dataset Construction and Uncertainty

Zonal monthly mean time series of each trace gas species (in volume mixing ratio, *VMR*) and aerosol (as extinction ratio) have been calculated for each instrument on the SPARC Data Initiative dataset grid, using 5-degree latitude bins (with mid-points at 87.5°S, 82.5°S, 77.5°S, ..., 87.5°N) and 28 pressure levels (300, 250, 200, 170, 150, 130, 115, 100, 90, 80, 70, 50, 30, 20, 15, 10, 7, 5, 3, 2, 1.5, 1, 0.7, 0.5, 0.3, 0.2, 0.15, and 0.1 hPa). To this end, profile data have been carefully screened before binning and a hybrid log-linear interpolation in the vertical has been performed (i.e., the VMR is interpolated linearly in log-pressure). For instruments that provide data on an altitude grid, a conversion from altitude to pressure levels is performed using retrieved temperature/pressure profiles (as is the case for MIPAS, ACE-FTS, and ACE-MAESTRO) or meteorological analyses (ECMWF for OSIRIS, GOMOS, and SCIAMACHY, NCEP for SAGE I and III/M3M, MERRA for SAGE II, MERRA-2 for SAGE III/ISS, UKMO for POAM II/III, and GMAO/GEOS-5 for OMPS-LP). Similarly, this information is used to convert retrieved number densities into *VMR*, where needed. It should be noted, that using different ancillary data for the grid- and unit-conversions will introduce an additional source of uncertainty, which has not been quantified here (see also discussion in Hubert et al., 2016). Any known problems in the ancillary temperature/pressure data that were used to convert measured species from their native units to VMR and pressure grids, however, have been fixed by an updated retrieval algorithm or minimized with empirical corrections. For example, problems in the older SAGE II (v6.2) temperature/pressure auxiliary files, mainly in the tropics above 2 hPa, were empirically corrected (Froidevaux et al., 2015) before being incorporated in the SPARC Data Initiative gridded dataset (SPARC, 2017). The anomalous temperature problem in SAGE II (v6.2) has been fixed in the latest v7.0 retrieval, which is used in the updated SPARC Data Initiative gridded dataset and this manuscript. Both SAGE III/ISS (v5.1) and SAGE II (v7.0) data were also updated to remove/minimize the effects of altitude registration errors in the auxiliary temperature profiles (Wang et al., 2020).

Along with the zonal monthly mean value, the standard deviation and the number of averaged data values are given for each grid point, as well as the average day of month, and the minimum, mean, and maximum local solar times for these values (see Figure 5 and Table 6 for an illustration and summary of the variables included in each SPARC Data Initiative dataset file). Note that the methodology for the calculation of the ACE-FTS gridded datasets has changed since SPARC (2017). While for the older gridded datasets, data were binned for each midpoint between the Data Initiative pressures levels, interpolation to these levels is now used (matching what has been done in Koo et al., 2017). The methodology for the calculation of ACE-MAESTRO datasets is done in the same way as for ACE-FTS. For the new SAGE III/ISS gridded datasets, the same approach was followed as for the other SAGE instruments. For OMPS, the observations are handled in exactly the same way as those from SCIAMACHY with exception of the rejection of measurements within the South Atlantic Anomaly (SAA) region. While for SCIAMACHY, a fixed latitude-longitude range is used, the SAA flags from the level-1 product are used for OMPS.

Interpretation of the differences between the individual trace gas and aerosol datasets will need to take into account several sources of uncertainty, including systematic errors of both the measurements and the dataset construction. Random measurement errors have little impact on the zonal monthly means, however measurement biases (e.g., related to retrieval errors) will introduce systematic differences between an individual instrument's mean field and the truth. Differences in the mean fields

from the truth arise also from sampling biases (see *Toohey et al.* (2013) for the SPARC Data Initiative; *Sofieva et al.* (2014), *Míllan et al.* (2016), *Míllan et al.* (2018), and *Kloss et al.* (2019) for other related studies) and differences in the averaging technique used to produce the gridded datasets (*Funke and von Clarmann*, 2012). Since the overall uncertainty of the gridded data is not accessible in a consistent way from bottom-up estimates for all of the instruments included in the SPARC Data

Initiative (a task now being addressed by SPARC TUNER), we use here as an approximate measure of the uncertainty in each zonal monthly mean field, the standard error of the mean (*SEM*):

$$SEM = \sigma/\sqrt{n}, \tag{1}$$

where $\sigma$ is the standard deviation of the measurements and $n$ the number of measurements at each grid point. The range of twice the *SEM* can be roughly interpreted as the 95% confidence interval of the monthly mean under the assumption of Gaussian statistics and independent errors. Although sampling patterns and densities differ greatly between different instruments, the

*SEM* has been shown to generally produce a conservative estimate of the true random error in the mean for both solar occultation and dense sampling patterns (*Toohey and von Clarmann*, 2013). This is due to the fact that sampling by satellite instruments is roughly uniform with respect to longitude. It should be noted, however, that the *SEM* does not reflect the potential influence of irregular or incomplete sampling of the month and latitude band, which can produce sampling biases in the mean fields (*Toohey et al.*, 2013).

## 3.2   Evaluation Methodology

### 3.2.1   Climatological Validation Approach

The SPARC Data Initiative introduces a complementary approach of testing data quality using vertically resolved, zonal monthly mean gridded datasets of trace gas observations for comparison, rather than using profile-to-profile evaluations based on measurement coincidences, which has been done extensively in the literature. We coin this methodology with the term

*climatological validation approach* where 'climatological' in this context is not used to refer to a time-averaged climate state (which should be reproduced by free-running models, averaged over many years) but to year-by-year values (which free-running models would not be expected to match). The climatological approach was chosen because multiple measurements can in principle be averaged to reduce random measurement errors, leaving the systematic error (or bias, although it needs to be noted that here this bias is defined as relative to the multi-instrument mean and not an absolute truth). Comparing these

gridded datasets has the advantage of removing much of the natural variability inherent to trace gas observations from both in-situ sensors and measurements from space (Hegglin et al., 2008; Hegglin et al., 2013; Tegtmeier et al., 2013) and yields information on the behaviour of the retrievals resolved in latitude and height. In addition, monthly mean comparisons allow for testing how well the instruments' measurement characteristics are capable of resolving geophysical features (e.g., interannual variability, seasonality, or periodicities). The climatological validation approach is applied to all evaluations and its advantages

and disadvantages will be discussed where appropriate.

However, when using the climatological validation approach, some general guidelines should be followed. As highlighted in the sampling study by Toohey et al. (2013) as an integral part of the SPARC Data Initiative, sampling biases in the gridded datasets may contribute to the derived biases; this requires careful consideration in the interpretation of the results, as was attempted throughout SPARC (2017) and also in this update at least in a qualitative manner. Toohey et al. (2013) investigated the impact of 15 of the here presented instrument's sampling patterns (not including LIMS, SMR, OMPS or SAGE III/ISS) on the gridded datasets using chemical fields from a chemistry climate model as idealised truth. The evaluation found sampling biases of up to 10% for $O_3$ monthly means, and up to 20% for annual means for some instruments, generally in atmospheric regions with high natural variability such as the high-latitude stratosphere or the the UTLS. Longer-lived species with lower variability such as $H_2O$ show smaller sampling biases (except in the UTLS). Non-uniform sampling in both space and time thereby contribute to these sampling biases. While Toohey et al. (2013) have characterised the sampling biases for ozone and water vapour only, the resulting bias patterns can be taken as guidelines for trace gases with similar source and sink characteristics (or lifetimes). Similar findings have been highlighted by Damadeo et al. (2018), particularly for data sets with very sparse sampling patterns.

Another important aspect of our approach is that trace gas timeseries are compared without any modification, such as the application of averaging kernels, to account for different vertical resolutions. We consider our simplified approach as justified, because in most cases the vertical resolutions of the limb sounders are quite similar, and the degree to which a priori information influences the retrieved profiles is usually limited. Exceptions are discussed where they appear.

Furthermore, highly structured and transient features, which can for example arise from different modes of natural variability such as the Quasi-Biennial Oscillation (QBO), the El Niño-Southern Oscillation (ENSO) (e.g., Diallo et al., 2019), or Stratospheric Sudden Warmings (SSWs) (e.g., Manney et al., 2009) and which may not be resolved by some instruments, will most likely average out in the zonal monthly mean fields. Nonetheless, it is best to compare zonal mean fields averaged over the exact same years and for the maximum time period for which all instruments overlap (ideally for more than 4-5 years). When this is not possible, as many years as possible should be included, keeping in mind a potential tradeoff with underlying trends in a given trace gas over the time period considered. For most species, SPARC (2017) concluded that expected trends are generally smaller than inter-instrument differences.

Where the instruments' temporal coverage allowed for it, inter-instrument differences should be tested for different time periods to get a sense of the influence of temporal inconsistencies in the comparison. Again, SPARC (2017) concluded that the general structure in the different instruments' biases relative to another did not significantly change. However, there are some examples where the previous conclusion was not applicable. SAGE II versus HALOE differences in particular show inter-instrument differences changed over time, which was indicative of a drift in one of the instruments or an influence of volcanic aerosol that could not be fully accounted for in the retrieval. Note, in this case, the change in the biases was not attributable to sampling, since the instruments were compared over the same time periods (see SPARC, 2017).

Finally, within the SPARC Data Initiative, agreement between instruments is defined using the terminology specified in Table 7. All these numbers indicating a certain level of agreement are with respect to the multi-instrument mean (MIM, see

Section 3.2.3), so that where two instruments show excellent agreement of $\pm 2.5\%$, the two instruments could show a maximum difference of 5% between them.

### 3.2.2 Evaluation Diagnostics

A set of standard diagnostics is used to investigate the differences between the time series obtained from the different instruments. The diagnostics include comparisons of annual or zonal monthly mean trace gas fields, vertical and meridional mean profiles, seasonal cycles for a single year or averaged over multiple years, and multi-annual averages of latitude-month evolution. Additional evaluations of inter-annual variability and known tracer-specific features (such as the tape-recorder signal in water vapour or the QBO signal in ozone) which test the physical consistency of the datasets, were also carried out and those not presented here can be found in SPARC (2017). The evaluation methods for the trace gas species time series and more examples are more thoroughly described in Hegglin et al. (2013), Tegtmeier et al. (2013), and SPARC (2017).

### 3.2.3 Multi-Instrument Mean (MIM) Reference

The SPARC Data Initiative's approach is to use the multi-instrument mean (*MIM*) as a reference to which all instruments are compared. The *MIM* is calculated by taking the annual or monthly mean of all available instrument datasets within a given time period of interest, aiming at maximum spatial and temporal data coverage for each instrument in order to limit the impact of sampling bias.Note that the *MIM* does not represent the best estimate of the atmospheric state, rather is motivated by the need that it does not favour a certain instrument. Most datasets are included in its calculation regardless of their quality and without any weighting applied to them. In particular, the datasets from instruments with sparse sampling have the same weight as datasets from instruments with much higher sampling in the calculation of the *MIM*. Only if measurements from a particular instrument are deemed unrealistic (i.e., outside the $\pm 3\sigma$ range), or if another version of a specific trace gas data product is available from the same instrument, are they not included. The relative percentage differences between the trace gas mixing ratios of an instrument ($\chi_i$) and the *MIM* ($\chi_{MIM}$) are then given by:

$$100 * (\chi_i - \chi_{MIM})/\chi_{MIM}. \tag{2}$$

One always has to keep in mind when interpreting relative differences with respect to the *MIM* that the composition of instruments from which the *MIM* was calculated may have changed between time periods. Hence, changes in derived differences are not to be interpreted as changes in the performance (or drifts) of an individual instrument. Also, if there is an unphysical behaviour in one instrument, the *MIM* and thus the differences with respect to the *MIM* of the other instruments will most certainly reflect this unphysical behaviour as well, although we have tried to eliminate the largest outliers. Finally, if one instrument does not have global coverage for every month some sampling biases may be introduced into the *MIM* (see discussion in Section 3.2.1). Due to its changing nature, the *MIM* is thus not made available via the Zenodo data archive.

### 3.2.4 Summary Evaluation

Finally, a summary evaluation (seen in Figures 15-17 and discussed in Section 5) is presented, which provides an estimate of the uncertainty in our knowledge of the atmospheric mean state of a given trace gas. This uncertainty is expressed as the relative standard deviation (i.e., calculated relative to the *MIM*) over all instrument values at a given latitude-pressure grid point, or in other words, the spread between the datasets around the *MIM*.

## 4 Examples of SPARC Data Initiative Trace Gas Evaluations

The approach of the SPARC Data Initiative for evaluating chemical trace gas datasets from stratospheric limb sounders is illustrated in the following providing updates to the ozone (Tegtmeier et al., 2013) and water vapour evaluations (Hegglin et al., 2013), and presenting additional examples based on $CH_4$, CO, different nitrogen-containing species like NO, $NO_2$, $HNO_3$, and $NO_y$, and $HO_2$ measurements. These species were chosen to highlight particular differences in the evaluation approach that were necessary to account for the wide range of average stratospheric lifetimes valid for the lower stratosphere among the species considered (e.g., 8 years for $CH_4$, 3 months for CO, seconds for $HO_2$). Note, the definitions and abbreviations of different altitude regions in the atmosphere as used throughout this study is given in Table 8.

### 4.1 Ozone ($O_3$)

Ozone is one of the most important trace species in the stratosphere due to its absorption of biologically harmful ultraviolet radiation and its role in determining the temperature structure of the atmosphere. A systematic comparison of the SPARC Data Initiative ozone datasets has been provided in Tegtmeier et al. (2013) and SPARC (2017), revealing that the uncertainty in our knowledge of the $O_3$ mean state is smallest in the tropical MS and midlatitude LS and MS (see Table 8 for abbreviations). Notable differences between the datasets, on the other hand, exist in the tropical LS and at high latitudes. Here, the multi-instrument spread increases to $\pm30\%$ at the tropical tropopause (hence indicating considerable disagreement between the instruments) and $\pm15\%$ at polar latitudes (reasonably good agreement), which is partially related to inter-instrumental differences in vertical resolution and geographical sampling.

It should be noted that diurnal ozone variations are of $\sim10\%$ below 1 hPa and grow with increasing altitude up to more than 100% for upper mesospheric levels (e.g., Wang et al., 1996; Schneider et al., 2005). In addition, the impact of temperature uncertainties on the conversion from altitude to pressure during the gridded dataset production may cause additional errors that are particularly pronounced in the LM. Therefore, the mesospheric ozone observations were not corrected (as was done for the nitrogen-containing species, see Section 4.5). Instead, we present the ozone evaluations up to 1 hPa only.

An update of Figure 2 from Tegtmeier et al. (2013) is given in Figure 6 including new versions of SAGE II, SMR, OSIRIS, MIPAS, GOMOS, SCIAMACHY, ACE-FTS, ACE-MAESTRO, Aura-MLS, and HIRDLS ozone datasets. Note, MIPAS measured in a high-spectral resolution measurement mode between 2002 and 2004 (hereafter called MIPAS(1)), which switched to a low-spectral resolution measurement mode after 2004 (hereafter called MIPAS(2)). The latter led to the opportunity to

measure at a higher vertical resolution. In addition, new datasets obtained from OMPS-LP and SAGE III/ISS have been added. Tables 1 and 9 provide detailed information on time period, vertical range, vertical resolution and other information on the different data versions evaluated here. Overall the updated datasets agree better with notably smaller differences found for SMR, SCIAMACHY, ACE-FTS, GOMOS and MIPAS.

For SAGE II, the updated data version (v7.0) shows very similar structures in the relative differences to the MIM as version v6.2 used in Tegtmeier et al. (2013), albeit tending to more negative values throughout the atmosphere. Some of the rapid transitions between positive and negative values is a result of the combination of seasonal and diurnal sampling biases during the last few years of the mission (as evaluated here), when sampling became more sparse.

    For SMR, a new data product (v3.1) is evaluated here, based on the frequency mode 2 that monitors the band 544.102-

544.902 GHz. This product has been improved from earlier versions (not included in SPARC, 2017) by adjusting the line broadening constant and removing the pointing offset (Murtagh et al., 2018). Compared to the SMR frequency mode 1, version 2.1 ozone product (included in SPARC, 2017), the negative bias of 10-20% in the upper stratosphere has been reduced to values of 2.5-10% (Figure 6), thus showing very good to good agreement with the other instruments. The updated MIPAS(2) ozone (v224) benefits from better temperature data in the mesosphere and optimization of spectroscopic data for some spectral

regions. In comparison to the old MIPAS(2) ozone (v220), differences in the upper stratosphere are now reduced to 2.5-5%, which is about half of their original amount. SCIAMACHY provides an updated data version (V3-5) based on a new retrieval algorithm (Jia et al., 2015), which improved the retrievals considerably compared to the previously evaluated version (V2.5; Tegtmeier et al., 2013), with a positive bias in the MS and US now reduced from 10-20% to 2.5-10%.

    Updated ACE-FTS ozone (v3.6) in the MS and US shows considerably smaller differences to the MIM (mostly up to 5%,

Figure 6) than the old dataset (v2.2), which had a low bias in the MS of up to 10% and a high bias in the US of up to 10-20% (Tegtmeier et al., 2013). Interpolation of mixing ratios to the SPARC Data Initiative grid in log-pressure, data filtering based on quality flag information (Sheese et al., 2015) and reduced nonphysical oscillations in the updated pressure and temperature retrievals all contribute to the improved performance (Koo et al., 2017; Waymark et al., 2013). The ACE-MAESTRO dataset (v3.13), on the other hand, has larger biases than the previously evaluated version (v2.1; Tegtmeier et al., 2013). In particular,

the low bias in the LS and the high bias in the US increased from 2.5-5% to 10-20% (Figure 6, see also Bognar et al., 2019). Both MAESTRO versions use ACE-FTS temperature profiles in the retrieval, which requires information on the relative time difference between the measurements. For v3.13, this time difference is determined from MAESTRO $O_2$ slant column and ACE-FTS air mass slant column instead of using a constant value based on the best match between the ozone profiles. However, it is not clear if these changes cause the larger biases or if they are related to other issues of the v3.13 processing.

This is under investigation.

    The GOMOS $O_3$ dataset (v5.0) used previously has shown a substantial positive bias in the LS (30%) and UT (80%) (Tegt-meier et al., 2013) due to the high sensitivity of the retrieval algorithms to the aerosol extinction model. The new GOMOS datasets (ALGOM2s; Sofieva et al., 2017), whilst similar to additionally available v6.01 data (not used here) at higher altitudes, are based on a new $O_3$ profile inversion algorithm, which is optimized by enhancing the spectral inversion at visible

wavelengths for the UTLS, thus decreasing the impact of the aerosol model. As a result, GOMOS performs much better with

excellent agreement in the LS (Figure 6). In the UT, GOMOS retrieves lower ozone values than the other instruments, with differences to the MIM of 20 to 50%.

New $O_3$ data products from IUP-OMPS (Arosio et al., 2018) and USask-OMPS (based on a 2D retrieval) agree very well with the other datasets in the middle and upper stratosphere (Figure 6). The two data products are based on different retrieval algorithms, but show very similar structure with positive differences of 2.5-5% in the MS and US increasing up to 10-20% at the SH high latitudes and higher deviations of up to 50% in the tropical UTLS. The new $O_3$ data product from SAGE III/ISS (v5.1) agrees also well with the other datasets with a reasonably good agreement up to the US. For this work, the 'AO3' product was used because it has reduced noise compared to the 'MLR' product, particularly in the UT and US (see Wang et al. (2020) for details).

In summary, the updated $O_3$ datasets show improved agreement in most regions of the atmosphere (Figure 15). In particular, the 1-$\sigma$ multi-instrument spread in the UT decreased significantly at all latitudes from $\pm45\%$ on average to $\pm25\%$, among other things due to improved GOMOS performance. The region of very good agreement (1-$\sigma$ of $\pm5\%$), previously restricted to below 3 hPa, extends now further up into the US reaching the level of 1 hPa. In the LM, agreement also improves, with maximum deviations of $\pm30\%$ due to POAM III not being included in the updated evaluations. At polar latitudes, however, deviations are still large with maximum values of $\pm30\%$ found in the Antarctic LS, indicating considerable disagreement between the datasets.

## 4.2 Water Vapor ($H_2O$)

$H_2O$ is the single most important natural greenhouse gas and provides a positive feedback to climate change driven by anthropogenic emissions of carbon dioxide and other greenhouse gases. $H_2O$ is also a key constituent in atmospheric chemistry as source gas of the hydroxyl (OH) radical, which controls the lifetime of atmospheric pollutants, ozone, and greenhouse gases.

A comprehensive assessment of the SPARC Data Initiative $H_2O$ gridded datasets has been provided by Hegglin et al. (2013) and SPARC (2017). These evaluations revealed that the uncertainty in our knowledge of the $H_2O$ mean state is best in the LS and MS, with a relative uncertainty of only $\pm2$-6%. However, substantial biases were found between the datasets in the LM ($\pm15\%$), the polar regions ($\pm10$-15%), and the UTLS below 100 hPa ($\pm30$-50%), where sampling issues add uncertainty due to large gradients and high natural variability. However, once these biases are removed, the instruments showed very good agreement in the magnitude and structure of interannual variability.

Figure 7 shows an update of Figure 5 from Hegglin et al. (2013) including new data versions for ACE-FTS, Aura-MLS, MIPAS(1) and MIPAS(2), SAGE II and SCIAMACHY, and adding new datasets obtained from HIRDLS, ACE-MAESTRO, and SAGE III/ISS. Tables 1 and 10 provide detailed information on data versions, time period, vertical range, vertical resolution and other information on the different data versions evaluated here. LIMS and UARS-MLS (although having measured during an earlier period) are also added for comparison. All other datasets remain the same. Notable changes in the difference patterns arising from the updated data versions are identified in the following.

SAGE II (v7.0) shows large changes when compared to SAGE II (v6.2) used in Hegglin et al. (2013) and SPARC (2017), with positive differences replacing negative differences over large parts of the stratosphere. In the MS, the differences to the

MIM have decreased from between -5% and -10% (v6.2) to values mostly within $\pm2.5\%$ (v7.0) (Figure 7), now indicating excellent agreement with the other datasets. Much smaller differences to the MIM ($\pm5\%$) indicating very good agreement are also found in the UTLS, where large negative biases (>10-20%) existed in the previous version (v6.2) (Hegglin et al., 2013). This overall improvement is a consequence of modifying a spectral filter channel correction in the SAGE II retrieval (Thomason, 2004) using SAGE III/M3M as the basis for comparison in v7.0 instead of HALOE in v6.2 (Damadeo et al., 2013; see also Hegglin et al., 2014). In the US, on the other hand, differences from the MIM have increased from near zero to 5% and higher.

The new MIPAS(1) (V3o_H2O_21) and MIPAS(2) (V5r_H2O_224) data versions show generally very similar features in the differences to the MIM as the earlier data versions V3o_H2O_13 and V5r_H2O_220 used in Hegglin et al. (2013), respectively. MIPAS(2) exhibits some improvements in the tropical US, where differences to the MIM decreased from around 10% to 5% in the newer version. MIPAS(1) improved in the LM, where differences to the MIM decreased from >10% in V3o_H2O_13 evaluated in Hegglin et al. (2013) to smaller or even slightly negative values (between 2.5% to -5%). As a consequence, the new data versions of MIPAS(1) and MIPAS(2) seem more similar in character throughout the stratosphere and LM, except in the UTLS where MIPAS(2) generally shows positive differences to the MIM (>10%), while MIPAS(1) shows both positive (>5%) and negative differences to the MIM (>-5%) depending on the region. Note, it is expected that the application of averaging kernels would likely improve the comparison (which should be tested in future work).

The new ACE-FTS (v3.6) and Aura-MLS (v4.2) data versions show both slight improvements in the UTLS, and Aura-MLS also has slightly smaller positive differences to the MIM in the US. The negative bias seen in Aura MLS around 200 hPa in the evaluation of Hegglin et al. (2013), which extended the findings by Vömel et al. (2007) based on balloon soundings to all latitudes, is, however, still apparent. ACE-MAESTRO (v31), a new instrument in the comparison, shows rather large positive differences to the MIM (mostly >10-20%) across its measurement range in the UTLS (except between approximately 300 and 205 hPa in the extratropics, where negative of similar size are found). The wet bias in the tropical LS is a known issue for this version of ACE-MAESTRO (Lossow et al., 2019).

SCIAMACHY's negative bias to the MIM of around 10% found for data version v3.0 by Hegglin et al. (2013) in the NH LS slightly improved in the version evaluated here (v4.0) to 5%, and also the positive bias when compared to the MIM in the tropical UTLS (from 20% to 10%). HIRDLS (v7.0) exhibits a negative bias of >10% with respect to the MIM extending across the MS, and SMR (v2.0) an even larger negative bias of >20%.

While LIMS (v6.0) and UARS-MLS (v6) are not directly comparable to the other instruments due to the time period they measured in, the very different character in the differences still highlights that trends in $H_2O$ are of minor importance when compared to inter-instrument differences. UARS-MLS shows a very uniform negative bias with respect to the MIM of -10% whereas LIMS exhibits a positive bias in the extratropical LS and MS, and a more negative bias across the US and in the tropical LS. A part of the negative $H_2O$ bias for the US in LIMS may be due to the increases in $CH_4$ and its conversion to $H_2O$ during the intervening years.

The new instrument SAGE III/ISS (v5.1) shows excellent agreement with the MIM across the MS, US, and into the LM (with relative differences of $\pm\,2.5\%$ only), although the data can be less trusted at altitudes above 0.5 hPa, where strong positive

relative differences from the MIM ($> 20\%$) are found. This feature persists even when comparing datasets from instruments available during the same years (2017-2018) (including ACE-FTS and Aura MLS) (not shown) and is likely due to some reminiscent profiles that 'keel over' to very high values in the USLM, potentially biasing the mean field high. These profiles will be filtered out and/or corrected in future versions. In the UT and LS, SAGE III/ISS (v5.1) shows generally negative differences to the MIM, with the values improving from $<$-20% at 300 hPa to -5% around 30 hPa.

Overall, the update in the $H_2O$ datasets has only led to some small improvements and only in some regions of the atmosphere (see Figure 15). In the NH LS, the 1-$\sigma$ multi-instrument spread decreased from $\pm10\%$ to $\pm5\%$, and in the tropical UTLS from $\pm20\%$ to $\pm10\%$, among other reasons due to improved performance of SCIAMACHY and SAGE II. In the US, on the other hand, the multi-instrument spread increased slightly from $\pm10\%$ to $\pm12.5\%$, most likely due to the changes found in the new data version of SAGE II.

## 4.3 Methane ($CH_4$)

$CH_4$ is the most abundant hydrocarbon in the atmosphere and with a lifetime of around 8 years (Lelieveld et al., 1998) is considered long-lived. It is a very effective greenhouse gas and the second-largest contributor to anthropogenic radiative forcing since preindustrial times after $CO_2$. $CH_4$ is a source gas for stratospheric water vapour (resulting in a positive climate feedback), affects stratospheric ozone chemistry, and in the troposphere acts to reduce the atmosphere's oxidizing capacity.

The earliest $CH_4$ measurements from space were obtained from SAMS on Nimbus-7 between 1979 and 1981 (Taylor, 1987), followed by measurements from ATMOS since the mid-1980s (Gunson et al., 1996), and from ISAMS (Taylor et al., 1993) and CLAES (Roche et al., 1993) on UARS (along with HALOE). As mentioned above, these datasets were not considered in the SPARC Data Initiative. The first vertically resolved satellite datasets of $CH_4$ available to the SPARC Data Initiative were made by HALOE in 1991. MIPAS started measuring $CH_4$ in 2002 providing about 4 years of overlap (although with a major gap in 2004). From 2004 onwards there are also ACE-FTS measurements available for comparison. Tables 1 and 11 provide information on the availability of $CH_4$ measurements, including data version, time period, height range, vertical resolution, and references relevant for the data product.

Figure 8 shows meridional profiles of $CH_4$ at different pressure levels for August averaged over 1998-2008. These comparisons provide information on the latitudinal distribution of $CH_4$ and latitude-height dependency of the differences between the instruments. At 50 hPa, the instruments tend to agree very well with each other mostly within $\pm5\%$. The same is largely true for the 10 hPa level. In both cases, ACE-FTS (v3.6) and HALOE (v19) agree best with each other, while MIPAS(2) (v224) seems to have somewhat higher discrepancies from the MIM than the other instruments and also exhibits differences that vary more with latitude. At 5 hPa, however, the differences of the instruments with respect to the MIM increase to $\pm20\%$. Here, HALOE is closest to the MIM, ACE-FTS shows largest negative values and MIPAS(1) (v21) largest positive values. The deterioration in the agreement between the instruments with height is qualitatively consistent with the results of SPARC (2017). However, the new data versions used here agree quantitatively much better with each other, particularly at the 50 hPa and 10 hPa levels.

We now turn to an example which can be used to test the physical consistency of the available datasets. To this end, the latitude-time evolution of $CH_4$ for the different instruments at 2 hPa is shown in Figure 9. ACE-FTS and HALOE fields

have been constructed using linear interpolation to fill in data gaps that arise from their sparse latitude-time sampling patterns. Figure 9 reveals local maxima located in the tropics just off the equator in the respective summer hemisphere, distinct features that were found in earlier studies (e.g., Jones and Pyle, 1984; Ruth et al., 1997) and attributed to the equatorial semiannual oscillation (Choi and Holton, 1991). The maxima in the trace gas thereby coincide with maxima in upwelling, which brings

younger air (less depleted in $CH_4$) to higher altitudes. Tropical $CH_4$ thus should show a semi-annual cycle. Photochemistry, on the other hand, causes minima at high latitudes during summer and autumn, with $CH_4$ lifetimes decreasing to 4 months at these altitudes (Solomon et al., 1986; Randel et al., 1998).

HALOE captures the tropical semiannual oscillation well and also indicates the high-latitude minima during the summer months. MIPAS shows very similar features, but due to its better spatio-temporal sampling extends further into the polar

regions, revealing the full extent and timing of these features. The tropical maxima in both MIPAS(1) and MIPAS(2) are stronger than in HALOE. ACE-FTS exhibits a much noisier field due to its limited sampling and hence exhibits sharp maxima and edges especially in the tropics, where the instrument scans through only once a season. While datasets in equivalent latitude would help to reduce the noise, this quantity was not available to the SPARC Data Initiative. Knowledge of the representativeness of ACE-FTS in geographical latitude is however still valuable for model-measurement comparisons.

The difference plots indicate a low bias in HALOE and ACE-FTS versus a known high bias in MIPAS(1). MIPAS(2), despite exhibiting a somewhat patchier difference field, provides supporting evidence for a high bias in MIPAS(1) at this pressure level. Compared to the data versions used in SPARC (2017), the new data versions used here agree generally better even at this level. While HALOE's difference field to the MIM remains the same (no new data version available), ACE-FTS has somewhat less noise, now tending to more negative values, MIPAS(1) shows smaller differences especially in the tropical and mid-latitude

regions, and MIPAS(2) shows slightly increased differences across the time-latitude domain. It is important to note that $CH_4$ showed only small trends in the troposphere over the time period 1998-2008, thus a trend in this trace gas is not expected to contribute significantly to the inter-instrument differences. An evaluation limited to the year 2005 (during which all instruments were reporting data) mostly confirms the results described here (not shown).

The overall impact of updated data versions on our knowledge of the mean state of the atmosphere in terms of $CH_4$ is shown

in Figure 15. Compared to SPARC (2017), the new data versions have led to decreases in the 1-$\sigma$ multi-instrument spread across the UTLS and MS from $\pm10\%$ to $\pm5\%$. A decrease of around 5% in the 1-$\sigma$ multi-instrument spread is also found across the USLM in comparison with SPARC (2017), although the values are much more variable in this region.

### 4.4  Carbon monoxide (CO) comparisons

Carbon monoxide (CO) has a lifetime of approximately three months in the UT and LS. In the troposphere, CO impacts air

quality and has an indirect radiative forcing effect, since it scavenges OH that would otherwise react with (and deplete) the greenhouse gases methane and ozone (Daniel and Solomon, 1998). Due to its intermediate lifetime, it is often used as tracer to identify troposphere-stratosphere exchange (e.g., Hoor et al., 2004; Hegglin et al., 2009). In the lower stratosphere, CO reaches a background value ranging between 8 and 15 ppbv (Flocke et al., 1999) as determined by the equilibrium between its production (from methane oxidation) and loss (from CO oxidation).

Only a few limb sounders provide CO measurements, with the zonal monthly mean fields from SMR, MIPAS, ACE-FTS, and Aura-MLS contributing to the SPARC Data Initiative. The earliest dataset that would offer CO, but which is not included in the comparisons here, can be obtained from SAMS on Nimbus 7 (although with a very high noise level; Taylor, 1987). Other

useful CO measurements were obtained by ATMOS on the Space Shuttle (Gunson et al., 1996), and from ISAMS on UARS (Taylor et al., 1993). Tables 1 and 12 compile information on the availability of CO measurements, including time period, height range, vertical resolution, and references relevant for the data product used in this report.

For CO, we focus first on the zonal annual mean evaluation as shown in Figure 10, which is one of the standard evaluations in the SPARC Data Initiative (as also shown in Figures 6 and 7). ACE-FTS and Aura-MLS are averaged over the period 2004-

2009, while MIPAS(2) is averaged over 2005-2009, MIPAS(1) over 2002-2004, and SMR over 2003-2004. The figure reveals large differences in the structure and values of CO as measured by the different instruments. Nevertheless, common features are minimum values around 15 ppbv in the LS and MS, and strongly increasing values towards the USLM with maxima in the polar regions. These large values stem from the photodissociation of $CO_2$ in the mesosphere and subsequent downward transport (Solomon et al., 1985). Increasing values can also be seen when moving towards the UT (with tropospheric CO coming mostly

from anthropogenic sources). The mid-infrared sensors MIPAS(1) (v20), MIPAS(2) (v222), and ACE-FTS (v3.6) agree best. SMR (v2.1) CO exhibits a fair amount of noise, which stems from the fact that the CO was retrieved about 2 days per month and during a limited time period from October 2003 to October 2004 only. SMR does not reproduce the low background values of 8-15 ppbv expected in the LS to MS. Aura-MLS (v4.2), on the other hand, shows stratospheric CO values of smaller than 10 ppbv that are somewhat lower than those observed by MIPAS and ACE-FTS (see also Pumphrey et al., 2007). Aura-MLS

also shows a local minimum in CO in the tropical LM (around 0.2 hPa), which is not seen in other datasets. Aura-MLS and also SMR do not reproduce the same downward and poleward sloping trace gas isopleths in the LS as seen by MIPAS and ACE-FTS, a typical feature observed for long-lived trace gases as a result of transport and mixing within the Brewer-Dobson circulation (Tung, 1982).

In comparison with the CO evaluations in SPARC (2017), significant improvements are found for the new data versions of

MIPAS(1) and ACE-FTS. For these instruments, the relative biases with respect to the MIM in the tropical MS have decreased from 10-20% to ±5%. While the shortcomings in Aura-MLS were already pointed out in SPARC (2017), the relative biases with respect to the MIM in the LM (around ±10%) are now much closer to ACE-FTS and MIPAS(2). Positive biases of more than 50% in Aura-MLS (v3) in the UTLS have also decreased to 20%, although the isopleths are still relatively flat compared to those found by the other instruments.

In addition, Figure 11 shows deseasonalized anomalies for CO at three different pressure levels in either the tropics or extra-tropics. The instruments all capture the interannual variability well. Despite its limited tropical sampling, ACE-FTS seems to capture the interannual variability in the tropical UT at 200 hPa well, and notably better than data version 2.6 used for SPARC (2017). It is also noteworthy that the shortcomings of Aura-MLS in reproducing the zonal annual mean are not hampering the ability of the retrieval to observe the correct interannual variability in these time series, hence still pointing out the usefulness of the Aura-MLS product for such evaluations.

Overall, our knowledge of the mean state for CO as expressed by the 1-$\sigma$ multi-instrument spread (see Figure 15) has improved across the USLM by about 5%. In the UTLS and MS, however, the 1-$\sigma$ spread remains similar, at above $\pm 30\%$. At least in the LS, this is largely due to the persisting problems in the CO distribution obtained from Aura-MLS. Note that SMR and MIPAS(1) were not included in Figure 15, so to remain comparable with the summary evaluation of CO in SPARC Data Initiative report (SPARC, 2017).

## 4.5 Nitrogen species (NO, $NO_2$, $NO_x$, $HNO_3$ and $NO_y$) comparisons

Total reactive nitrogen ($NO_y$) is the sum of all atmospheric reactive nitrogen species ($NO_y = NO + NO_2 + NO_3 + HONO + HNO_3 + HNO_4 + peroxyacetylnitrate (PAN) + RONO_2 + ClONO_2 + 2 \times N_2O_5 + BrONO_2 + organic nitrate + particulate nitrate$) (NRC, 1984 p.34), with largest contributions from nitric oxide (NO), nitrogen dioxide ($NO_2$) and nitric acid ($HNO_3$). While $HNO_3$ and $NO_x$ constitute 80-100% of all possible species of $NO_y$ in the LS, PAN can constitute as much as 20-50% in the tropical UT and extratropical UTLS (i.e., altitudes below 200 hPa) (Kendo et al., 1997; Fadnavis et al., 2014). Tropospheric $NO_y$ originates mostly from sources of NO and $NO_2$ (together known as the nitrogen oxide family, $NO_x$) released from fossil fuel burning, lightning, chemical processes in soils, and biomass burning. In the stratosphere, $NO_y$ is primarily produced from the oxidation of $N_2O$, which originates from soil and ocean emissions, biomass and fossil fuel burning, livestock manure and fertilization in agriculture. Another important source is the enhancement of upper atmospheric $NO_x$ through ionizing energetic particle precipitation (Solomon et al., 1982) and the $NO_x$ downward transport inside the polar vortex (Funke et al., 2005a). Reactive nitrogen species play an important role in stratospheric ozone chemistry through different mechanisms including the catalytic $NO_x$ cycle (Crutzen, 1970), the role of $HNO_3$ in polar stratospheric cloud formation (Fahey et al., 2001) and $NO_2$-driven conversion of halogens into reservoir substances. Stratospheric nitrogen will remain a future research focus as unregulated $N_2O$ emissions are expected to become the most important ozone-depleting emission during the 21st century (Ravishankara et al., 2009).

Sunlight-driven conversion between stratospheric NO and $NO_2$ causes a strong diurnal cycle in both species with large NO abundances during daytime, large $NO_2$ abundances during nighttime and steep gradients at sunrise and sunset in both species. A direct comparison of satellite-based NO and $NO_2$ measurements (which correspond to different local solar times, LST) is not possible, unless the dependence on the solar zenith angle (SZA) is taken into account. Solar occultation measurements made at SZA = 90° (NO from HALOE and ACE-FTS, $NO_2$ from SAGE II, HALOE, POAM II, POAM III, SAGE III/M3M and ACE-FTS) can be compared amongst each other if separated into local sunrise and sunset. Limb scattering and emission measurements (NO from MIPAS and SMR, $NO_2$ from LIMS, OSIRIS, SCIAMACHY, MIPAS, and HIRDLS) and stellar occultation measurements ($NO_2$ from GOMOS) correspond to different SZAs and need to be scaled to a common LST. We follow the approach to scale the NO measurements from ACE-FTS and SMR as well as the $NO_2$ measurements from OSIRIS, SCIAMACHY, and ACE-FTS with a chemical box model (McLinden et al., 2010) to the LST of the MIPAS measurements 10am/pm. $NO_2$ from HIRDLS (June 2005 to May 2006) has been scaled to 10am/pm with the SD-WACCM Version 3 (Garcia et al., 2007). Tables 13-15 summarise information on the availability of NO, $NO_2$, and $HNO_3$ measurements, including data version, time period, height range, vertical resolution, and references relevant for the data product used in this study. For these

species, updated data versions are available from ACE-FTS (v3.6), GOMOS (v6.01), HIRDLS (v7.0), MIPAS(1) (v20), SAGE II (v7.0), and SCIAMACHY (v4-0).

$NO_x$ shows only a weak diurnal cycle in the LS to MS and is available from HALOE, ACE-FTS and MIPAS based on the sum of NO and $NO_2$. OSIRIS and SCIAMACHY measure $NO_2$ but not NO and their $NO_x$ datasets are compiled with the help of a chemical box model (McLinden et al., 2010). In the following evaluations the $NO_x$ datasets from ACE-FTS, HIRDLS, OSIRIS, and SCIAMACHY are scaled to 10am and 10pm, respectively.

The nitrogen species $HNO_3$ (from LIMS, UARS-MLS, SMR, MIPAS, ACE-FTS, Aura-MLS, and HIRDLS) and also the reactive nitrogen family $NO_y$ (from ACE-FTS, MIPAS, and a combination of the Odin measurements of OSIRIS and SMR) are long lived, except for some diurnal variations of $HNO_3$ in the LM. Note, not all reactive nitrogen species that make up $NO_y$ are measured by the stratospheric limb sounders presented here. The $NO_y$ datasets from ACE-FTS (based on the methodology of Jones et al. (2011), except for the vertical binning) and MIPAS (Funke et al., 2014) are compiled from NO, $NO_2$, $HNO_3$, $HNO_4$, 2 x $N_2O_5$ and $ClONO_2$ (six-species datasets), all directly measured by the instruments. The $NO_y$ Odin dataset (Brohede et al., 2008) is based on $NO_2$ from OSIRIS, $HNO_3$ from SMR and NO, 2 x $N_2O_5$ and $ClONO_2$ taken from scan-based chemical box model simulations (McLinden et al., 2010), while $HNO_4$ is not included (five-species dataset). Note, the ACE-FTS and Odin $NO_y$ products are daytime datasets and do not include polar night data as opposed to MIPAS. In all figures the instrument names will be completed by lower indices giving the number of species used to compile the dataset, e.g., Odin5 for the Odin five-species dataset. It should also be noted that although available from both MIPAS and ACE-FTS, none of the $NO_y$ datasets presented here includes PAN, which can be a significant contribution to $NO_y$ at the lower end of the altitude range shown.

We present here the evaluation of the seasonal cycle of the nitrogen species NO, $NO_2$, $NO_x$, $HNO_3$ and $NO_y$ in the mid-latitudes (30°S-60°S and 30°N-60°N) and tropics (10 hPa 20°S-20°N) at 10 hPa (Figure 12). The latitude bands and pressure level have been chosen to include as many species and instruments as possible. While the NO maximum can be found around 1 hPa, the $HNO_3$ maximum is situated much lower in the atmosphere at around 30 hPa. The choice of evaluations at the 10 hPa level in the MS thereby ensures that both species are abundant. For NO, ACE-FTS shows good agreement with MIPAS except for NH mid-latitudes during boreal winter when ACE-FTS can be up to 25% lower. In particular in the SH midlatitudes, the new data version of ACE-FTS (v3.6) has led to clear improvements and the consistently too low NO values (ACE-FTS v2.2) are now much closer to MIPAS. Scaled SMR data agrees well with the other two datasets in the US to LM, but shows large deviations in the MS and is thus omitted from the comparison in Figure 12.

The $NO_2$ comparison (Figure 12, 2nd row) in the mid-latitudes shows a very good agreement of all datasets except for ACE-FTS and HIRDLS during boreal winter. The seasonal cycle of $NO_2$ from ACE-FTS and HIRDLS in the NH, and to some degree also in the SH, has a larger amplitude than the one derived from the other three instruments. In the tropics, all instruments agree on a very weak seasonal signal except for HIRDLS, which displays an annual cycle with an amplitude of 50%. Over the whole measurement range (LS to US), the datasets from MIPAS, OSIRIS and SCIAMACHY agree better with each other than with ACE-FTS or HIRDLS. Compared to the old data versions (SPARC, 2017), largest improvement is

found for the updated ACE-FTS (v3.6) in the SH mid-latitudes, where the negative bias has been removed, consistent with NO evaluations.

The $NO_x$ seasonal cycles of all datasets agree well on the phase, but show some deviations in the amplitude of the signal (Figure 12, 3rd row). For the mid-latitudes, absolute values of ACE-FTS $NO_x$ are considerably lower than the other instruments during the respective winter season consistent with the findings of the NO and $NO_2$ evaluations. The latter characteristic also causes a larger amplitude of the ACE-FTS seasonal cycle in the both mid-latitude bands. In the tropics, datasets agree well with a relatively weak seasonal cycle that is most pronounced in MIPAS. Again, largest improvement is found for ACE-FTS (v3.6) in the SH mid-latitudes and NH mid-latitudes during winter.

The comparison of the $HNO_3$ seasonal cycle (Figure 12, 4th row) includes in addition to ACE-FTS, HIRDLS and MIPAS, also the SMR and Aura-MLS datasets. All datasets can be evaluated without chemical scaling and show mostly a very good agreement of the mean values except for higher ACE-FTS values in the SH mid-latitudes during austral winter. The updated HIRDLS dataset (v7.0) shows an improved performance compared to the old data version (v6.0, SPARC, 2017), since the too low $HNO_3$ values during boreal autumn and the resulting semiannual signal are now removed. For all regions above 30 hPa, Aura-MLS and HIRDLS are on the low side while ACE-FTS, MIPAS and SMR are on the high side. Below 30 hPa, the situation is reversed.

Finally, evaluations of the $NO_y$ seasonal cycle (Figure 12, 5th row) show some severe differences (although not necessarily in the mean value, just the amplitude), most notably in the SH mid-latitudes where the seasonal cycle from Odin is completely opposite to the one from ACE-FTS and MIPAS. These deviations can be understood from the OSIRIS $NO_2$ and $NO_x$ as well as the SMR $HNO_3$ seasonal cycles in the SH, which show a smaller amplitude than the respective MIPAS and ACE-FTS datasets. In general, we expect increasing $NO_y$ values during the dynamically quiescent spring and summer time, and this is observed by ACE-FTS and MIPAS. In the NH, the $NO_y$ maximum is observed in boreal autumn by all three instruments. In the SH spring, Odin shows a secondary maximum that is less pronounced than in the NH, but this provides for a better agreement with the other two datasets. For ACE-FTS, the too low $NO_x$ values in the SH and NH boreal winter cancel out with the too high $HNO_3$ values, resulting in an overall good $NO_y$ agreement with MIPAS. The overall annual mean state of $NO_y$ is well known and the three datasets show excellent agreement (Figure 16) with differences smaller than $\pm 5\%$. However, deviations can be larger for individual months (up to $\pm 10\%$, Figure 12) and cancel out in the annual mean.

Apart from the climatological and seasonal differences between the datasets, it is of interest to evaluate how well the instruments detect signals of interannual variability. Figure 13 shows the time series of $NO_2$ mean values (upper panels) and deseasonalized anomalies (lower panels) for the tropical latitude band 20°S-20°N at 10 hPa. We focus on the evaluation of the $NO_2$ interannual anomalies of the longer time series SAGE II and HALOE in comparison with interannual variability of ACE-FTS, MIPAS, OSIRIS, SCIAMACHY, GOMOS and HIRDLS. Anomalies calculated in an additive sense by subtracting monthly multi-year mean values for each month might also display a diurnal cycle and are therefore not suitable evaluation tools for unscaled datasets. However, anomalies calculated in a multiplicative sense as percentage deviations from the monthly multi-year mean values are less affected by the diurnal variations. Since no scaled versions of SAGE II and HALOE data

are available the comparison focuses on multiplicative anomalies of the sunset/nighttime $NO_2$ datasets including SAGE II, HALOE, ACE-FTS local sunset datasets and MIPAS, OSIRIS, SCIAMACHY, GOMOS 10 pm and HIRDLS night datasets.

The comparison of the mean values (upper panel) shows a very good agreement of MIPAS, GOMOS and scaled SCIA-MACHY measurements. Scaled OSIRIS data are somewhat lower than the other three datasets. Diurnal $NO_2$ variations be-
tween 10pm and local sunset at the 10 hPa level are so small, that SAGE II, HALOE and ACE-FTS data taken at local sunset mostly agree with the other datasets for the overlap period 2003-2005. From 2003 onwards the multiplicative anomalies of all datasets display the expected QBO signal with the best agreement between MIPAS, OSIRIS, GOMOS and SCIAMACHY. The three years of HIRDLS measurements display a larger amplitude of the QBO signal and also larger month-to-month fluctuations, possibly due its higher vertical resolution (which should be tested in future work). Interannual anomalies from ACE-FTS
agree for some months with the other datasets, but show large deviations for other months. Due to the sparse sampling it is not possible to diagnose a QBO signal in the ACE-FTS time series. Local sunset evaluations from SAGE II and HALOE show also large month-to-month variations but agree reasonably well on their interannual variability and display the QBO signal over the whole time period. The same is not true, however, for the local sunrise evaluations of the two instruments where HALOE shows only a weak and SAGE II no clear indication of a QBO signal (SPARC, 2017).
The overall knowledge on the atmospheric mean state of the different trace gases treated in this section as expressed by the $1$-$\sigma$ multi-instrument spread is shown in Figure 16. In comparison to earlier evaluations (SPARC, 2017), the updated nitrogen data sets show a slightly improved agreement. In particular the scaled ACE-FTS data sets agree better with the other time series in terms of absolute bias and seasonal cycle.

## 4.6  Hydroperoxyl ($HO_2$) comparisons

Hydroperoxyl ($HO_2$) together with the hydrogen atom (H) and hydroxyl (OH) form the $HO_x$ -family. $HO_2$ is formed in the reaction between a hydrogen atom (H) and molecular oxygen ($O_2$), or between ozone ($O_3$) and OH. OH affects stratospheric ozone chemistry through its role in the $HO_x$ catalytic reaction cycle that destroys ozone. The $HO_x$ cycle was the first catalytic reaction cycle to be identified (Bates and Nicolet, 1950). $HO_x$ chemistry dominates ozone destruction above 40 km, while $NO_x$ dominates ozone destruction in the MS (Salawitch et al., 2005). In the troposphere, $HO_2$ is generated as an intermediate
product of the oxidation of many hydrocarbons.

Measurements of $HO_2$ are available from instruments that measure in the sub-mm/microwave wavelength bands, namely SMILES, SMR, and Aura-MLS. Other available $HO_2$ datasets are restricted to balloon campaigns, such as from the Far Infrared Spectrometer (FIRS-2) (Johnson et al., 1995; Jucks et al., 1998). There is no temporal overlap between the three satellite instruments, since SMR currently only provides $HO_2$ data as research product during one year (October 2003-2004).
SMILES on the other hand operated between October 2009 to April 2010 only. While SMILES measures the full diurnal cycle, Aura-MLS measures at 1:30am/pm, and SMR at 6:30am/pm. Since $HO_2$ does not exhibit very strong variations during the day, daytime datasets are compared only. Tables 2 and 16 compile information on the availability of $HO_x$ measurements, including data version, time period, height range, vertical resolution, and references relevant for the data product used in this study.

Figure 14 shows the zonal monthly mean evaluation between Aura-MLS and SMILES for November 2009 and February 2010. SMR is not shown due to a very limited temporal and spatial coverage (see Figure 4.23.2 in SPARC, 2017). As seen in the MIM, mixing ratios are similar in both months in the tropics (where SZAs do not vary much with season), indicating only a weak seasonal cycle in the daytime zonal monthly mean field. Lowest mixing ratios are found in the polar region of the winter hemisphere (during high SZA conditions), indicating a somewhat more pronounced seasonal cycle in these regions of the atmosphere. The differences to the MIM indicate very good (up to $\pm5\%$) to excellent (up to $\pm2.5\%$) agreement between SMILES and Aura-MLS, except in the lower part of the measurement range (around 20 hPa) where differences to the MIM increase to $\pm10\%$ and more. The results presented here are comparable to (if not somewhat better than) what was found in SPARC (2017), where multi-year monthly mean $HO_2$ fields were used for the comparison.

## 5  Summary Evaluations

The SPARC Data Initiative provides an estimate of the systematic uncertainty in our knowledge of the measured fields' mean state derived from the inter-instrument spread defined as $\pm1\sigma$. Figure 15 shows these fields for the long-lived trace gases. Note, we adopt the same vocabulary (see Table 7) for the summary comparisons (based on relative standard deviations) as used earlier for instrument specific evaluations (based on relative differences). For $CH_4$, the uncertainty is smallest in the tropical and mid-latitude MS and LS and larger towards the UTLS, the US and LM. The same has been found for other long-lived trace gases such as $O_3$, $H_2O$, $N_2O$, and HF. In contrast, the trace gases CFC-11 (or $CCl_3F$), CFC-12 (or $CCl_2F_2$), and $SF_6$ show the best agreement in the UTLS and larger deviations in the MS. Nearly all trace gases show larger deviations in the polar regions than at lower latitudes, colorbluewhich is at least partially due to increased sampling biases found at higher latitudes. Datasets of CO, which is a trace gas with an intermediate lifetime, are characterized by large relative differences throughout most of the measurement range. The large CO differences in the annual zonal mean structure ($\pm30\%$ in the LS) should be further addressed in forthcoming retrieval revisions. Overall, the $\pm1\sigma$ multi-instrument spread has decreased for all long-lived trace gas species by up to 10% since SPARC (2017), except possibly for CO, indicating a more consolidated knowledge of the state of the atmosphere resulting from improvements in the retrievals of these species.

The agreement of the nitrogen species NO, $NO_2$, and $HNO_3$, as derived from the relative deviations between the datasets, depends strongly on the atmospheric distribution of the respective gas with larger relative differences in regions of smaller mixing ratios (Figure 16). While NO and $NO_x$ agree very well in the tropical and subtropical MS and US, $NO_2$ and $HNO_3$ have larger deviations in the US and show the best agreement in the tropical and mid-latitude MS and for $HNO_3$ also in the LS. All datasets (except for $HNO_3$ and $NO_y$ in the Northern Hemisphere) have considerably larger deviations in the polar regions, at least in part again because of sampling issues and the large atmospheric variability that is less well sampled by the measurements going into the monthly mean datasets (cf. Toohey et al, 2013). Finally, the $NO_y$ datasets show excellent agreement throughout most of the measurement range except for the polar latitude LM. Overall, the $\pm1\sigma$ multi-instrument spread in the nitrogen species has decreased only slightly (by 5%) when compared to SPARC (2017).

The agreement between datasets of chlorine compounds (Figure 17) and shorter-lived species depends strongly on the lifetime of the trace gas considered. HCl, which is longer-lived, exhibits very good agreement and the day-time datasets of the shorter-lived ClO show good to reasonable agreement in the MS and US where mixing ratios are highest. HOCl, which is short-lived, shows mostly reasonable agreement in the US during night-time. $HO_2$ is available from a small number of instruments only and is thus not included in the synopsis plots, although the $HO_2$ comparisons show promising results with mostly good agreement throughout the MS, US, and LM. The large deviations between the datasets of shorter-lived species stem partially from the difficulty of accounting for the strong diurnal cycles these trace gases exhibit. Scaling of the data to a common day- or nighttime using a chemical box model helped improve the comparisons in some cases. However, it remains a challenge to estimate how much these deviations are related to errors introduced by the scaling procedures and how much of the deviations correspond to direct measurement differences. Overall, the $\pm 1\sigma$ multi-instrument spread in the chlorine-containing species has improved for HCl, but has remained very similar for ClO and HOCl when compared to SPARC (2017).

## 6    Conclusions

This paper presents an overview and update of the evaluations performed within the WCRP SPARC Data Initiative as published in the SPARC Data Initiative Report (SPARC, 2017). To date, the SPARC Data Initiative represents the most comprehensive assessment of stratospheric composition measurements obtained from an international suite of limb sounders from various space agencies and other national institutions. The SPARC Data Initiative thereby offers the first systematic assessment of the availability of chemical trace gas and aerosol observations from satellite limb sounders; provides these observations in a common and easy-to-handle data format (zonal monthly means); and presents a detailed comparison between these datasets, importantly covering different generations of satellite limb instruments and contrasting the products of different agencies around the world. Here we extended the SPARC (2017) evaluations, which covered the period 1978-2010, out to the end of 2018, and used the most recent data versions that have become available in the meantime. New observations from OMPS-LP (on SUOMI NPP) and SAGE III/ISS are also added to the original list presented in SPARC (2017), which included LIMS, SAGE I/II, SAGE III/M3M, HALOE, UARS-MLS, POAM II/III, OSIRIS, SMR, MIPAS, GOMOS, SCIAMACHY, ACE-FTS, ACE-MAESTRO, Aura-MLS, HIRDLS, and SMILES. colorblue(Note, aerosol evaluations and zonal monthly mean timeseries data will be presented in a follow-on study).

The SPARC Data Initiative comparisons are based on vertically-resolved zonal monthly mean datasets of 26 different atmospheric constituents, including the stratospheric trace gases of primary interest, $O_3$ and $H_2O$, major long-lived trace gases ($SF_6$, $N_2O$, HF, $CCl_3F$, $CCl_2F_2$, $NO_y$), trace gases with intermediate lifetimes (HCl, $CH_4$, CO, $HNO_3$), and shorter-lived trace gases important to stratospheric chemistry including nitrogen-containing species (NO, $NO_2$, $NO_x$, $N_2O_5$, $HNO_4$), halogens (BrO, ClO, $ClONO_2$, HOCl), and other minor species (OH, $HO_2$, $CH_2O$, $CH_3CN$), and aerosol. The observations considered have been compiled on a common latitude-pressure grid, covering the region from the upper troposphere to the lower mesosphere (300-0.1 hPa) with a latitudinal resolution of $5°$. The zonal monthly mean time series are available from the Zenodo data archive (doi:10.5281/zenodo.4265393). A consistent file format was designed and is being used across the

different composition measurements and instruments, so as to allow for easy handling by the user (see Popp et al. (2020) for a discussion of the importance of a consistent data format in the provision of observational datasets).

The trace gas time series have then been evaluated by a common approach, comparing multi-year annual or monthly mean fields, allowing for maximum overlap between different instruments. By evaluating zonal monthly mean averages, the SPARC Data Initiative has taken a 'climatological' approach to data validation (Hegglin et al., 2008; Hegglin et al., 2013; Tegtmeier et al., 2013; SPARC, 2017) in contrast to the more common approach of using coincident profile measurements. The clima-
tological comparison method averages over multiple measurements, thereby reducing both instrument noise and geophysical variability from single profile comparisons, and offering a top-down instead of a bottom-up assessment of the (systematic) biases between different measurements. The climatological validation method has therewith the advantage that it is consistent for all instrument comparisons, avoids sensitivity to chosen limits defining coincident measurements, and produces larger sample sizes, which should in theory minimise the random part of the measurement error. This climatological approach, however, has
the disadvantage that climatological means can be biased due to non-uniformity of sampling or potential long-term trends in the trace gases. The extent to which the monthly and annual zonal mean datasets are representative of the true mean has been evaluated as part of the SPARC Data Initiative for two trace gases $O_3$ and $H_2O$ in a separate paper by Toohey et al. (2013). This study yields information on the potential sampling bias in the zonal monthly mean fields of these tracers and instruments, and is providing an approximate measure of the sampling bias also for trace gases with similar lifetimes to users who examine
variability and trends, or perform comparisons with free-running models.

The findings of the trace gas datasets comparisons presented here are generally consistent with the results of previous validation efforts based on the classical validation approach using profile coincidences (where available). Instruments with sparser sampling show noisier zonal means. Profiles with wide averaging kernels do not resolve sharp structures such as those found across the tropopause region. However, the climatological approach yields generally more comprehensive information on mea-
surement uncertainty in terms of latitude-pressure range covered. The comparisons of the datasets have in many cases improved our knowledge of the systematic biases between the available data products. Although not shown here, the comparison results generally do not change substantially when changing the number of years going into a averaged field or, in case of the longer-lived species, when calculating instrument differences for a month instead of a year. From this, it follows that the comparisons shown yield relatively robust conclusions on instrument/retrieval performance (see SPARC (2017) for detailed examples).
The conclusions from the SPARC Data Initiative highlight the use (or necessity) of observations from multiple instruments in order to characterize retrieval behaviour and overall observation quality as a function of latitude and pressure (or altitude). The small number of stratospheric limb sounders currently remaining in space (with most of them being long past their expected lifetime) and the even smaller number of planned future missions will likely have serious implications. These may impact not only our ability to perform a robust assessment of the quality of stratospheric composition measurements, but more importantly
to derive stratospheric composition changes from these measurements, which are needed to better understand the state of the ozone layer that protects life on Earth and its response to (as well as feedbacks on) climate change. As such, the gridded trace gas datasets from the SPARC Data Initiative may serve as an atlas and reference of stratospheric composition mean state and variability during the 'golden age' of limb satellite sounding of the atmosphere well into the future.

## 7  Data availability

All SPARC Data Initiative zonal monthly mean datasets can be found in the Zenodo data archive (Hegglin et al., 2020, doi:10.5281/zenodo.4265393).

*Author contributions.*  MIH and ST have designed and co-led the SPARC Data Initiative, performed all the evaluations, and written the text. The instrument PIs and their research staff have compiled the SPARC Data Initiative datasets to their best current knowledge and contributed to the writing and interpretation of the evaluation results.

*Competing interests.*  The authors declare no competing interests.

*Acknowledgements.*  While the SPARC Data Initiative has been driven from a user perspective, the measurement partners have been critical
to its success. These partners to whom the SPARC Data Initiative extends its thanks include the relevant instrument teams, the various space agencies (CSA, ESA, NASA, JAXA, SNSA, and other national agencies), and organizations such as CEOS-ACC and IGACO. We thank the World's Climate Research Programme (WCRP) for travel funding through the SPARC office to support our activity. The SPARC Data Initiative also thanks the International Space Science Institute in Bern (ISSI) who supported the activity through their ISSI International Team activity program and facilitated two successful team meetings in Bern. The work of MIH within the SPARC Data Initiative was
funded by the CSA SSEP (9SCIGRA-29), the ESA STSE-SPIN (4000105291/12/I-NB), and more recently for water vapour evaluations the ESA (Contract No. 4000123554) via the Water_Vapour_cci project of ESA's Climate Change Initiative (CCI). The work from ST was funded from the WGL project TransBrom and the EU project SHIVA (FP7-ENV-2007-1-226224). Work at the Jet Propulsion Laboratory, California Institute of Technology, was funded by the National Aeronautics and Space Administration (NASA). The Atmospheric Chemistry Experiment is a Canadian-led mission mainly supported by the CSA. Development of the ACE-FTS gridded datasets was supported by grants
from the Canadian Foundation for Climate and Atmospheric Sciences and the CSA. MIPAS data analysis and validation was supported by the German Federal Ministry for Economic Affairs and Energy under project number 50EE1547 and by the ESA CCI-O3 project. BF acknowledges support by the Spanish MCINN (ESP2017-87143-R) and EC FEDER funds. Development of SCIAMACHY and IUP-OMPS gridded datasets at the University of Bremen was funded in parts by the German Research Foundation (DFG) Research Units SHARP (FOR1095) and VolImpact (FOR2820), the German Aerospace Agency (DLR) SADOS project, ESA SQWG and Ozone CCI projects,
EU/ECMWF C3S project, and the University and State of Bremen. AR and CA also acknowledge the German HLRN (High-Performance Computer Center North) and the thread-safe FORTRAN library GALAHAD. CA acknowledges the support by the PRIME program of the German Academic Exchange Service (DAAD) and ESA's Living Planet Fellowship SOLVE. Work on HIRDLS was supported in the US by the National Aeronautics and Space Administration (NASA) and in the UK by the National Environmental Research Council (NERC). Development of Odin/SMR gridded datasets was supported by the Swedish National Space Agency (SNSA).

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

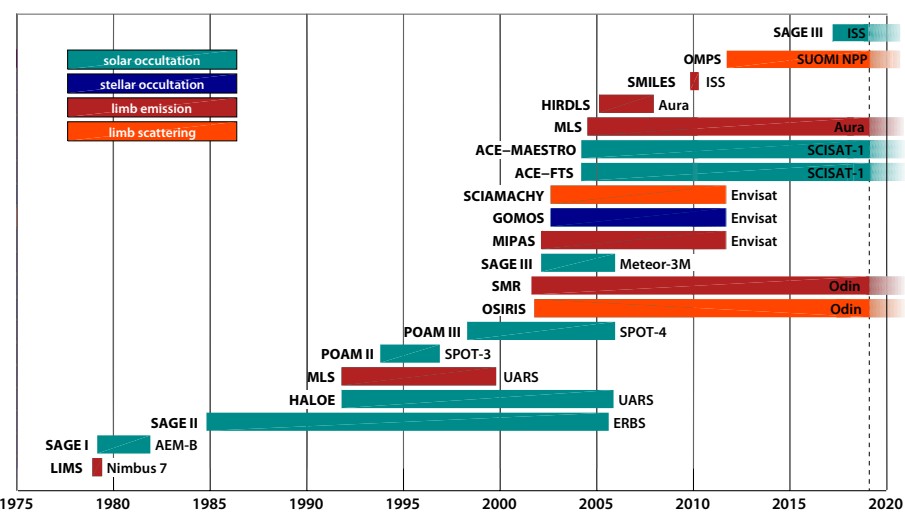

**Figure 1.** Mission lifetime of limb satellite instruments (left hand side of bars) evaluated within the SPARC Data Initiative. Also indicated are the mission platforms (right hand side of bars). The colors classify the instruments according to their observation geometry. Note that the SPARC Data Initiative Report (SPARC, 2017) only evaluated zonal monthly mean datasets up to 2010. Here, we evaluate the datasets out to 2018. Seven satellite limb sounders remain currently in space: Aura-MLS, ACE-FTS, ACE-MAESTRO, Odin/SMR, OSIRIS, OMPS, and SAGE III/ISS, of which the first five long passed their expected lifetimes.

**SPARC Data Initiative**

Columns: $O_3$, $H_2O$, $CH_4$, $N_2O$, $CCl_3F$, $CCl_2F_2$, $CO$, $HF$, $SF_6$, $NO$, $NO_2$, $NO_x$, $HNO_3$, $HNO_4$, $N_2O_5$, $ClONO_2$, $NO_y$, $HCl$, $ClO$, $HOCl$, $BrO$, $OH$, $HO_2$, $CH_2O$, $CH_3CN$, aerosol

Rows: ACE-FTS, Aura-MLS, GOMOS, HALOE, HIRDLS, LIMS, MAESTRO, MIPAS, OSIRIS, POAM II, POAM III, SAGE I, SAGE II, SAGE III, SCIAMACHY, SMILES, Odin/SMR, UARS-MLS, OMPS, SAGE III/ISS

d   derived with help of a chemical box model

m   merged and derived from OSIRIS $NO_2$ and Odin/SMR $HNO_3$ data

lc   with limited coverage

**Figure 2.** Coloured boxes indicate SPARC Data Initiative zonal monthly mean timeseries of atmospheric constituents, listed by instruments and available from Zenodo (doi:10.5281/zenodo.4265393). Dark blue are timeseries originally submitted and evaluated in SPARC (2017), light blue updated timeseries based on new data versions, and orange newly added instruments and/or timeseries.

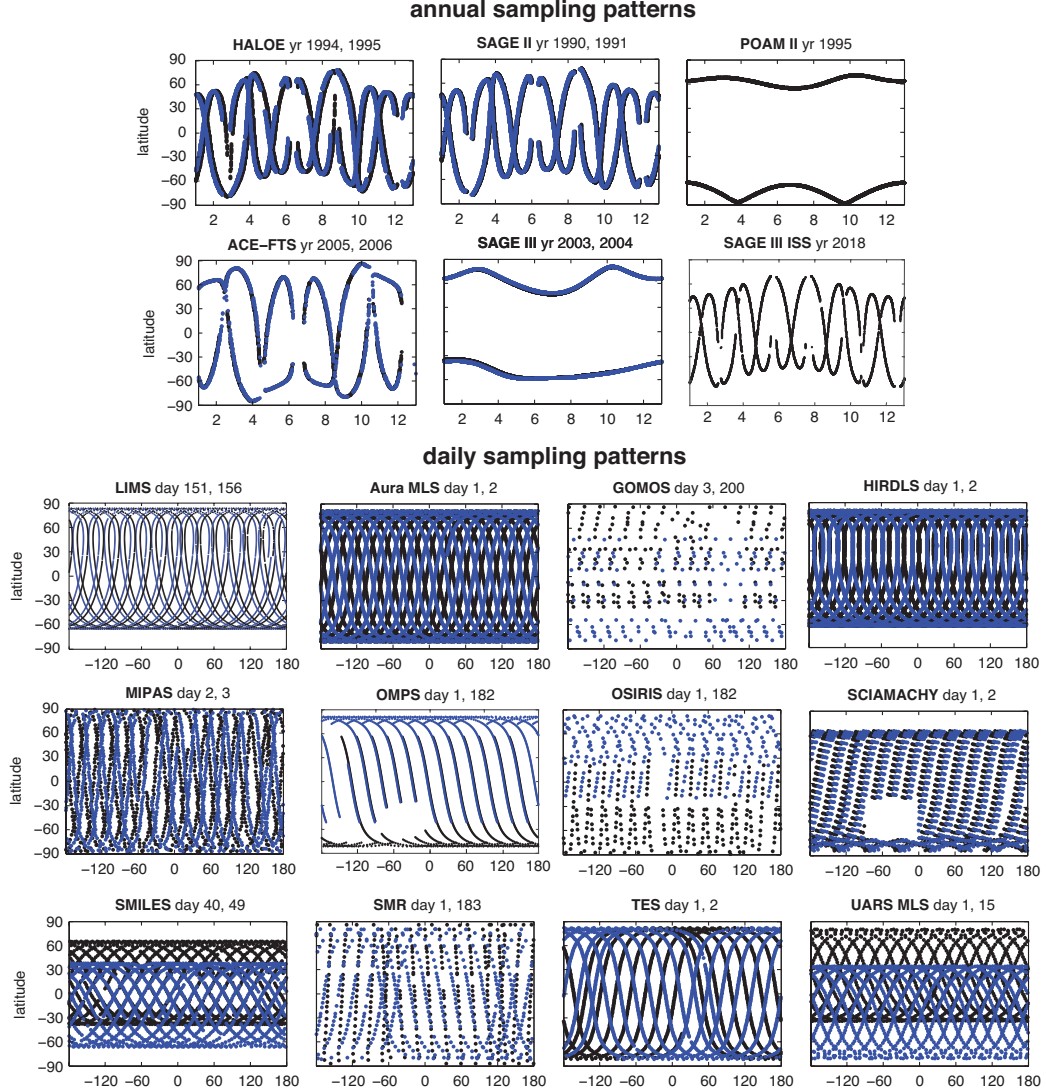

**Figure 3.** Representative sampling patterns for the instruments are shown in time-latitude space for solar occultation sounders to reflect annual sampling patterns (upper two rows) and in longitude-latitude space for emission/scattering and stellar occultation sounders to reflect daily sampling patterns (lower three rows). Different years or days are chosen to give a sense of change in the observed sampling patterns over time. See also Figure 1 in Toohey et al. (2013) for the resulting measurement density in latitude-time space for the original SPARC Data Initiative instruments. Note, the sampling patterns of ACE-MAESTRO and POAM III are the same for ACE-FTS and POAM II, respectively. The sampling pattern of SAGE I is very similar to that of SAGE II and HALOE. The gap in the sampling seen in OMPS and SCIAMACHY over South America is the result of the South Atlantic Anomaly, a dip in Earth's magnetic field that allows charged particles to penetrate lower into the atmosphere and as a consequence causes irregularities in the recorded spectral signals by these instruments.

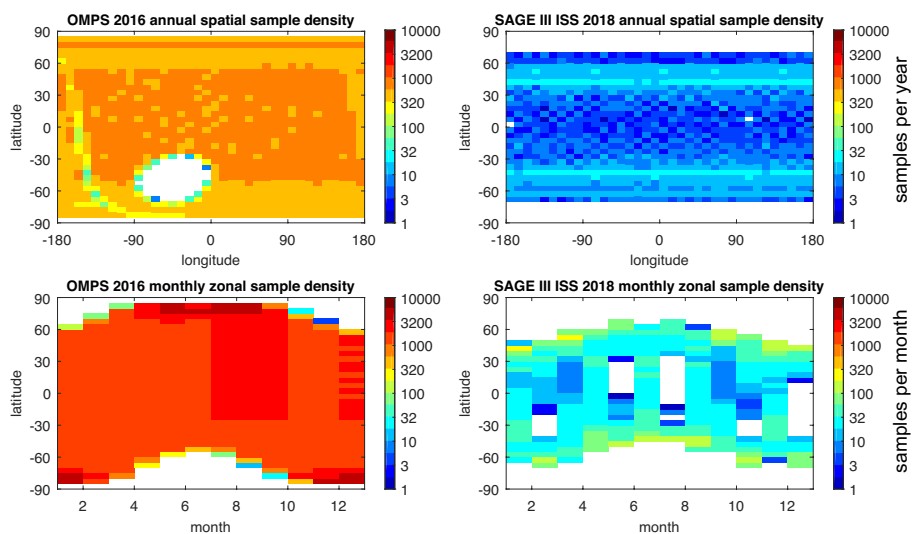

**Figure 4.** Annual (upper panels) and monthly (lower panels) sample density for OMPS-LP SUOMI (left) and SAGE III/ISS (right). Note that for OMPS-LP the daily and for SAGE the annual sampling pattern is shown. The OMPS sampling gap over South America results from a filter that removes measurements affected by the South Atlantic Anomaly, which causes increased noise in measured radiances from transient particle strikes to the instrument detector.

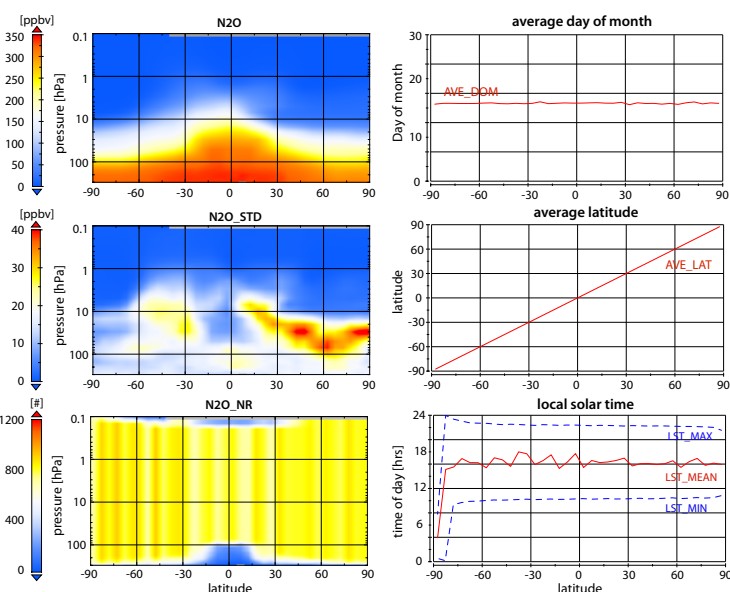

**Figure 5.** Variables in a typical data file that follows SPARC Data Initiative standards are $N_2O$, $N_2O$ standard deviation ($N_2O\_STD$), $N_2O$ number, average day of month, average latitude, and minimum, mean, and maximum local solar time (LST_MIN, LST_MAX, and LST_MEAN). This example shows April data from the 2008 MIPAS zonal monthly mean file.

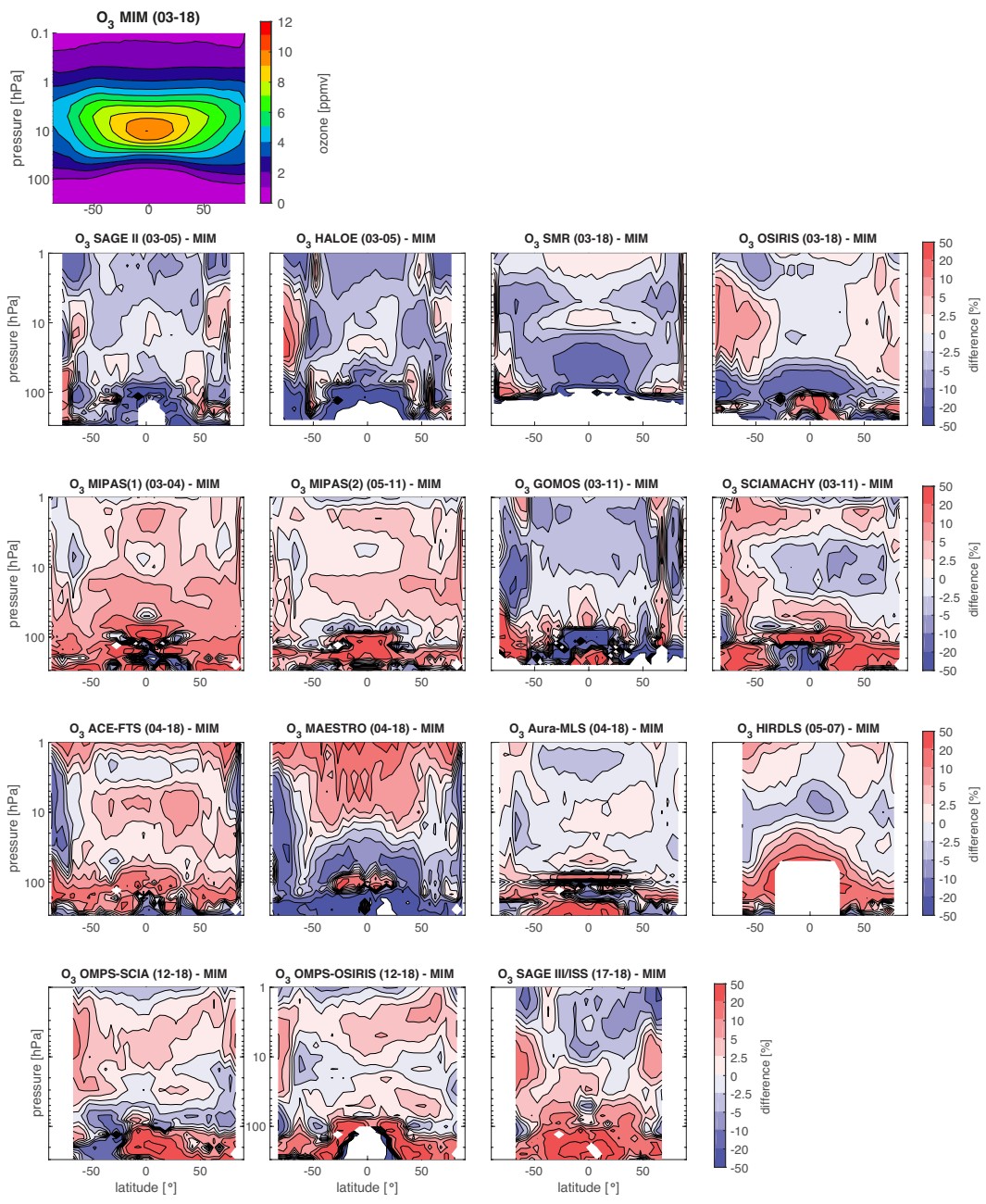

**Figure 6.** Cross-sections of the MIM annual zonal mean ozone for 2003-2018 and differences between the individual instruments and the MIM are shown (update from Figure 2 in Tegtmeier et al., 2013). The MIM includes SAGE II, HALOE, SMR, OSIRIS, MIPAS(1) and MIPAS(2), GOMOS, SCIAMACHY, ACE-FTS, ACE-MAESTRO, Aura-MLS, HIRDLS, IUP-OMPS, USask-OMPS-LP and SAGE III/ISS. Note that while none of the instruments covers the full time period, detailed evaluations of shorter time periods (e.g., 2012-2018, 2005-2010) give very similar results.

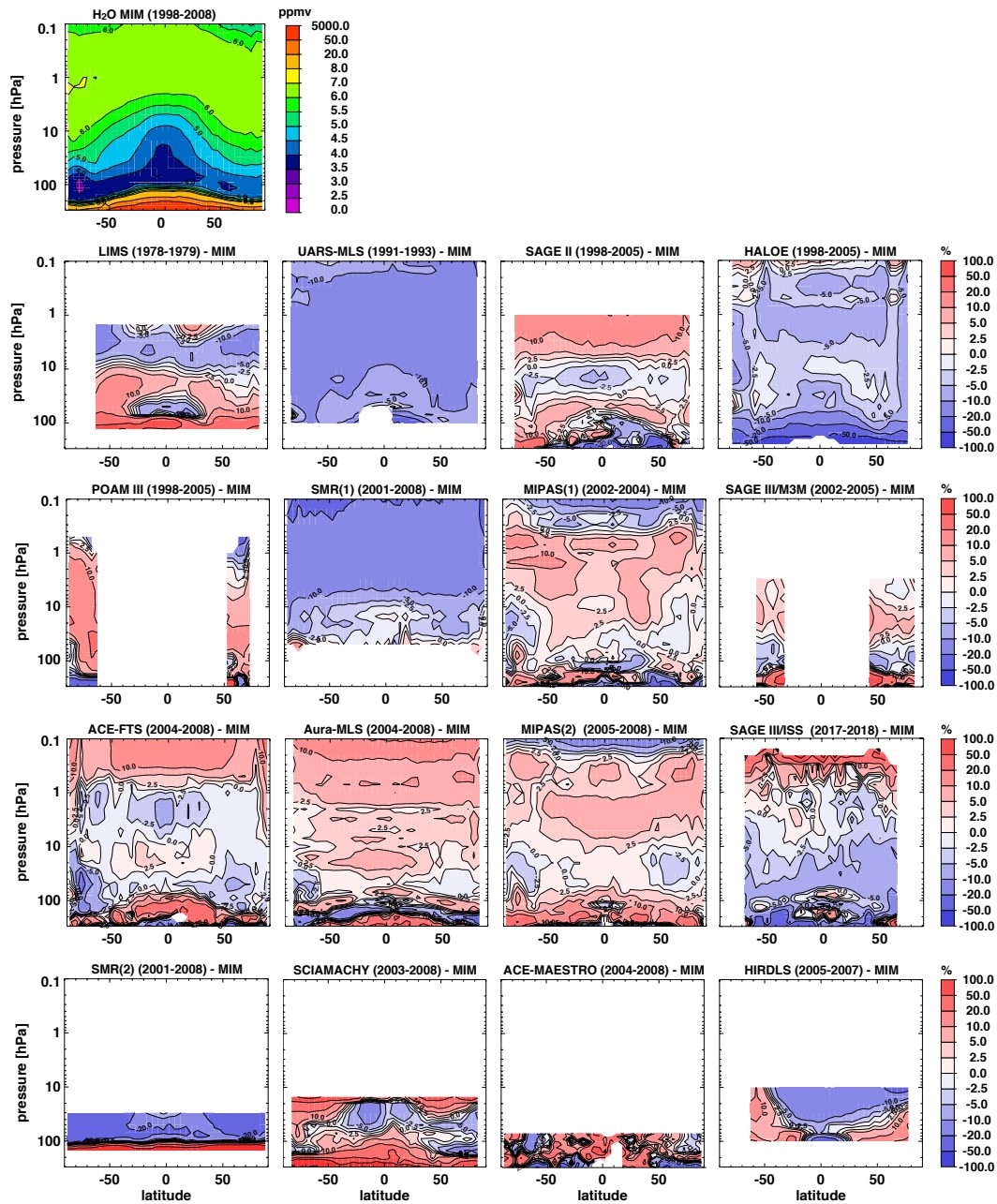

**Figure 7.** Cross-sections of the MIM annual zonal mean water vapour for 1998-2008 and differences between the individual instruments and the MIM are shown (update from Figure 5 in Hegglin et al., 2013). Note that LIMS, UARS-MLS, ACE-MAESTRO, SMR(2), HIRDLS, and SAGE III/ISS are not included in the calculation of the MIM to allow for a more direct comparison with Hegglin et al. (2013). Note that while none of the instruments covers the full time period, detailed evaluations of shorter time periods (e.g., 2005-2010) give very similar results.

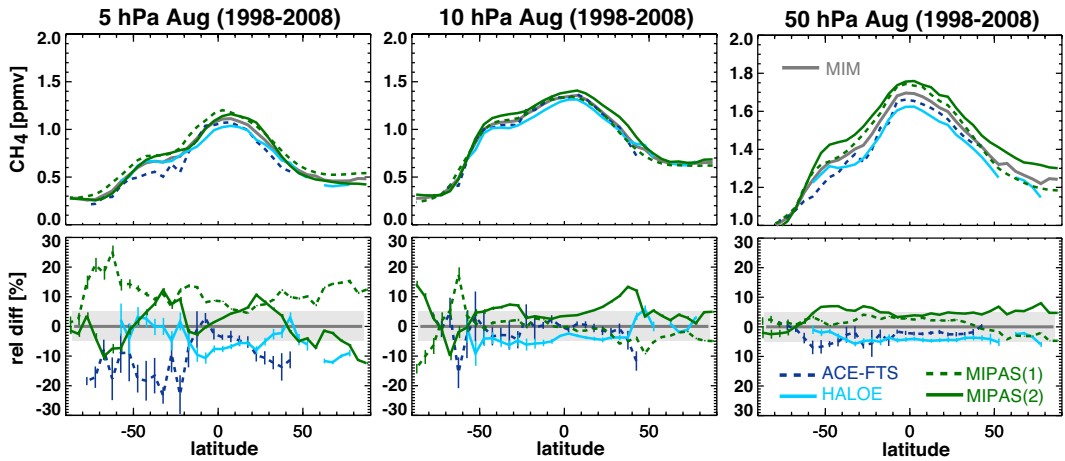

**Figure 8.** Meridional profiles of zonal monthly mean $CH_4$ at 5, 10, and 50 hPa and averaged over 1998-2008 are shown for the different instruments and the MIM (upper panels). Differences between the individual instruments and the MIM are shown in the lower panels. The grey shading indicates where the relative differences are smaller than $\pm 5\%$ (thus where the datasets show very good to excellent agreement). Error bars indicate the uncertainty in the relative differences based on the SEM of each instrument.

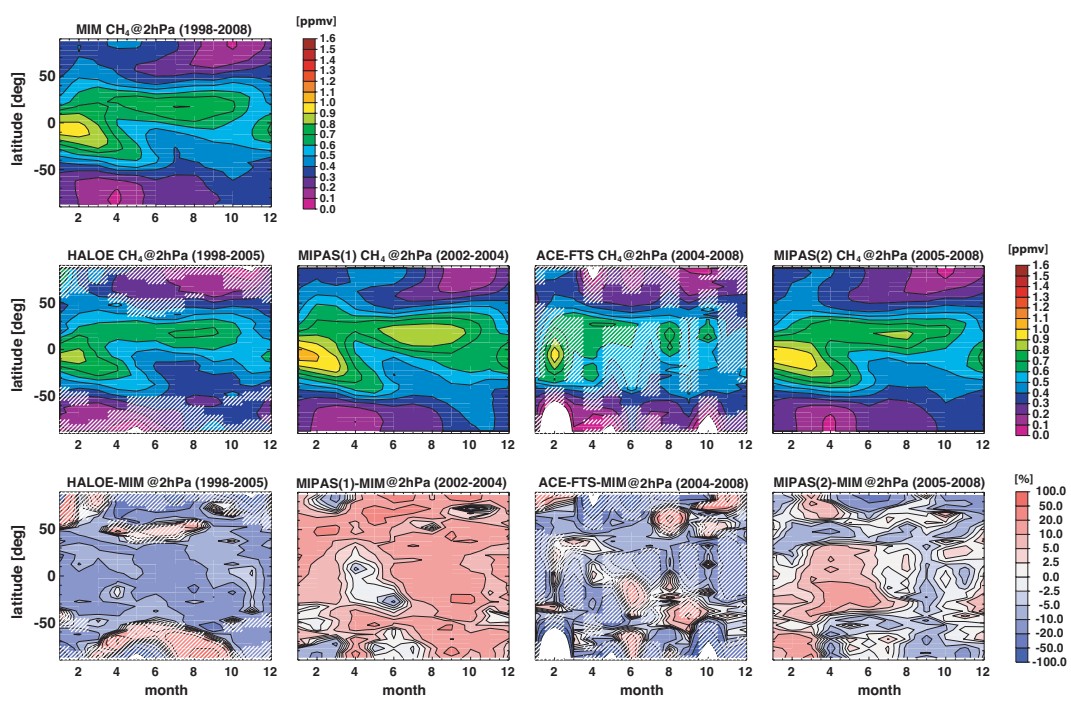

**Figure 9.** Latitude-time evolution of zonal monthly mean $CH_4$ at 2 hPa and averaged over 1998-2008. Shown are absolute values for the MIM (top panel) and the different instruments (middle row), and for the relative differences with respect to the MIM (lower row). Note that HALOE and ACE-FTS show linearly interpolated fields, with hatched regions indicating where no measurements are available.

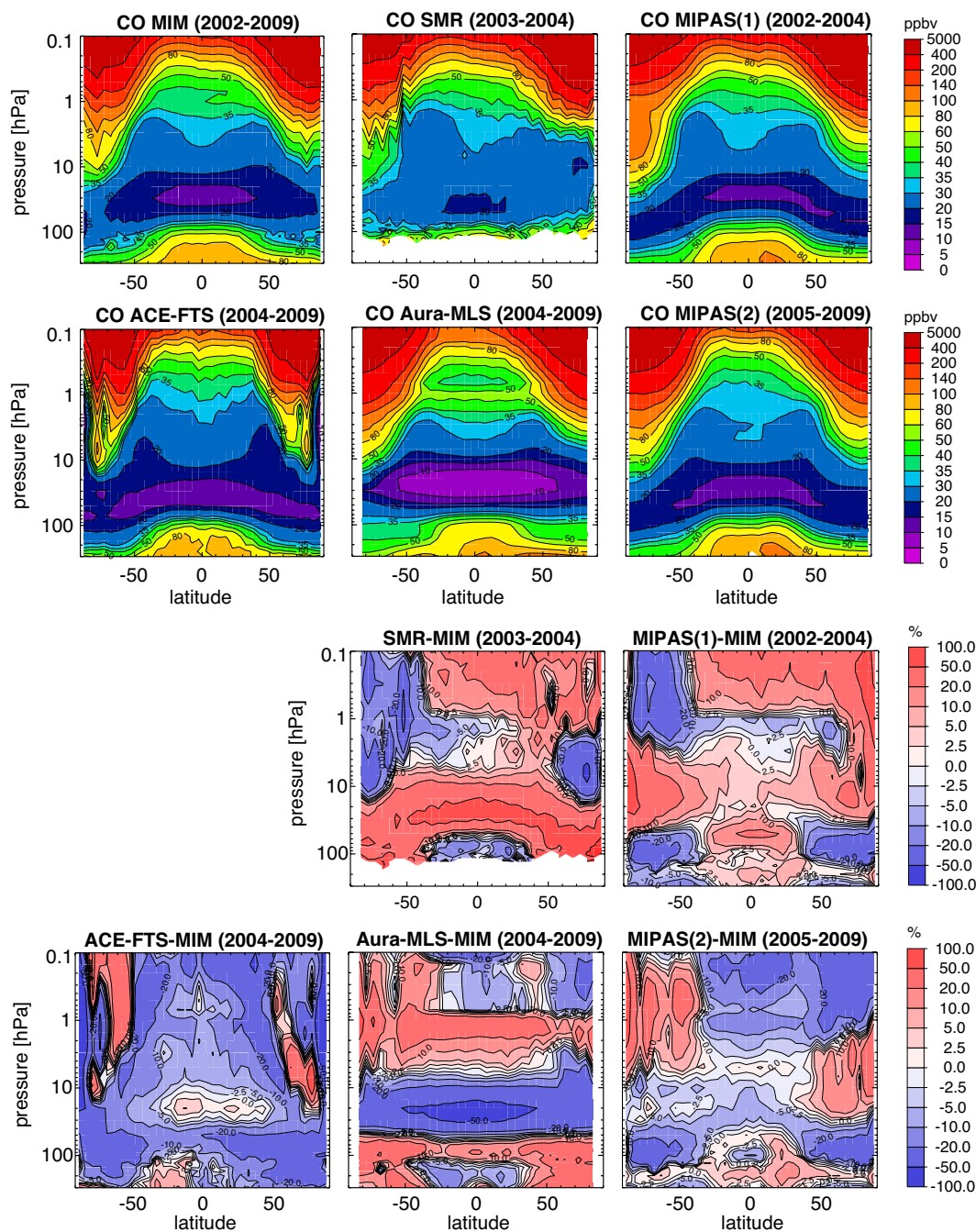

**Figure 10.** Annual zonal mean CO cross sections are shown for the MIM, SMR, MIPAS(1), ACE-FTS, MIPAS(2), and Aura-MLS averaged over 2002-2009 (upper half). Also shown are the relative differences between the individual instruments and the MIM (lower half). Note, SMR and MIPAS(1) are excluded from the calculation of the MIM.

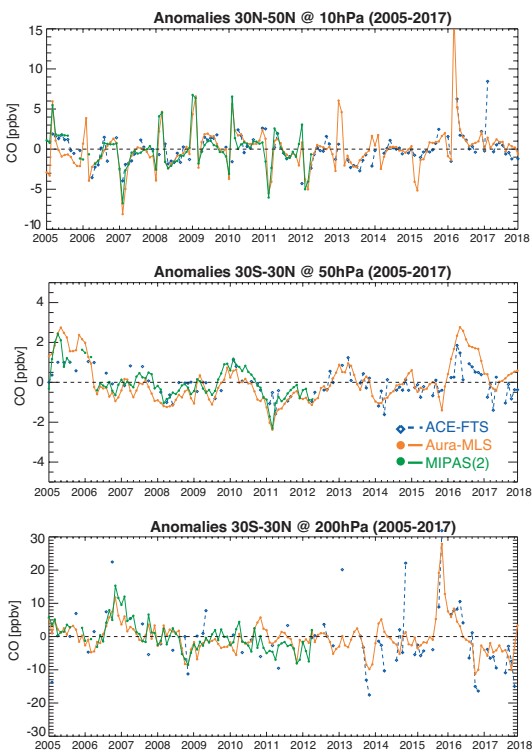

**Figure 11.** Deseasonalized CO anomalies from ACE-FTS, MIPAS(2), and Aura-MLS are shown for $30°$N-$50°$N at 10 hPa (upper panel), and 50 hPa (middle panel) and 200 hPa (lower panel) for $30°$S-$30°$N and the time period 2005-2017.

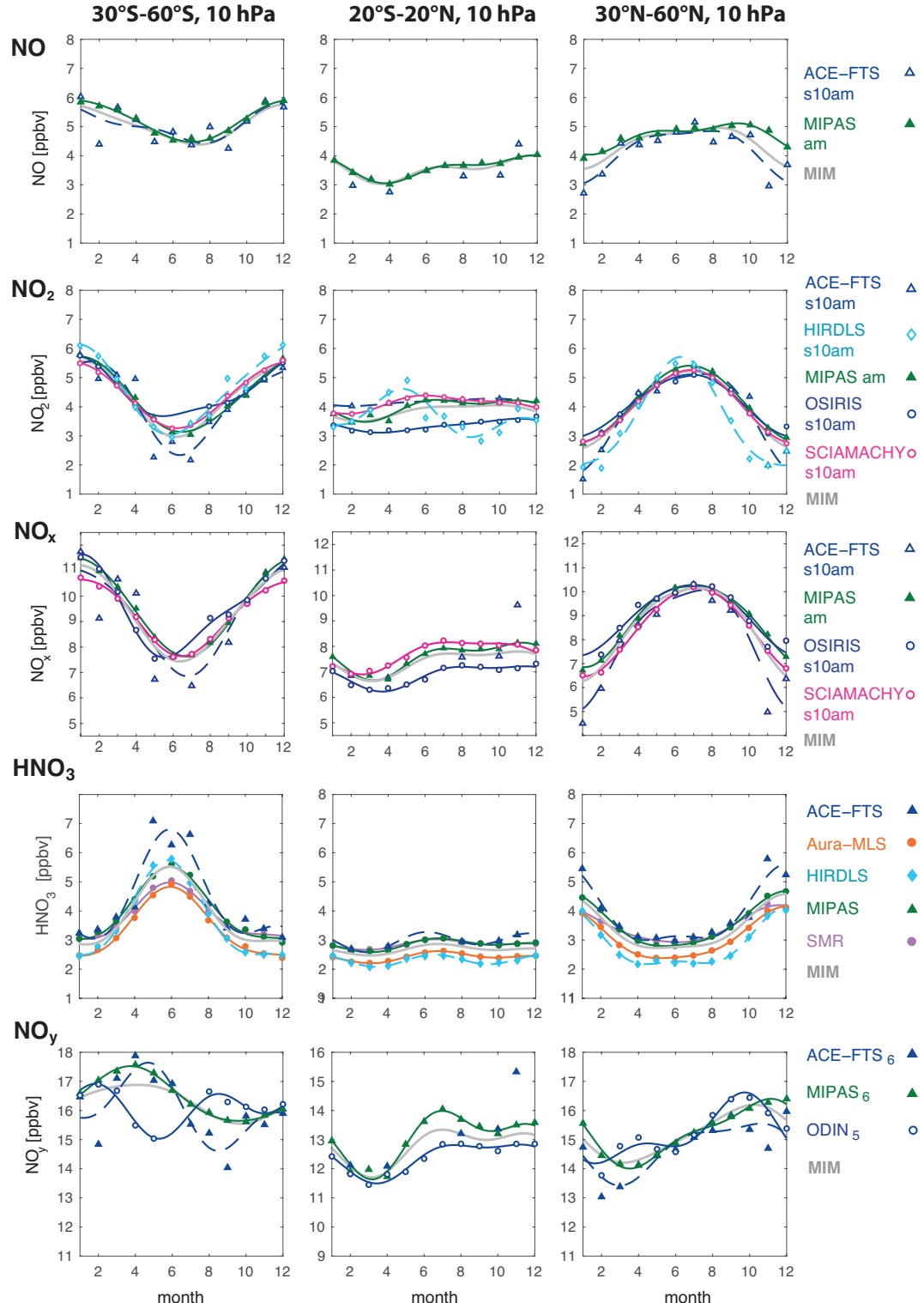

**Figure 12.** The seasonal cycle of NO, $NO_2$, NOx, $HNO_3$ and $NO_y$ is displayed for the SH mid-latitudes (30°S-60°S, leftmost panels), the tropics (10 hPa 20°S-20°N, middle panels), and NH mid-latitudes (30°N-60°N, leftmost panels) at 10 hPa for the time period 2005-2010. Note, the NO, $NO_2$, and NOx seasonal cycles are based on 10 am datasets. The *s10pm* denotes datasets scaled to 10 pm.

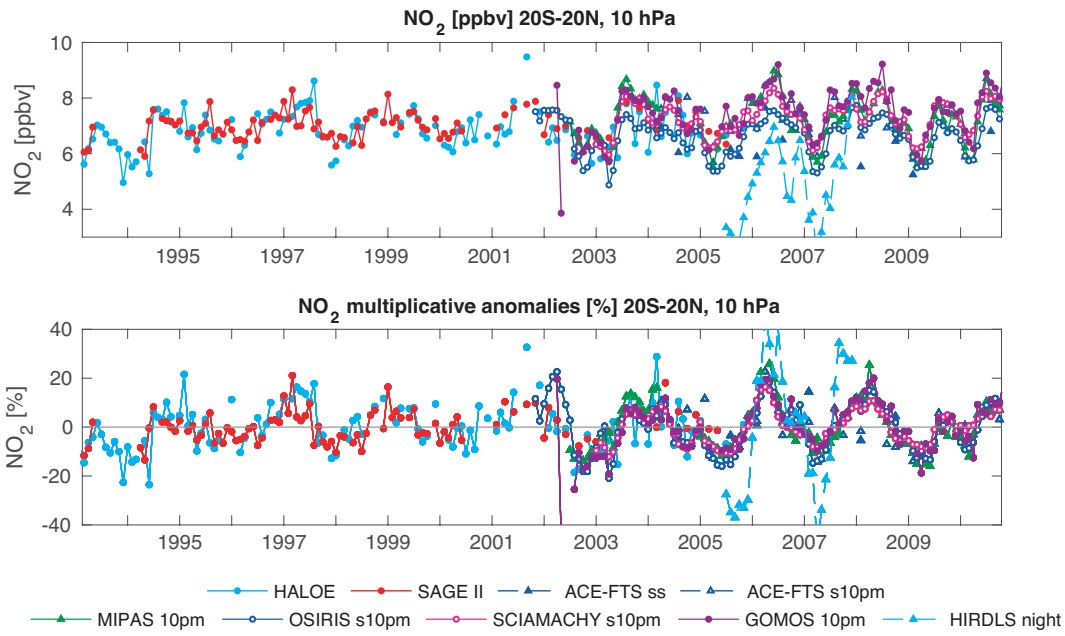

**Figure 13.** Time series of tropical $NO_2$ mean values (upper panels) and deseasonalized anomalies (lower panels) between $20°$S-$20°$N at 10 hPa for 1993-2010. Data sets correspond to local sunset or to 10 pm LSTs as described in the text. The *s10pm* denotes zonal monthly mean fields scaled to 10 pm, the *ss* zonal monthly mean fields from sunset measurements.

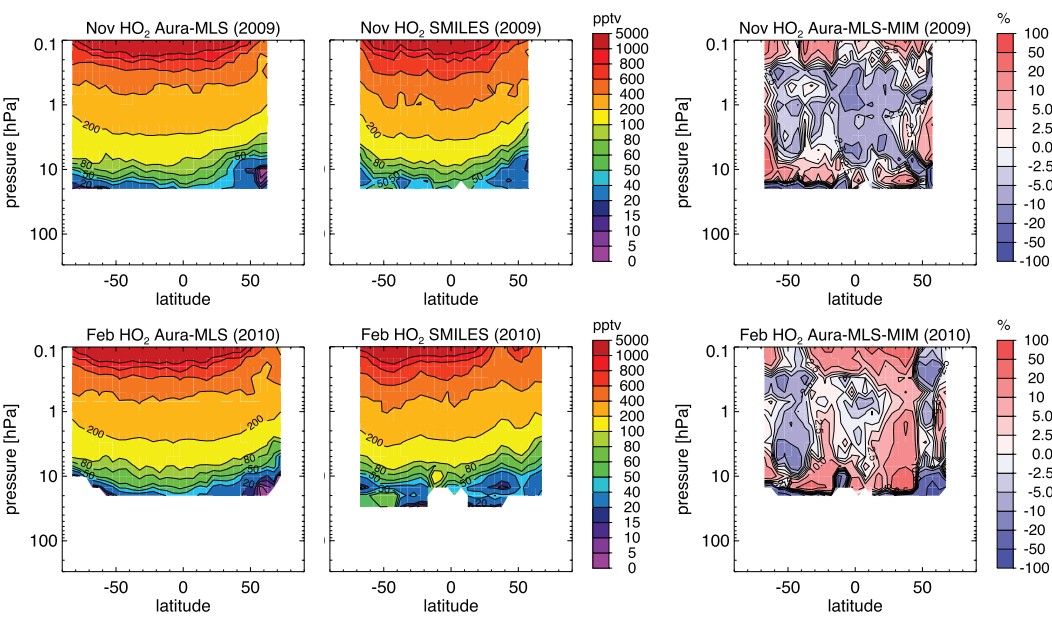

**Figure 14.** zonal monthly mean HO$_2$ cross sections for Aura-MLS and SMILES daytime (LT) data (left two columns) and Aura-MLS relative differences from the MIM (right column) are shown for November 2009 (upper row) and February 2010 (lower row), respectively. Note, the SMILES relative differences from the MIM would look exactly opposite of these figures and thus are not shown.

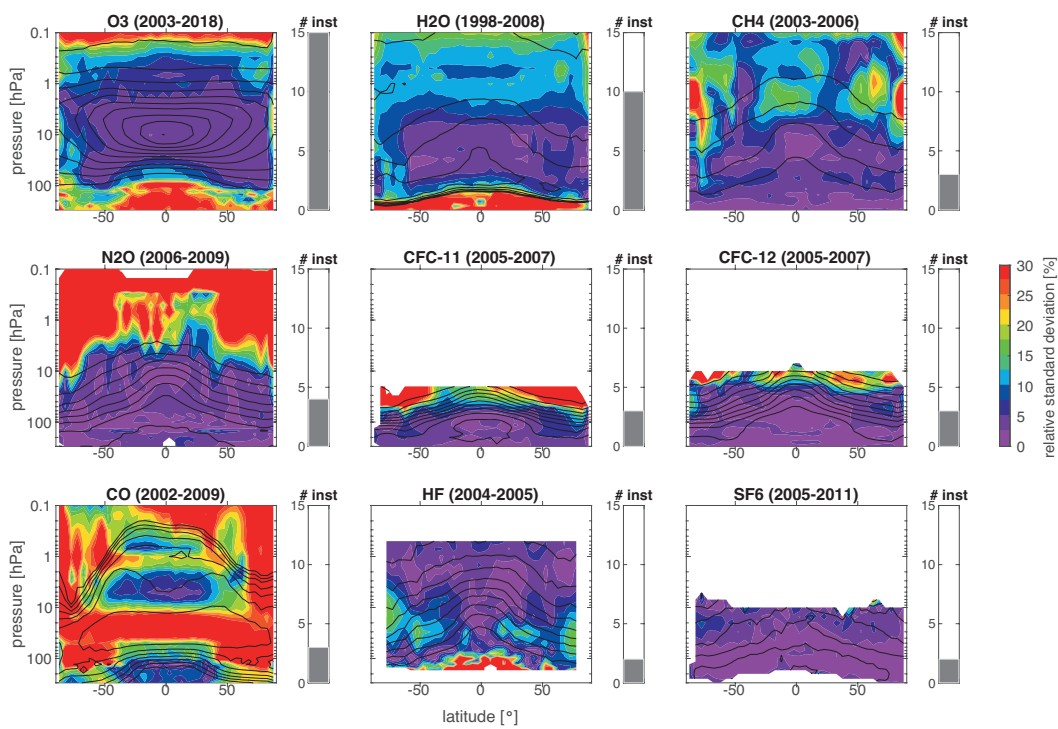

**Figure 15.** Synopsis of the uncertainty in the annual zonal mean state of the longer-lived species evaluated within the SPARC Data Initiative. The relative standard deviation over all instruments' multi-annual zonal mean datasets is presented for different chemical trace gas species (colour contours). The relative standard deviations are calculated by dividing the absolute standard deviations by the MIM. The black contour lines in each panel represent the MIM trace gas distribution for each species. The number of instruments included is given by the right-hand grey bar. Note that the time periods used depend on the availability of the instruments included in the assessment and hence differ from trace gas to trace gas.

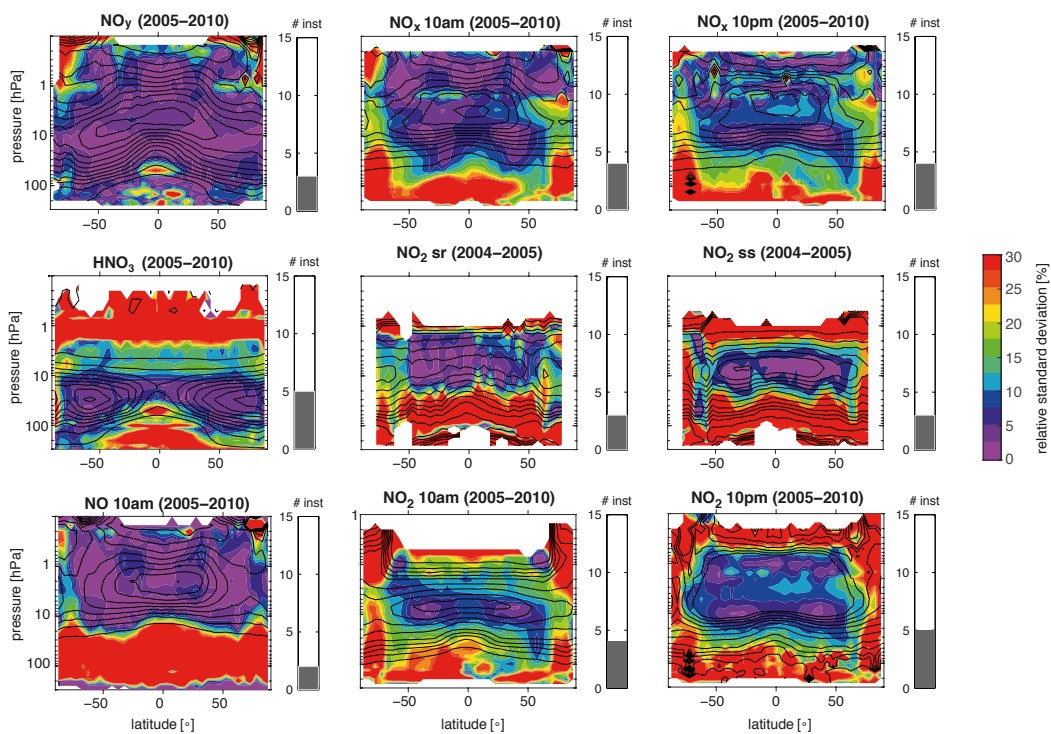

**Figure 16.** Same as Figure 14, but for nitrogen containing species. The assessment of the uncertainty in the annual mean state of NO, $NO_x$, and $NO_2$ is based on gridded datasets corresponding to 10am and 10pm, and for the latter also on datasets corresponding to local sunrise (sr) and local sunset (ss). Note that some of the included datasets have been derived by scaling the individual measurements with a chemical box model to 10am/10pm local solar time (LST). See SPARC (2017) for more detailed information. Reactive nitrogen ($NO_x$) is here defined as $NO+NO_2$. The odd nitrogen family ($NO_y$) is defined as $NO_x+HNO_3+2xN_2O_5+ClONO_2+HNO_4$. For $N_2O_5$, $ClONO_2$, and $HNO_4$ an assessment of the uncertainty in the annual mean field cannot be provided since no data products at the same local solar time are available.

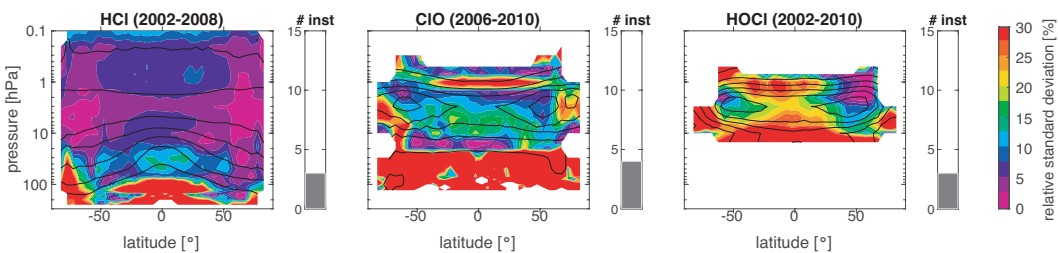

**Figure 17.** Same as Figure 14, but for chlorine-containing species. The assessment of the uncertainty in the annual mean state is based on ClO daytime and HOCl night-time datasets. Note that for ClO, the dataset from SMR is included which has been derived by scaling the individual measurements with a chemical box model to 1:30 pm LST. See SPARC (2017) for more detailed information.

**Table 1.** Data versions used for the construction of the gridded zonal monthly mean datasets submitted to the SPARC Data Initiative assessment and as deposited in the Zenodo data archive (except for aerosol). Italics indicate data versions that have been updated since SPARC (2017). Bold italics indicates datasets that have been added recently, i.e., were not part of SPARC (2017).

| Instrument | $O_3$ | $H_2O$ | $CH_4$ | $N_2O$ | $CCl_3F$ | $CCl_2F_2$ | CO | HF | $SF_6$ | NO | $NO_2$ | $NO_x$ | $HNO_3$ |
|---|---|---|---|---|---|---|---|---|---|---|---|---|---|
| ACE-FTS | *v3.6* | *v3.6* | *v3.6* | *v3.6* | *v3.6* | *v3.6* | *v3.6* | *v3.6* | *v3.6* | *v3.6* | *v3.6* | *v3.6* | *v3.6* |
| Aura-MLS | *v4.2* | *v4.2* | | *v4.2* | | | *v4.2* | | | | | | *v4.2* |
| GOMOS | *v6.01* | | | | | | | | | | *v6.01* | | |
| HALOE | v19 | v19 | v19 | | | | | v19 | | v19 | v19 | v19 | |
| HIRDLS | *v7.0* | ***v7.0*** | | ***v7.0*** | *v7.0* | *v7.0* | | | | | *v7.0* | | *v7.0* |
| LIMS | v6.0 | v6.0 | | | | | | | | | v6.0 | | v6.0 |
| ACE-MAESTRO | *v3.13* | ***v31*** | | | | | | | | | | | |
| MIPAS(1) | *v21* | *v20* | *v21* | *v21* | *v20* | *v20* | *v20* | | *v20* | *v20* | *v20* | *v20* | *v22* |
| MIPAS(2) | *v224* | *v220* | *v224* | *v224* | *v220* | *v220* | *v220* | | *v222* | *v220* | *v220* | *v220* | *v224* |
| OSIRIS | *v5.10* | | | | | | | | | | v3.0 | v3.0 | |
| POAM II | v6.0 | | | | | | | | | | v6.0 | | |
| POAM III | v4.0 | v4.0 | | | | | | | | | v4.0 | | |
| SAGE I | v5.9 | | | | | | | | | | | | |
| SAGE II | *v7.0* | *v7.0* | | | | | | | | | *v7.0* | | |
| SAGE III | v4.0 | v4.0 | | | | | | | | | v4.0 | | |
| SCIAMACHY | *v3-5* | *v4-2* | | | | | | | | | *v4-0* | *v4-0* | |
| SMILES | v2.1.5 | | | | | | | | | | | | v2.0.1 |
| Odin/SMR(1) | *v3.1* | v2.1 | | v2.1 | | | v2.1 | | | v2.1 | | | v2.0 |
| Odin/SMR(2) | | v2.0 | | | | | | | | | | | |
| UARS-MLS | v5 | v6 | | | | | | | | | | | v6 |
| IUP-OMPS | ***v2-6*** | | | | | | | | | | | | |
| USask-OMPS | ***v1.1.0*** | | | | | | | | | | | | |
| SAGE III/ISS | ***v5.1*** | ***v5.1*** | | | | | | | | | | | |

**Table 2.** Table 1 continued. Note, Odin $NO_y$ v3.0 is based on both OSIRIS and SMR data, hence has a double entry.

| Instrument | $HNO_4$ | $N_2O_5$ | $ClONO_2$ | $NO_y$ | HCl | ClO | HOCl | BrO | OH | $HO_2$ | $CH_2O$ | $CH_3CN$ | aerosol |
|---|---|---|---|---|---|---|---|---|---|---|---|---|---|
| ACE-FTS | *v3.6* | *v3.6* | *v3.6* | *v3.6* | *v3.6* | | | | | | *v3.6* | | |
| Aura-MLS | | | | | *v4.2* | v3.3 | v3.3 | | v3.3 | v3.3 | | | |
| GOMOS | | | | | | | | | | | | | *v6.01* |
| AERGOM | | | | | | | | | | | | | *v1* |
| HALOE | | | | | v19 | | | | | | | | v19 |
| HIRDLS | | *v7.0* | *v7.0* | | | | | | | | | | |
| MIPAS(1) | *v20* | *v21* | *v21* | *v22* | | *v20* | *v20* | | | | *v20* | | |
| MIPAS(2) | *v220* | *v222* | *v222* | *v224* | *v220* | | | | | | | | |
| OSIRIS | | | | *v3.0* | | | | v5 | | | | | v5.7 |
| POAM II | | | | | | | | | | | | | v6.0 |
| POAM III | | | | | | | | | | | | | v4.0 |
| SAGE II | | | | | | | | | | | | | v7.0 |
| SAGE III/M3M | | | | | | | | | | | | | *v4.0* |
| SCIAMACHY | | | | | | | | v4.1 | | | | | *v1.4* |
| SMILES | | | | | v2.1.5 | v2.0.1 | v2.1.5 | v2.0.1 | | v2.1.5 | | v2.0.1 | |
| Odin/SMR | | | | *v3.0* | | v2.1 | | | | v2.1 | | | |

**Table 3.** Instruments classified according to their observation geometry and wavelength categories. Only instruments that participated in the SPARC Data Initiative are listed.

| | Microwave/Sub-mm<br>100 $\mu$m - 10 cm | Mid-IR<br>2.5 - 20 $\mu$m | Near-IR<br>1 - 2.5 $\mu$m | VIS/UV<br><1 $\mu$m |
|---|---|---|---|---|
| **Limb emission** | UARS-MLS<br>Aura-MLS<br>Odin/SMR<br>SMILES | LIMS<br>MIPAS<br>HIRDLS | | |
| **Solar occultation** | | HALOE<br>ACE-FTS | POAM II/III<br>SAGE I/II/III<br><br>SAGE III/ISS | POAM II/III<br>SAGE I/II/III<br>ACE-MAESTRO<br>SAGE III/ISS |
| **Stellar or lunar occultation** | | | SAGE III/ISS | GOMOS<br>SAGE III/ISS |
| **Limb scattering** | | | SCIAMACHY | SCIAMACHY<br>OSIRIS<br>OMPS-LP |

**Table 4.** Satellite instrument characteristics including satellite platform, observation period, spatial coverage, vertical range, vertical resolution, data density, local time (LT) at equator, LT of measurement, inclination, and instrument references. Note that the vertical range is generally species-dependent and the vertical resolution is often species- and altitude-dependent, see Tables 9-16 for details. For instruments providing data on a pressure grid, the vertical range is also given in km.

| Instrument Platform | Obs. period | Spatial coverage | Vertical range | Vertical res. | Data density [prof./day] | LT at equator | LT of meas | Incl. | References |
|---|---|---|---|---|---|---|---|---|---|
| **LIMS** Nimbus 7 | 11/1978 to 05/1979 | 64°S-84°N (daily) | 250-0.01 hPa 10-80 km | 3.7 km | 3000 | a: 11:51 am d: 11:51 pm | a: 1 pm d: 11 pm | 99.3° | Gille and Russell (1984) |
| **SAGE I** AEM-2 | 2/1979 to 11/1981 | 75°S-75°N (~one month) | surface/cloud top to 55 km | 1 km | 30 | N/A | sunrise sunset | 56° | McCormick et al. (1979) |
| **SAGE II** ERBS | 10/1984 to 08/2005 | 75°S-75°N (~one month) | surface/cloud top to 70 km | 1 km | 30 | N/A | sunrise sunset | 57° | Mauldin III et al. (1985) McCormick et al. (1989) |
| **UARS-MLS** UARS | 10/1991 to 10/1999 | 80°S-80°N (~two months) | 100-0.01 hPa 17-80 km | 3.5-5 km | 1318 | N/A | N/A | 57° | Waters et al. (1993) Livesey et al. (2003) |
| **HALOE** UARS | 10/1991 to 11/2005 | 75°S-75°N (~one month) | 250-0.002 hPa 10-90 km | 2.5 km | 30 | N/A | sunrise sunset | 57° | Russell et al. (1993) |
| **POAM II** SPOT-3 | 10/1993 to 11/1996 | 88°S-63°S 55°N-71°N (one year) | 15-50 km | 1 km | 28 | a: 10:30 pm d: 10:30 am | N/A | 98.7° | Glaccum et al. (1996) |
| **POAM III** SPOT-4 | 04/1998 to 12/2005 | 88°S-63°S 55°N-71°N (over one year) | 5-60 km | 1 km | 28 | a: 10:30 pm d: 10:30 am | N/A | 98.7° | Lucke et al. (1999) |
| **SMR** Odin | 07/2001 to present | 82°S-82°N (daily) | 100-0.001 hPa 15-120 km | 2-6 km | 600-975 | a: 6:30 pm d: 6:30 am | a: 6:30 pm d: 6:30 am | 97.8° | Murtagh et al. (2002) |
| **OSIRIS** Odin | 10/2001 to present | 82°S-82°N (daily, no winter) | 10-60 km | 2 km | 300-975 | a: 6:30 pm d: 6:30 am | a: 6:30 pm d: 6:30 am | 97.8° | Murtagh et al. (2002) |
| **SAGE III** Meteor-3M | 02/2002 to 12/2005 | 60°S-30°S 40°N-80°N (~one month) | surface/cloud top to 100 km | 1 km | 30 | a: 9:30 am d: 9:30 pm | sunrise sunset | 99.6° | Mauldin et al. (1998) Thomason et al. (2010) |

**Table 5.** Table 4 continued. Note, the low (high) data data density entries in MIPAS refer to the two different measurement modes MIPAS had been measuring in before and after 2004, referred to as MIPAS(1) and MIPAS(2) in later tables, respectively. Also note, the ascending part of the SCIAMACHY orbit lies mostly in darkness, resulting in only few measurements which are not included in the SPARC Data Initiative gridded datasets (thus the ascending measurement LT is not listed).

| Instrument Platform | Obs. period | Spatial coverage | Vertical range | Vertical res. | Data density | LT at equator | LT of meas | Incl. | References |
|---|---|---|---|---|---|---|---|---|---|
| **GOMOS** Envisat | 04/2002 to 04/2012 | 90°S-90°N (daily, no summer poles) | 15-100 km | 2-4 km | 100-300 (night obs only) | a: 10 pm d: 10 am | a: 10-12 pm d: 8-10.30 am | 98.55° | Bertaux et al. (2010) |
| **MIPAS** Envisat | 03/2002 to 04/2012 | 90°S-90°N (daily) | 6 km/cloud top to 70 km | 3.5-5 km 2.7-3.5 km | 1000 1300 | a: 10 pm d: 10 am | a: 10 pm d: 10 am | 98.55° | Fischer et al. (2008) |
| **SCIAMACHY** ENVISAT | 08/2002 to 04/2012 | 85°S-85°N (65° for winter hemisphere) | 10-60 km | 3-5 km | 364-1456 | a: 10 pm d: 10 am | d: 10 am | 98.55° | Burrows et al. (1995) |
| **ACE-FTS** SCISAT-1 | 03/2004 to present | 85°S-85°N (~one season) | 5-95 km | 3-4 km | 30 | N/A | N/A | 74° | Bernath et al. (2005) |
| **ACE-MAESTRO** SCISAT-1 | 03/2004 to present | 85°S-85°N (~one season) | 5-60 km | 2 km | 30 | N/A | N/A | 74° | McElroy et al. (2007) |
| **Aura-MLS** Aura | 07/2004 to present | 82°S-82°N (daily) | 316-0.002 hPa 9-150 km | 3 km | 3500 | a: 1:43 pm d: 1:43 am | N/A | 98.21° | Waters et al. (2006) |
| **HIRDLS** Aura | 02/2005 to 03/2008 | 65°S-82°N (daily) | 420-0.1 hPa 10-55 km | 1 km | 5600 | a: 1:43 pm d: 1:43 am | N/A N/A | 98.21° | Gille et al. (2008) |
| **SMILES** ISS | 10/2009 to 04/2010 | 38°S-65°N (daily) | 100-0.0005 hPa 16-96 km | 3-5 km | 1620 | N/A | N/A | 51.6° | Kikuchi et al. (2010) |
| **SAGE III** ISS | 06/2017 to present | 75°S-75°N (~one month) | surface/cloud top to 100 km | 0.75-1 km | 30 | N/A | sunrise sunset | 51.6° | Wang et al. (2020) |
| **OMPS-LP** Suomi NPP | 02/2012 to present | 81.5°S-81.5°N | 8-60 km | 2-5 km | 2000 | a 1:30 pm d 1:30 am | 1:30 pm | 98.7° | Jaross et al. (2014) |

**Table 6.** Description of the content included in each of the SPARC Data Initiative data files, here using $N_2O$ as an example variable. Each file includes the timeseries of zonal monthly mean data for one year. Not-a-number values are filled in with '-999.0'. See Figure 4 for an example. Note, while much effort has been put into applying a consistent file format across the different instruments, some files may still differ from the description here.

| Variable name | Long name | Variable type |
| --- | --- | --- |
| time | time | 1D |
| plev | pressure | 1D |
| lat | latitude | 1D |
| N2O | volume mixing ratio of N2O in air | Geo3D |
| N2O_NR | number of N2O measurements | Geo3D |
| N2O_STD | volume mixing ratio of N2O in air standard deviation | Geo3D |
| AVE_DOM | average day of month | Geo2D |
| AVE_LAT | average latitude | Geo2D |
| LST_MIN | minimum local solar time | Geo2D |
| LST_MAX | maximum local solar time | Geo2D |
| LST_MEAN | mean of local solar time | Geo2D |

**Table 7.** Terminology used to define agreement between instruments with respect to the multi-instrument mean (*MIM*).

| Description | deviation from MIM |
|---|:---:|
| Excellent agreement | ±2.5% |
| Very good agreement | ±5% |
| Good agreement | ±10% |
| Reasonably good agreement | ±20% |
| Considerable disagreement | ±50% |
| Large disagreement | ±100% |

**Table 8.** Definitions and abbreviations of different atmospheric regions as used in this study. The full height range corresponds to about 9-65 km. The tropopause is latitude dependent (approx. 200-300 hPa in the extratropics and 80-100 hPa in the tropics depending on season), while the transition between the stratosphere and the mesosphere (i.e., the stratopause) is here defined uniformly across all latitudes as the 1 hPa pressure level. Note that the abbreviations are often used in combination (e.g., UTLS for upper troposphere and lower stratosphere and USLM for upper stratosphere and lower mesosphere).

| Region | Abbreviation | Lower boundary | Upper boundary |
|---|---|---|---|
| Upper Troposphere | **UT** | 300 hPa | tropopause |
| Lower Stratosphere | **LS** | tropopause | 30 hPa |
| Middle Stratosphere | **MS** | 30 hPa | 5 hPa |
| Upper Stratosphere | **US** | 5 hPa | 1 hPa |
| Lower Mesosphere | **LM** | 1 | 0.1 hPa |

**Table 9.** Time period, vertical range, vertical resolution, references and other comments for $O_3$ measurements. Note, *tp* refers to tropopause and *c.t.* to cloud top in this table.

| Instrument (version) | Time Period | Vertical range | Vertical Resolution | References | Additional Comments |
|---|---|---|---|---|---|
| LIMS (v6.0) | 11/1978-05/1979 | 10-50 km | 3.7 km | Remsberg et al. (2007; 2021) | |
| SAGE I (v5.9) | 10/1984-08/2005 | surface/c.t. to 50 km | 1-2.5 km | McCormick et al. (1989) | data above 3 hPa excluded |
| SAGE II (v7.0) | 10/1984-08/2005 | surface/c.t. to 70 km | 1 km | Wang et al. (2002) Damadeo et al. (2013) | |
| UARS-MLS (v5) | 10/1991-06/1997 | 18-45 km 45-80 km | 3-4 km 5-8 km | Livesey et al. (2003) | |
| HALOE (v19) | 10/1991-11/2005 | tp to 80 km | 3.5 km | Grooss and Russell (2005) | data below tp excluded |
| SAGE III/M3M (v4.0) | 05/2002-12/2005 | surface/c.t. to 70 km | 1 km | Wang et al. (2006) | only solar products |
| POAM II (v6.0) | 10/1993-11/1996 | 15-50 km | 1 km | Lumpe et al. (1997) Roche et al. (1997) | |
| POAM III (v4.0) | 04/1998-12/2005 | 5-60 km | 1 km | Lumpe et al. (2006) Randall et al. (2003) | |
| SMR (v3.1) | 08/2001-present | 170-0.3 hPa | 3-4 km | Murtagh et al. (2018) | |
| OSIRIS (v5.10) | 11/2001-present | tp to 59.5 km | 2.2-3.5 km | Bourassa et al. (2018) Degenstein et al. (2009) | |
| GOMOS (ALGOM2s) | 04/2002-04/2012 | 15-100 km | 2-3 km | Sofieva et al. (2017) | |
| MIPAS MIPAS(1) (v21) MIPAS(2) (v224) | 03/2002-03/2004 01/2005-04/2012 | 6-68 km 6-70 km | 3.5-5.0 km 2.7-3.5 km | Steck et al. (2007) Laeng et al. (2014) | change in spectral resolution in 2004 |
| SCIAMACHY (v4) | 09/2002-04/2012 | 11-25 km | 3-5 km | Jia et al. (2015) | |
| ACE-FTS (v3.6) | 03/2004-present | 5-95 km | 3-4 km | Sheese et al. (2017) | |
| ACE-MAESTRO (v3.13) | 03/2004-present | 5-60 km | 1-2 km | Bognar et al. (2019) | |
| Aura-MLS (v4.2) | 08/2004-present | 261-0.02 hPa | 2.5-5 km | Livesey et al. (2018) Hubert et al. (2016) | |
| HIRDLS (v7.0) | 02/2005-03/2008 | 422-0.1 hPa | 1 km | Gille and Gray (2012) | data degrade after 12/2007 |
| IUP-OMPS (v2.6) | 02/2012 | 8-60 km | 2-5 km | Arosio et al. (2018) | |
| USask-OMPS (v1.1.0) | 02/2012-present | tp to 58 km | 1.5-2 km | Zawada et al. (2018) | |
| SAGE III/ISS (v5.1) | 06/2017-present | surface/c.t. to 70 km | 0.75 km | Wang et al. (2020) | |

**Table 10.** Time period, vertical range, vertical resolution, references and other comments for $H_2O$ measurements. Note, *tp* refers to tropopause, *p* to pressure, and *c.t.* to cloud top in this table.

| Instrument (version) | Time Period | Vertical range | Vertical Resolution | References | Additional Comments |
|---|---|---|---|---|---|
| **LIMS** (v6.0) | 11/1978-05/1979 | 10-50 km | 3.7 km | Remsberg et al. (2009; 2021) | |
| **SAGE II** (v7.0) | 10/1984-08/2005 | surface/c.t. to 50 km | 1-2.5 km | Thomason et al. (2004) Damadeo et al. (2013) | data above 3 hPa excluded |
| **UARS-MLS** (v6) | 10/1991-03/1993 | 18-50 km 50-80 km | 3-4 km 5-7 km | Pumphrey (1999) | $H_2O$ stops early, radiometer failure |
| **HALOE** (v19) | 10/1991-11/2005 | 10-80 km | 3.5 km | Grooss and Russell (2005) | data below tp are excluded |
| **SAGE III** (v4.0) | 05/2002-12/2005 | surface/c.t. to 50 km | 1.5 km | Thomason et al. (2010) | only solar products used here |
| **POAM III** (v4.0) | 04/1998-12/2005 | 5-45 km | 1-2 km | Lumpe et al. (2006) Lucke et al. (1999) | |
| **SMR** SMR(2) (v2-0) SMR(1) (v2-1) | 07/2001-present 07/2001-present | 16-20 km 20-75 km | 3-4 km 3 km | Urban (2008) Urban et al. (2007, 2012) | 544 GHz-band 489 GHz-band |
| **MIPAS** MIPAS(1) (V3o_H2O_21) MIPAS(2) (V5r_H2O_224) | 03/2002-03/2004 01/2005-04/2012 | 6 km/c.t. to 70 km 6 km/c.t. to 70 km | 4-5 km 2-3.7 km | Milz et al. (2005) Milz et al. (2009) von Clarmann et al. (2009) | change in spectral resolution in 2004 |
| **SCIAMACHY** (v4.2) | 09/2002-04/2012 | 11-25 km | 3-5 km | Weigel et al. (2016) Weaver et al. (2019) | |
| **ACE-FTS** (v3.6) | 03/2004-present | 5 km/c.t. to 101 km | 3-4 km | Sheese et al. (2017) Lossow et al. (2019) | |
| **ACE-MAESTRO** (v31) | 03/2004-present | 5 km/c.t. to 20 km | 1-2 km | Sioris et al. (2010, 2016) Lossow et al. (2019) | |
| **Aura-MLS** (v4.2) | 08/2004-present | 316-100 hPa 100-0.2 hPa < 0.1 hPa | 2-3 km 3-4 km 6-11 km | Read et al. (2007) Lambert et al. (2007) Livesey et al. (2018) | |
| **HIRDLS** (v7.0) | 02/2005-03/2008 | 100-10 hPa | 1 km | Gille and Gray (2012) Lossow et al. (2019) | values high at p >100 hPa values low at p <40 hPa |
| **SAGE III/ISS** (v5.1) | 06/2017-present | 5 km/c.t.-100 km | 1.5 km | Davis et al. (2021) | |

**Table 11.** Time period, vertical range, vertical resolution, references and other comments for $CH_4$ measurements. Note, *tp* refers to tropopause and *c.t.* to cloud top in this table.

| Instrument (version) | Time Period | Vertical range | Vertical Resolution | References | Additional Comments |
|---|---|---|---|---|---|
| **HALOE** (v19) | 10/1991-11/2005 | tp up to 80 km | 3.5 km | Grooss and Russell (2005) | |
| **MIPAS** MIPAS-1 (v21) MIPAS-2 (v224) | 03/2002-03/2004 01/2005-04/2012 | 6 km/c.t. to 70 km 6 km/c.t. to 70 km | 4-5 km 2-3.7 km | Glatthor et al. (2005), von Clarmann et al. (2009) Plieninger et al. (2016) | change in spectral resolution in 2004 |
| **ACE-FTS** (v3.6) | 03/2004-present | 5 km/c.t. to 75 km | 3-4 km | Plieninger et al. (2016) Olsen et al. (2016) | |

**Table 12.** Time period, vertical range, vertical resolution, references and other comments for CO measurements. Note, *c.t.* to cloud top in this table.

| Instrument (version) | Time Period | Vertical range | Vertical Resolution | References | Additional Comments |
|---|---|---|---|---|---|
| **SMR** (v2) | 10/2003-09/2004 | ~17-110 km | 3-4 km | Dupuy et al. (2004) | |
| **MIPAS** MIPAS-1 (v20) MIPAS-2 (v220) | 03/2002-03/2004 01/2005-04/2012 | 6 km/c.t. to 70 km 6 km/c.t. to 70 km | 3.5-8 km 3.5-8 km | Funke et al. (2009) Funke et al. (2009) | change in spectral resolution in 2004 |
| **ACE-FTS** (v3.6) | 03/2004-present | 5 km/c.t. to 110 km | 3-4 km | Sheese et al. (2017) | |
| **Aura-MLS** (v4.2) | 08/2004-present | 215-0.1 hPa 0.1-0.005 hPa | 4-5 km (UTLS) 6 km (above) | Livesey et al. (2018) Pumphrey et al. (2007) | |

**Table 13.** Time period, vertical range, vertical resolution, references and other comments for NO measurements. Note, *c.t.* to cloud top in this table.

| Instrument | Time Period | Vertical range | Vertical Resolution | References | Additional Comments |
|---|---|---|---|---|---|
| **HALOE** v19 | 10/1991-11/2005 | 10-140 km | 3.5 km | Grooss and Russell (2005) | |
| **ACE-FTS** v3.6 | 03/2004-present | 6 km/c.t.-107 km | 3-4 km | Sheese et al. (2016) | |
| **SMR** v2.1 | 10/2003-present | 30-60 km 80-110 km | 4-6 km 6-8 km | Sheese et al. (2013) | only 1 day per month prior to 04/2007 |
| **MIPAS** MIPAS v20 MIPAS V220 | 03/2002-03/2004 01/2005-04/2012 | 12-70 km | 3.5-5 km 2.5-6 km | Funke et al., 2005b Funke et al., 2014 | change in spectral resolution in 2004 |

**Table 14.** Time period, vertical range, vertical resolution, references and other comments for $NO_2$ measurements. Note, *c.t.* to cloud top in this table.

| Instrument | Time Period | Vertical range | Vertical Resolution | References | Additional Comments |
|---|---|---|---|---|---|
| **LIMS** v6.0 | 11/1978-05/1979 | 10-50 km plus mesosphere for polar night | 3.7 km | Remsberg et al. (2010; 2021) | |
| **SAGE II** v7.0 | 10/1984-08/2005 | surface/c.t. to 50 km | 1.5 km ($<$38 km) 5 km ($>$38 km) | Cunnold et al. (1991) | only SS data used |
| **HALOE** v19 | 10/1991-11/2005 | 10-50 km | 2.5 km | Grooss and Russell (2005) | |
| **POAM II** v6.0 | 10/1993-11/1996 | 20-40 km | 1.5-2.5 km | Lumpe et al. (2002) Randall et al. (2002) | |
| **POAM III** v4.0 | 04/1998-12/2005 | 20-40 km | 1.5-2.5 km | Lumpe et al. (2002) Randall et al. (2002) | |
| **OSIRIS** v3-0 | 10/2001-present | 13-45 km | 2 km | Brohede et al. (2007) | |
| **SAGE III** v4.0 | 05/2002-12/2005 | surface/c.t. to 50 km | 1.5 km | Mauldin et al. (1998) | only SO data used |
| **MIPAS** MIPAS V20 MIPAS V220 | 03/2002-03/2004 01/2005-04/2012 | 12-50 (12-70) km for day (night) measurements | 3-6 km 2.5-6 km | Funke et al. (2005) Funke et al. (2014) | change in spectral resolution in 2004 |
| **GOMOS** v6.01 | 03/2002-04/2012 | 20-50 km | 4 km | Kyrölä et al. (2010) | |
| **SCIAMACHY** v4-0 | 09/2002-04/2012 | 9 km/c.t.-48 km | 3-5 km | Bauer et al. (2012) | reference valid for v3.1 |
| **ACE-FTS** v3.6 | 03/2004-present | 7 km/c.t.-52 km | 3-4 km | Sheese et al. (2016) | |
| **HIRDLS** v7.0 | 01/2005-01/2008 | 20-50 km | 1 km | Gille and Gray (2011) Belmont Rivas et al. (2014) | |

**Table 15.** Time period, vertical range, vertical resolution, references and other comments for $HNO_3$ measurements.

| Instrument | Time Period | Vertical range | Vertical Resolution | References | Additional Comments |
|---|---|---|---|---|---|
| **LIMS** v6.0 | 11/1978-05/1979 | 10 km/c.t.-50 km | 3.7 km | Remsberg et al. (2010; 2021) | original vertical resolution is 2 km but adjusted to make compatible with lower resolution LIMS products |
| **UARS-MLS** v6 | 10/1991-10/1999 | 100-4.6 hPa | 5-10 km | Livesey et al. (2003) | data with significant (1-3 ppbv) low bias at p <15 hPa and high bias below the VMR peak |
| **SMR** v2.0 | 07/2001-present | 18-45 km | 1.5-2 km | Urban et al. (2006) Urban et al. (2009) | empirical scaling applied |
| **MIPAS** MIPAS V22 MIPAS V224 | 03/2002-03/2004 01/2005-04/2012 | 6 km/c.t. to 70 km | 4-6 km 3-5 km | Mengistu Tsidu et al. (2005) Wang et al. (2007) von Clarmann et al. (2009) | change in spectral resolution in 2004 |
| **ACE-FTS** v3.6 | 03/2004-present | 5 km/c.t.-62 km | 3-4 km | Sheese et al. (2016) | |
| **Aura-MLS** v4.2 | 08/2004-present | 215-1.5 hPa | 3-5 km | Santee et al. (2007) Livesey et al. (2018) Fiorucci et al. (2013) | |
| **HIRDLS** v7.0 | 01/2005-01/2008 | 215 - 5.1 hPa | 1 km | Gille and Gray (2012) | latitude range 63°S-80°N |
| **SMILES** v2.0.1 | 10/2009-04/2010 | 18-45 km | 3-4 km | Kreyling et al. (2013) | bias due to problems in spectroscopic parameter and altitude shift |

**Table 16.** Time period, vertical range, vertical resolution, references and other comments for $HO_2$ measurements.

| Instrument | Time Period | Vertical range | Vertical Resolution | References | Additional Comments |
|---|---|---|---|---|---|
| **SMR** v2 | 10/2003-10/2004 | 30-60 km (10-0.3 hPa) | 3-4 km | Khosravi et al. (2013) | |
| **Aura-MLS** v3.3 | 07/2004-present | 22-0.0046 hPa | 4-10 km | Pickett et al. (2008) Khosravi et al. (2013) | daytime fields with night-time mean as background correction |
| **SMILES** v2.0.1 | 10/2009-04/2010 | 26-95 km (20-0.001hPa) | 4-5 km | Kreyling et al. (2013) Khosravi et al. (2013) Kuribayashi et al. (2013) | |