# Peer review of "Overview and update of the SPARC Data Initiative: Comparison of stratospheric composition measurements from satellite limb sounders"

_Earth System Science Data, 2020_

## Referee Comment (RC1) · Sean Davis (Referee) · 23 Dec 2020

This paper is an update to an ambitious effort to assess currently available satellite limb measurements of stratospheric trace gases. The authors are to be applauded for their comprehensive assessment of a number of different data sets produced by different institutions and spanning multiple decades. It is encouraging to see that for several of the species, the use of updated retrievals results in better agreement than in the earlier version of the SPARC DI data set. This data set will no doubt be useful to the observational and modeling communities for studies of stratospheric composition. I

have only a few minor comments and recommendations before it is accepted to ESSD.

Page 3, lines 15-20 – This data also contributed to several of the S-RIP chapters/papers, and I think that is worth mentioning here somewhere.

Page 9, Lines 1-3 – As I understand it there are multiple MIPAS retrievals from different groups. Could the authors please provide some justification for why they choose the IMK retrieval, and/or provide any information and references concerning known differences between the retrievals?

Page 5, line 22 – the reference to appendix table A4 seems quite out of order. Additionally, I don't understand the distinction between the figures and tables in the "appendix" versus the main text. Content-wise, it seems like the material in the appendix belongs in the paper itself and is not really an appendix.

Page 10, section 2.14 – It would be helpful if the authors mentioned the end date and reason for the end of HIRDLS data.

Page 11, section 2.16 – It looks like the authors are using two different versions of OMPS (based on table A5). Which is the primary one they are considering? Reference to/discussion of the version they are using here would be helpful. Also, I believe there is yet another OMPS-LP retrieval that is not included here (Kramarova et al., 2014). As with the MIPAS discussion it would be helpful to have some insight into the choices the authors have made and justifications for excluding certain products, and what the known major differences are between the retrievals.

Kramarova, N. A., Nash, E. R., Newman, P. A., Bhartia, P. K., McPeters, R. D., Rault, D. F., Seftor, C. J., Xu, P. Q. and Labow, G. J.: Measuring the Antarctic ozone hole with the new Ozone Mapping and Profiler Suite (OMPS), Atmospheric Chemistry and Physics, 14(5), 2353–2361, doi:10.5194/acp-14-2353-2014, 2014.

Page 11, lines 26-28 – I think the term "climatology" is a confusing term to use to describe this data set. As the authors acknowledge here, a climatology typically refers

to some long term mean state. But in this paper, "climatology" is being used to describe a time series. The authors also use the term "climatology" (e.g., "climatological approach") as a stand in for "gridded data set" when contrasting their approach to profile-to-profile coincident comparisons (e.g., sentence starting line 28). I also find this terminology confusing. The data set the authors have produced is a gridded time series data set, and I think it is more accurate to describe it as such.

Page 11, starting line 28 – It seems as though one of the main advantages of the approach used here (comparing gridded data sets) is that all data from each sensor are used in the comparison, as opposed to profile-profile comparisons where some profiles simply don't meet the chosen coincidence criteria. I believe this is the reduction in random error the authors are referring to here. However, this benefit must be weighed against the sampling bias (e.g., as addressed in Toohey et al 2013) that is introduced when one grids data. It's not totally obvious how these two factors compete, and some acknowledgement of this balance would be appreciated.

Page 12, line 16 – What do the authors mean by hybrid log-linear here? Do you mean interpolating the log VMR linearly in altitude, or interpolating the VMR linearly in log pressure? I'm guessing the latter, but please clarify.

Page 12 lines 16-20 – It appears as though the authors are using the most convenient method for converting to VMR on a pressure grid for each individual data set. I don't mean to belittle this approach because it would be a rather Herculean task to use a common data source for all the different instruments. And even then some of the retrievals may use p/T in their retrieval "upstream" of what is available to the public. Nevertheless, I think it is important to recognize that this grid conversion using different ancillary data as a possible source of uncertainty. I am not aware of any work that has attempted to quantify this source of uncertainty, but any additional discussion or references related to this issue would be very helpful.

Page 14, paragraph line 17 – 22 – This paragraph doesn't make any sense and should

probably be removed. It is addressing some evaluation that is not shown in the paper, and doesn't really even explain what the result is from this evaluation.

Page 15, line 25 – spectroscopical -> spectroscopic

Page 15, line 30 – considerable -> considerably

Page 16, line 22 – The Wang et al paper is now published

Page 16, lines 22-24 – The altitude registration problem is easily corrected, as outlined in the appendix of Wang et al 2020. The authors should implement this correction.

Page 17, line 31 – "also slightly" -> "also has slightly"

Page 21, line 25 – "mechanism" -> "mechanisms"

Page 22, line 5 – I think you mean "time" here instead of "date"

Page 24, line 19 – this paragraph ends abruptly. Can you say something about how this compares to SPARC 2017, as is done for the other species?

Page 23, line 23-25 – This is a run on sentence.

Table A5 – as previously mentioned, Wang et al. paper has been published now.

Table A6 – should cite Davis et al. for the SAGE III/ISS water vapor

Davis, S. M., Damadeo, R., Flittner, D., Rosenlof, K. H., Park, M., Randel, W. J., et al. (2020). Validation of SAGE III/ISS solar water vapor data with correlative satellite and balloon‐borne measurements. Journal of Geophysical Research: Atmospheres, 125, e2020JD033803. https://doi.org/10.1029/2020JD033803

Data versions questions:

In general it is preferable to use the newest data set from each satellite. There is a new Aura MLS version 5 data set, which I assume will become the widely adopted version of the data to use. Could this be included in the data set? Similarly, there is a

new ACE-FTS version (4.1) that is the recommended version. Also, which version of MAESTRO data is being used here? It says "31" in the table, which I assume refers to v3.1. But there are several sub-versions of 3.1 (eg, 3.11, 3.12, ...). The latest version is 3.13 – is that what is being used?

———————————————

---

## Referee Comment (RC2) · Anonymous Referee #2 · 12 Jan 2021

**Review of**
**"Overview and update of the SPARC Data Initiative: Comparison of stratospheric composition measurements from satellite limb sounders"**
**by M. Hegglin et al.**

This manuscript presents an overview an update of the SPARC Data Initiative, in particular of the assessment of all satellite limb measurements of stratospheric trace gases. The data initiative is a really important and ambitious project. It aims to evaluate the state of the available datasets comparing them against the multi-instrument mean to gain an idea of the usefulness of such datasets.

The methodology used is to compare zonal means to identify biases among the datasets. My main concern with this manuscript is that such approach is too simplistic as here described, the drawbacks of such comparisons are mentioned in the text (i.e., the possible sampling biases and the geophysical variability) but not really incorporated into the analysis.

For example, in Toohey et al (2013) some instruments can potentially have ozone sampling induced biases greater than 10% (or even 20% in some locations -see Figure 3 of Toohey paper). Yet the authors define excellent agreement as 2.5% through-out the entire zonal mean disregarding the location of such biases, I encourage the authors to consider the value of the sampling biases and to define the agreement criteria accordingly. So, for ozone below 100hPa and near the poles excellent agreement could be defined as between 10% due to the expected sampling biases.

Similarly, when comparing "cross-sections" (as in Figure 5) for different time periods, each "climatology" will be affected by different variability. For example, the figure below shows "climatologies" constructed using MLS data for different time periods.

[Figure]

 As can be seen, some differences that will be interpreted as bias in the current analysis, are just natural variability (values up to 5 to 10% in the ozone example above). The methodology should identify areas of high variability and change the criteria agreement for such areas. They could also identify how many months are needed to decrease such variability within their current excellent agreement criteria and only include "climatologies" on such scales.

I understand that including the sampling biases and the variability into the analysis may be over-ambitious for the already ambitious project. But at least a more thorough discussion of the caveats of the agreement criteria is needed.

Comments:

P2 line 20:  the vertical resolution is not 1 to 4km according to the satellite instruments section.
P2 line 28:  There should be more recent citations for stratospheric dynamics and transport.

P3 line 4:  Delete the in-preparation paper.

P2 line 28: Space missing between N2O5 and HNO4.

P3 line 29: add "and" between CH2O and CH3CN

Figure 2 is hard to read. Consider getting rid of the alternating blues background and filling the cells using solid colors instead of Xs. For example

| SPARC data initiative | Mol1 | Mol2 | Mol3 | Mol4 | Mol5 | Mol6 | Mol7 | Mol9 | Mol10 |
|---|---|---|---|---|---|---|---|---|---|
| Inst1 | 🟩 | 🟩 | 🟩 | 🟩 | 🟩 |  |  | 🟩 | 🟩 |
| Inst2 | 🟩 | 🟩 |  |  | 🟩 |  |  |  |  |
| Inst3 | 🟦 | 🟦 | 🟦 |  | 🟦 | 🟦 | 🟦 | 🟦 |  |
| Inst4 | 🟧 | 🟧 |  |  |  |  |  |  |  |

P4 line 14: It is not clear to me what is meant by random sampling error? The sampling error is anything but random is determined by the sampling and the variability of the measured parameter. Are the "highly structured and transient features that may not be resolved by some instruments" refer on P4 line 25 what is meant by random sampling error?   Please clarify.

Section 2: This entire section is inconsistent. Some instrument subsections have information about its vertical resolution, some have information about the spectral range and spectral resolution, some have information about which retrieval they are using, some explain the measurement concept (i.e. GOMOS), some have FOV information, or tangent heights. Please be consistent, if you feel the need to explain on detail for an instrument, then explain it for the other, regardless if it is given on the tables.

P5 line 22:  Please add emission after nadir. They are other type of measurements using the nadir view.

Figure 3:  What is wrong with SCIAMACHY and OMPS around 45W and 60S? Is this related with the South Atlantic Anomaly, if those please mention it here or in the SCIAMACHY / OMPS sections.  This figure is also missing the sampling patterns of LIMS, MAESTRO and SAGEI.   If MAESTRO sampling is

missing because is the same as ACE-FTS, please make sure to specify that in the ACE-FTS subplot title or in the caption.

P6 line 17: delete space before 70S

P7 line 2: "was about 3 to 5km" contradicts the statement made in the introduction "1 to 4 km"

P7 line 15: if POAMII and POAMIII tracks are identical why show them on Figure 3.   (Just curious what happen to POAM1)

Section 2.6. I thought there were different retrieval versions for this instrument. Which one are you using and why?

P8 line 14: This is the same principle of sun occultation which was not explained previously. Why explain it here? or better yet Why not explain the sun occultation and other methods the first time they are introduced...

P8 line18: altitude resolution of GOMOS is 0.5 to 1.7 versus 3 to 5 UARS MLS are you should you do need to use the averaging kernels. Could you prove that the comparisons do not improve when comparing so dissimilar resolutions?

P8 line 26: I thought it was 0.025 cm-1   (see for example https://amt.copernicus.org/articles/2/337/2009/amt-2-337-2009.pdf )

P9 line 3: Please justify why are you using this retrieval and not any of the many others…

P9 line 23: "the latitude scanning (see figure 3) is the same each year".  That is not true, it changes slowly, the pattern may be the same but not the time of the measurements,  see below:

[Figure]

P9 line 25: Methodology discussions should be moved to section 3

P10 line 5:  Again, I am assuming that their sampling patterns are identical. In that case, Figure 3 should be labeled differently so that the reader know that their sampling pattern is the same..

P10 line 8 Methodology discussions should be moved to section 3

P10 line22: The instrument was not damaged. Please change to: Unfortunately, during launch, most of the aperture was obstructed by a plastic film used for insulation that became detached during the ascent to orbit.    (or something similar) https://doi.org/10.1117/12.623574

P10 line 24   Coverage is 65S – 82N    https://doi.org/10.1029/2007JD008824

P11 line 11: Coverage is between 38S-65N in north looking days and 65S-38N on south looking days so, as shown in figure 3, I could be argued that the coverage is actually 65S-65N.  What does nominal mean in this instance?

Section 2.17: I do not believe TES is used at all in the manuscript. Please delete this section or include examples using TES data.

P11 line 18: the apodized or unapodized detail was not mention for MIPAS.

P12 line10: The data preparation and handling of the dataset is a really important step of the whole endeavor. Please provide a brief description here, or in each instrument section. Also,  the SPARC data initiative does not have information on the new datasets included in this manuscript.

P12 line19: The conversion from altitude to pressure levels as well as the conversion from number density to volume mixing ratio will introduce an uncertainty. According to  10.5194/amt-9-2497-2016 this bias could be up to 5% in the upper stratosphere (Figure 8/section 6 of that paper).

P12 line16: is not clear to me what is meant by hybrid log linear interpolation, do you mean that you linearly interpolated on log(pre) instead of on pre. A brief explanation will suffice.

The terminology on Table 3 is barely used through-out the rest of the text. I recommend changing the blue-red color bar of Figures 5, 6, 8 etc, to a blue-white-red color bar with the white covering from -2.5 to 2.5. That way the reader could easily identify where the excellent agreement is found.

P12 line 24: The authors seem to use climatology file when they meant zonal monthly mean files. It is confusing.

Figure 4: Again not climatology file, zonal monthly mean file, space missing between and-LST_MEAN. MIPAS was not available in 2018. Do you mean 2008?

P12 line 29: Toohey et al 2013 is not the only study about sampling biases for limb instruments, there are more, for example: 10.5194/amt-7-1891-2014,  10.5194/acp-16-11521-2016, 10.5194/acp-18-4187-2018, 10.5194/amt-12-2129-2019

P13 line 26: How do the authors define if measurements from a given instrument are deemed unrealistic. I think the MIM will be more robust if the authors defined quantitively which measurements to include or exclude.

P14 line 17: Since the second summary is not going to show here, please delete this paragraph.

P16 Line15: Why are the author evaluating different retrievals for OMPS? If they are going to do that, they should include all the retrievals from MIPAS and OSIRIS, etc.

P16 line 23: Wang et al is published already. And it shows how to get rid of the altitude registration problem. Such correction should be implemented for SAGEIII/ISS. As well as for SAGEII since that product is also affected.

P17 line 27: I thought kernels were not considered at all through the whole study. Further I do not see what retrieving in log space has to do with the kernels. The averaging kernels are concentration dependent disregarding if they are retrieved in log space or not. Retrieving in log space only ensures that the retrieval will be positive.

P19 line 1: instead of the current figure 5 why did the authors do not show the figure 5 and 6 equivalent for CH4. It will presumably show the same results as Figure 7 but I will be consistent with the previous to sections.

Figure 7 what is the shaded region is this the excellent agreement region or the very good please specify in the caption.

Figure 8: the color bar units are missing. Is the level in the differences really 2 or 2.5.

P20 line11:  Not "climatologies", zonal monthly mean from SMR, MIPAS, …

P20 line18: Again, be consistent in Figure 5 and 6 you only show the MIM, not the individual instruments zonal means. Now, looking at those, why are the authors including SMR into the MIM when there is a clear high bias around 20hPa in the tropics and a clear artifact around 60N.

P21L16:  Were some instruments not included for O3, H2O in figure 14 due to differences in time period, Why is this important for CO but not for H2O, O3, etc?

P22 line 3: Why was ozone at higher altitudes not corrected using a chemical box model? There is a strong diurnal variation in the US-LM affecting those concentrations.

P22 line 25: Why does figure 11 does not include the MIM?  Also, this figure should include panels showing the difference versus the MIM to be consistent.   How are the authors evaluating the good agreement (P22 line27) or the agrees well (P22 line31) or the very good agreement without it.

What does the "s" stands for in ACE-FTS s10am etc?

Figure 12: What is the "s" or the "ss"

P24 line 1: sorry if this is obvious and I am just not understanding it.  How come if you compute anomalies as monthly – MYM  (multy year mean) might display a diurnal cycle while doing (monthly – MYM) / MYM *100.   does not?

P24 line 11: "possibly due to its higher vertical resolution"  This statement could be proven by applying for example the MIPAS averaging kernels to HIRDLS and repeating the comparison.

P24 line 17: Again, please be consistent, in previous sections there has always been mention of the previous evaluation.

P24 line32 SMR data is not used at all in Figure 13, i.e., the only comparison shown in the manuscript.

P25 line 1: please include the equivalent of Figure 5 and 6 for HO2. Showing the MIM and the difference versus the MIM for SMILES, MLS and SMR before discussing the November 2009 and February comparison.

Figure 13: are you comparing the MIM for 2 instruments. If you are, then just shown the MIM and then the differences.   Were the SMILES selected according to their local time or all local times were included in this comparison?

P25 Summary evaluations. Is the reader supposed to be using the definitions on Table 3 with Figure 14, 15 and 16? If it is, the color bar should reflect the values determined in such table 3.

P26 line1: Why is there no HO2 1-sigma multi-instrument summary as for the other molecules? There are other molecules with only 2 or 3 instruments as for SF6, HF, NO, HCl.

Why is there no BrO 1-sigma summary?

Figure 15 is not consistent with Figure 14, I understand that the authors are redoing the Figure from the SPARC DI but for this paper it will be much better is they kept a simple layout as in Figure 14. That is, remove the boxes and the chemistry explanation, etc.

P26 line 11 The acronyms (CSA, ESA, JAXA, etc) are not are not defined.

P26 line 16  These manuscript only updated the trace evaluation, as written it implies that it also evaluated the aerosols observations.

P26 line 26:  The doi zenovo link those not exist. As in, google cannot find it (neither the zenovo webpage). You have to go into the webpage and search for Hegglin to get the dataset. Please clarify

P27 line 5:  Why generally? It always produces larger sample sizes.

P27 line 9: Toohey et al 2013 only shows the sampling biases for Ozone and water vapor. As mention here it seems that it investigated the sampling biases for all molecules included in the SPARC data initiative.

Table 1- References column: Why are some years in brackets and some don't?  Cloud top is not an appropriate vertical range, please provide the lowest possible measured altitude in clear sky and then either on the text or as a tablenote specify that in the presence of clouds is from the cloud top.   Why do HALOE, UARS and SMR have pressure ranges as well as altitude ranges. Please be consistent.

Table 2: Why does MIPAS have two vertical resolutions? Due to the change in spectral resolution? Please clarify.

What happened to the ascending LT of meas for SCIAMACHY?

How come TES has a LT of meas but not MLS and HIRDLS.

For SAGEIII/ISS - Wang et al has already been published.

Table A1  I think this should be part of the main text. Also, I think the variable type is wrong:

N2O, N2O_NR and N2O_STD are 3d, [lat, pre, months]

AVE-dom, ave-lat, lst's are 2D. [lat, months]

But all are described as GEO2D.

Further, looking at some files, for example the OMPS-SASK or OSIRIS have other variables:
OZONE_CONCENTRATION_STANDARD_ERROR,  OZONE_VMR_STANDARD_ERROR
OZONE_COUNT_IN_BIN

Table A2. Why do some molecules used different versions? For example, MIPAS (1) uses v21 v20
Table A3. Why is v4.0 in SAGEIII/M3M in brackets?
Table A4. Occultation needs to be bold in the Stellar or lunar occultation section.
Table A5. LIMS reference, is there no published reference. This dataset is from 1979?  Perhaps this was meant to be Remsberg 2009?
MIPAS additional comments:   Meas mode switched in 2004 from high spectral to low spectral resolution.   Not from high to high
SAGEIII/ISS reference: Wang et al has been published.
HALOE: the vertical range states up to 80km, please provide the minimum altitude

Table A6. MIPAS: Meas mode switched in 2004 from high spectral to low spectral resolution.
SAGEIII/ISS cite 10.1029/2020JD033803.
Is there an upper vertical range limit for UARS MLS ?

Table A7 The additional comment for MIPAS in the previous tables is that the Meas mode switched in 2004 from high to low …, in here it says Change in spectral resolution in 2005. Please be consistent. And also the years do not change.

Table A8 should include https://doi.org/10.1029/2007JD008723  for MLS.

Table A9:  In the previous tables the authors have given the overall period for MIPAS. I Actually prefer the format in A9. But be consistent.

Table A10. No references for SAGEIII and SCIAMACHY, at least provide the one describing the instrument.
The day/night in MIPAS imply that you are taking the day night difference? If it those, this detail should be in the additional comments.

Table A11:  I do not understand the additional comment for UARS MLS, please clarify
Please use the vertical range MIPAS layout for all cloud top's in these tables.

Table A12: Why is there a (UTLS) in the AURA MLS row. The resolution according to the v3 quality document varies from 4 at 10hPa to 10 at 0.046 hPa.  Please clarify.

---

## Referee Comment (RC3) · Anonymous Referee #3 · 19 Jan 2021

This manuscript describes the results of the SPARC initiative to identify and compare nearly all existing remote sensing data of the atmosphere from limb sounders, along with updates since the initial report. This was a long and careful effort, and the results are extremely impressive. Anyone using limb sounding measurements in their work (or interested in the chemical composition of the stratosphere and mesosphere) will benefit from this project - to identify data availability, compare different instrumental data sets, or just to understand composition and chemistry in the region from about 200 hPa to 100 km altitude. The authors used a "top-down" approach, in which all

measurements are averaged into altitude-latitude bins and compared. For a project of this scope and completeness, this is probably the only feasible way to accomplish the goals and show the results in a finite amount of space. It is also a nice complement to more traditional methods of evaluating remote sensing data by comparing coincident profiles or measurements. Only some of the highlights are included here, but all data are available on a separate website. For many readers, the multi-instrument mean (MIM) will be most useful, but there are also carefully analyzed data on differences between each data set and the MIM, as well as details about sampling, and a brief description of the general level of understanding about each constituent. With a few minor revisions, this paper will be an excellent contribution to the literature.

P.17, SAGE II improvements - I am always cautious when data sets are reanalyzed to show better agreement. It is natural for the reanalysis to more closely approach the consensus value, but that is not necessarily the "true" value. The explanation is quite reasonable though; no changes needed.

Section 4.5 and following - Peroxyacetyl nitrate (PAN) is not mentioned in the paper, but it does contribute to NOy at the lower end of the altitude range shown (Figure 15). From in situ data from the NASA ATom mission, the fraction of PAN to NOy can be 1/3 to 1/2 in the tropics near 200 mbar. In the stratosphere (higher latitudes at 200 mbar), PAN is usually 10% or less. But it can still be a measurable fraction of NOy in the midlatitudes, probably more likely in the summer when the tropopause is higher. This will likely only affect the very bottom of the NOy plots, but it should probably be at least mentioned briefly in the text.

P.27, l.7 Besides the non-uniformity of sampling, another factor can be the long-term trends in various gases. For example, CFC11 and 12 reached a peak and are now decreasing. But for a gas like water, the long-term trend is not so obvious (and quite important for climate forcing). The "true" (or measured) MIM could be changing over time. This is beyond the scope of this paper, but could be (briefly) mentioned as a possibility in the section on H2O. My understanding of the MIM here is that it is the
average over the duration of the measurements, and does not change with time. If that is not correct, then I missed it in the text.

Some further, more specific comments:

P.1, l.8 It is difficult to separate "long-lived trace gases" and "transport tracers". There is considerable overlap. Just an observation; no changes needed in the text.

P.2, l.2 "nitrogens"? Maybe "nitrogen-containing species"

l.7 add comma after "climatologies" (if "which" refers back to "approach").

l.12 "intended summary"

P.3, l. 25, I was initially confused why TES did not appear here, but was on the list on P.2, l.6. It is explained on P.5, and is fine; no changes needed.

l.27 At some point NOx and NOy should probably be defined, although almost everyone knows what they are. I was curious how NOy would be handled, since it's hard to measure all components of NOy with limb sounding. Defining them later in the paper is OK; see comment above about components of NOy in the troposphere.

l.33, "data are"

P.4, l.29 "its" instead of "the above" (since I don't see any disadvantages listed above)

P.5, l.4, "JGR - Atmospheres"

P.6, l.5 vs. l.17, dates include S3/ISS or not?

P.8, l.12, "98.55 S"? Maybe drop the "S"?

l.22, "on board Envisat"

P.10, l.29 perhaps something like "demonstration of ultrasensitive sub-mm limb emission observations..." And I don't think 4 K should be hyphenated.

P.13, l.8, not sure why "roughly-uniform" is hyphenated. Or why "generally" and

"roughly" are both used together. Perhaps "is roughly uniform with respect to longitude".

l.17 "can be found"

P.14,l.17 Where can this second summary be found? In SPARC 2017?

P.15,l.25 "spectroscopic"

l.30, I don't think that MS and US have been defined.

P.16,l.28, What is the "LM"? OK, it's all in Table 4 (I missed the reference to it on P.12, l.7). But would it be worthwhile to include the stratopause in a figure somewhere too (like the top panel of Figure 4, where it won't get in the way too much)? Maybe not. It was not obvious to me whether the boundary between the US and LM is simply taken as an altitude or pressure level, or whether it has latitudinal structure (or varies with month). Not an important point, and I may have missed a description of this - fine as long as it is clear to an interested reader. (If needed, you could probably put any explanations in the Table 4 caption.)

l.33, "carbon dioxide and other anthropogenically emitted greenhouse gases" or something like that. And maybe combine the next two sentences as "H2O is also a key constituent in atmospheric chemistry as a source gas of the hydroxyl radical..."

P.18, l.15 Does USLM mean "upper stratosphere and lower mesosphere"?

P.19, l.27 Need to edit this sentence - something like "a somewhat patchier difference field, however, it provides supporting evidence..." or "a somewhat patchier difference field, however, which provides supporting evidence..." or "a somewhat patchier difference field, however, providing supporting evidence..." I'm not sure I follow the logic in this sentence.

l.33 I assume that "overlap year" describes when all three instruments were reporting data. Also, "confirms the results described here"

P.20, l.1 I would not hyphenate "mean state".

P.21, l.16 "for which they provided data."

l.18 OK, here are the definitions of NOy and NOx (fine with me). In the mid-troposphere, peroxyacetyl nitrate (PAN) is one of the major components of NOy. This is mostly below the altitudes shown here, but see comment above.

l.25 "mechanisms"

l.29 "conversion" instead of "exchange"?

l.32 "is taken into account"

P.22, l.25, "latitude bands"?

P.23, l.8, "consistent with" rather than "confirming"?

P.58 I did not notice the reference to table 3 while reading the text. That is a fine way to make all the descriptions quantitative.

Table A2, perhaps "added recently" or at least "newly added"

Figure 8 - missing units on the color bars. I am guessing they should be ppm and percent, but the lower one could be ppb. Also, some of the subscript "4" look a little too small and too close to the "CH" (like for HALOE).

Figure 10 - x-axis labels are a little confusing. Is there room to write "Jan 2006" etc.? If not, then having the year under the month should work. Similar comment for Figure 12.

Figure A1 - About the units on the color bars: In the middle figures, do the grid boxes represent the number of samples per year, and in the lower figures, the boxes represent the number of samples per month in each latitude bin for each month? I suppose so, but maybe that can be included in the figure (like in the color bar) or in the caption.

[Figure]

---

## Author Comment (AC1) · 15 Mar 2021

This manuscript presents an overview an update of the SPARC Data Initiative, in particular of the assessment of all satellite limb measurements of stratospheric trace gases. The data initiative is a really important and ambitious project. It aims to evaluate the state of the available datasets comparing them against the multi-instrument mean to gain an idea of the usefulness of such datasets.

Response: We thank the reviewer for his/her careful reading of the manuscript and

very helpful comments which helped improve our manuscript. Please find below our answers in blue.

The methodology used is to compare zonal means to identify biases among the datasets. My main concern with this manuscript is that such approach is too simplistic as here described, the drawbacks of such comparisons are mentioned in the text (i.e., the possible sampling biases and the geophysical variability) but not really incorporated into the analysis.

Response: We have shown in SPARC (2017), Hegglin et al. (2013) and Tegtmeier et al. (2013), that the results on biases derived from our approach are generally robust across evaluations based on evaluations using shorter or longer time periods. We have also compared our results to results obtained from validation approaches using coincident measurements and generally, the bias estimates are too, very similar. These comparisons have provided confidence that our approach is not too simplistic as implied by the reviewer. There are also a lot of advantages of our approach, for example that we get information on biases that are latitude-resolved, which is not usually provided in the coincident-validation approach. In particular, our evaluations provide the reader/user with a clear sense of whether an instrument's data fields should be used (also in a climatological sense) to evaluate models. Where an instrument deviates strongly from MIM (e.g., in polar region), one should not use it for zonal monthly mean evaluations (whether due to sampling bias or systematic measurement bias).

For example, in Toohey et al (2013) some instruments can potentially have ozone sampling induced biases greater than 10% (or even 20% in some locations -see Figure 3 of Toohey paper). Yet the authors define excellent agreement as 2.5% through-out the entire zonal mean disregarding the location of such biases, I encourage the authors to consider the value of the sampling biases and to define the agreement criteria accordingly.

Response: See below, we prefer to keep the same approach as in the already published literature on the SPARC Data Initiative, so to keep the results/numbers comparable.

So, for ozone below 100hPa and near the poles excellent agreement could be defined as between 10% due to the expected sampling biases.

Similarly, when comparing “cross-sections” (as in Figure 5) for different time periods, each “climatology” will be affected by different variability. For example, the figure below shows “climatologies” constructed using MLS data for different time periods.

As can be seen, some differences that will be interpreted as bias in the current analysis, are just natural variability (values up to 5 to 10% in the ozone example above). The methodology should identify areas of high variability and change the criteria agreement for such areas. They could also identify how many months are needed to decrease such variability within their current excellent agreement criteria and only include “climatologies” on such scales.

Response: See below for further discussion. We are fully aware of the possibility that natural variability may have an advers impact on our comparison methodology. However, the 5% to 10% sampling biases identified by the reviewer can be readily minimised by averaging over a longer time period that covers several QBO-cycles than that chosen in the above figure by the reviewer.

I understand that including the sampling biases and the variability into the analysis may be over-ambitious for the already ambitious project. But at least a more thorough discussion of the caveats of the agreement criteria is needed.

Response: We indeed opt to add a more detailed discussion of the above shortcoming of our methodology in the text as suggested by the reviewer (see new section 3.2.1), since a detailed evaluation of the influence of sampling biases is indeed beyond the scope of this manuscript. The reviewer is correct that natural variability has to be taken into account in the interpretation of the derived inter-instrument differences. We

would like to stress that we have taken account of sampling issues in several ways when performing our evaluations and also were following procedural precautions to be confident of the validity of our approach:

- We include as many years as possible with the maximum years of overlap in one evaluation. Where overlap was not possible, as many years as possible would be chosen keeping in mind a potential tradeoff with underlying trends. For most species, trends can be considered much smaller than inter-instrument differences.

- Where the instruments' temporal coverage allowed for it, we tested inter-instrument differences for different time periods to get a sense of the influence of temporal inconsistencies in the comparison. The general structure in the different instruments' biases relative to another thereby generally did not change.

- There are some examples where the previous conclusion was not applicable. When looking at SAGE II versus HALOE differences we found that the inter-instrument differences changed over time, which in this case was indicative of a drift in one of the instruments. The change was not attributable to sampling biases since the instruments were compared for over the same time periods. Please see SPARC report (2017) for a detailed discussion.

- Where possible, we also used multi-annual averages over time periods longer than 3-4 years so that obvious structures in tracer distributions from the QBO (such as seen in the reviewer's example when looking at the 2009-2012 versus 2015-2016 time periods) would cancel out.

Finally, while we agree with the reviewer's suggestion in principle, we would like to retain our definition of the different levels of agreement, mainly because we would like to remain comparable to and consistent with the SPARC report (2017). A further argument may be that the definition of levels of agreement is to some extent arbitrary anyway and would likely differ when defined by a data producer (who understands how difficult it is to measure in the lowermost stratosphere) versus a data user (who

needs a certain level of accuracy to be able to make a clear inference for, e.g., model performance).

Comments:

P2 line 20: the vertical resolution is not 1 to 4km according to the satellite instruments section.

Response: We now say approximately 1 to 5 km, since this range depends on the species and spans all vertical resolutions given in Table A5 (submitted manuscript) except the highest (SAGE III/ISS with 0.75 km) and lowest resolutions (UARS-MLS with 8 km) indicated.

P2 line 28: There should be more recent citations for stratospheric dynamics and transport.

Response: We added a few references and now add 'e.g.' not to imply this list to be exhaustive.

P3 line 4: Delete the in-preparation paper.

Response: Deleted.

P2 line 28: Space missing between N2O5 and HNO4.

Response: Changed.

P3 line 29: add "and" between CH2O and CH3CN

Response: Added.

Figure 2 is hard to read. Consider getting rid of the alternating blues background and filling the cells using solid colors instead of Xs.

Response: Thank you for the suggestion. We have changed the figure and hope it is now better readable.

P4 line 14: It is not clear to me what is meant by random sampling error? The sampling error is anything but random is determined by the sampling and the variability of the measured parameter. Are the “highly structured and transient features that may not be resolved by some instruments” refer on P4 line 25 what is meant by random sampling error? Please clarify.

Response: Thank you for spotting this error. We meant to say random measurement error. Corrected.

Section 2: This entire section is inconsistent. Some instrument subsections have information about its vertical resolution, some have information about the spectral range and spectral resolution, some have information about which retrieval they are using, some explain the measurement concept (i.e. GOMOS), some have FOV information, or tangent heights. Please be consistent, if you feel the need to explain on detail for an instrument, then explain it for the other, regardless if it is given on the tables.

Response: The instrument descriptions have now been fully revised and homogenised.

P5 line 22: Please add emission after nadir. They are other type of measurements using the nadir view.

Response: Added.

Figure 3: What is wrong with SCIAMACHY and OMPS around 45W and 60S? Is this related with the South Atlantic Anomaly, if those please mention it here or in the SCIAMACHY / OMPS sections. This figure is also missing the sampling patterns of LIMS, MAESTRO and SAGEI. If MAESTRO sampling is missing because is the same as ACE-FTS, please make sure to specify that in the ACE-FTS subplot title or in the caption.

Response: OMPS and SCIAMACHY sampling patterns are indeed affected by the South Atlantic Anomaly as correctly noted by the reviewer. This has been noted in the caption. We added a sampling pattern for LIMS. ACE-FTS has the same sampling

pattern as ACE-MAESTRO, and SAGE I is very similar to SAGE II, HALOE and SAGE III/ISS, which is now also noted in the caption.

P6 line 17: delete space before 70S

Response: Deleted.

P7 line 2: "was about 3 to 5km" contradicts the statement made in the introduction "1 to 4 km"

Response: See answer above.

P7 line 15: if POAMII and POAMIII tracks are identical why show them on Figure 3. (Just curious what happen to POAM1)

Response: POAM III is now removed from the figure. Note, POAM 1 was a much earlier instrument of a fundamentally different design and was never operational.

Section 2.6. I thought there were different retrieval versions for this instrument. Which one are you using and why?

Response: This section refers to Odin OSIRIS, for which there is only one retrieval version for this instrument, version 5.10.

P8 line 14: This is the same principle of sun occultation which was not explained previously. Why explain it here? or better yet Why not explain the sun occultation and other methods the first time they are introduced...

Response: The information is deleted, we now refer to the SPARC report for the information on different measurement techniques since this seems to be too much of a textbook material.

P8 line18: altitude resolution of GOMOS is 0.5 to 1.7 versus 3 to 5 UARS MLS are you should you do need to use the averaging kernels. Could you prove that the comparisons do not improve when comparing so dissimilar resolutions?

Response: Thanks for the suggestion. We agree that validation exercises of individual instruments need to take vertical resolution into account, ideally by applying the averaging kernels. Our evaluation approach, however, includes 18 instruments with varying vertical resolutions and also varying spatial coverage. We have discussed the impacts of such differences in the report and in companion publications (Neu et al., Toohey et al.) for selected trace gases and instruments. It is beyond the scope of the current activity to apply sampling patterns and averaging kernels to all data products. In addition, we want to stay consistent with and comparable to earlier results based on older data versions (Hegglin et al., 2013; Tegtmeier et al., 2013; SPARC, 2017), and have therefore decided to keep the current approach (see text in Section 3.2.1).

P8 line 26: I thought it was 0.025 cm-1 (see for example https://amt.copernicus.org/articles/2/337/2009/amt-2-337-2009.pdf )

Response: The 0.025 is the number you get if you apply the textbook formula to the maximum optical path difference. But in real life you do not get this, because an instrument is never perfectly aligned, the beam is not infinitesimal narrow etc. Thus, industry responsible for building MIPAS guaranteed only 0.035. This is the number we work with. The text remained unchanged.

P9 line 3: Please justify why are you using this retrieval and not any of the many others. . .

Response: See answer to reviewer 1. Some text is added.

P9 line 23: “the latitude scanning (see figure 3) is the same each year”. That is not true, it changes slowly, the pattern may be the same but not the time of the measurements, see below:

Response: Text has been changed.

P9 line 25: Methodology discussions should be moved to section 3

Response: Information moved.

P10 line 5: Again, I am assuming that their sampling patterns are identical. In that case, Figure 3 should be labeled differently so that the reader know that their sampling pattern is the same..

Response: Done.

P10 line 8 Methodology discussions should be moved to section 3

Response: Information moved.

P10 line22: The instrument was not damaged. Please change to: Unfortunately, during launch, most of the aperture was obstructed by a plastic film used for insulation that became detached during the ascent to orbit. (or something similar) https://doi.org/10.1117/12.623574

Response: Changed. However, we use the Gille et al. 2008 reference, which discusses the issue in detail as well.

P10 line 24 Coverage is 65S – 82N https://doi.org/10.1029/2007JD008824

Response: Changed, thanks for spotting this mistake!

P11 line 11: Coverage is between 38S-65N in north looking days and 65S-38N on south looking days so, as shown in figure 3, I could be argued that the coverage is actually 65S-65N. What does nominal mean in this instance?

Response: Note, the South looking days only happened during three occasions. We now explain this in the text and also deleted the notation of nominal: Three times during the observation period, in late November, middle of February, and beginning of April, the ISS turned 180 deg along its yaw axis, so that the field of view deflection was pointing southward, resulting in inverse hemispheric observation ranges (65 degS-38 degN).

Section 2.17: I do not believe TES is used at all in the manuscript. Please delete this section or include examples using TES data.

Response: Section is now deleted.

P11 line 18: the apodized or unapodized detail was not mention for MIPAS.

Response: Information is now deleted.

P12 line10: The data preparation and handling of the dataset is a really important step of the whole endeavor. Please provide a brief description here, or in each instrument section. Also, the SPARC data initiative does not have information on the new datasets included in this manuscript.

Response: We agree with the reviewer that the gridded dataset construction is one of the most important steps of the Data Initiative work and in fact has caused a substantial workload for the instrument PIs. It is due to this reason that some other available instruments/retrievals were not included in the study. However, it is hard to summarise all the crucial details of all the different approaches taken. The SPARC Data Initiative dedicated a whole chapter on this. Thus, we decided to highlight better where to find this information in the report and add the missing information on OMPS and SCIAMACHY as follows to Section 3.1: Note, for the new SAGE III/ISS gridded datasets, the same approach was followed as for the other SAGE instruments. For OMPS, the data are handled in exactly the same way as those from SCIAMACHY with exception of the rejection of measurements within the South Atlantic Anomaly (SAA) region. While for SCIAMACHY, a fixed latitude-longitude range is used, the SAA flags from the level-1 product are used for OMPS.

P12 line19: The conversion from altitude to pressure levels as well as the conversion from number density to volume mixing ratio will introduce an uncertainty. According to 10.5194/amt-9-2497-2016 this bias could be up to 5% in the upper stratosphere (Figure 8/section 6 of that paper).

Response: We have added a note on this issue and refer to Hubert et al. (2016).

P12 line16: is not clear to me what is meant by hybrid log linear interpolation, do you

mean that you linearly interpolated on log(pre) instead of on pre. A brief explanation will suffice. The terminology on Table 3 is barely used through-out the rest of the text. I recommend changing the blue-red color bar of Figures 5, 6, 8 etc, to a blue-white-red color bar with the white covering from -2.5 to 2.5. That way the reader could easily identify where the excellent agreement is found.

Response: We have added the following explanation: To this end, profile data have been carefully screened before binning and a hybrid log-linear interpolation in the vertical has been performed (i.e., the VMR is interpolated linearly in log-pressure). However, we have kept our original color scale for the difference plots since we had used it in this way in our previous papers and the SPARC (2017) report. This allows for easier comparison between old and new data versions in the different comparisons. Also, even for small differences it can be interesting to know if they are positive or negative.

P12 line 24: The authors seem to use climatology file when they meant zonal monthly mean files. It is confusing.

Response: We changed the notation throughout the manuscript.

Figure 4: Again not climatology file, zonal monthly mean file, space missing between and-LST_MEAN. MIPAS was not available in 2018. Do you mean 2008?

Response: Thank you for spotting these errors, now corrected.

P12 line 29: Toohey et al 2013 is not the only study about sampling biases for limb instruments, there are more, for example: 10.5194/amt-7-1891-2014, 10.5194/acp-16-11521-2016, 10.5194/acp-18-4187-2018, 10.5194/amt-12-2129-2019

Response: Thank you for pointing out these studies. Now added.

P13 line 26: How do the authors define if measurements from a given instrument are deemed unrealistic. I think the MIM will be more robust if the authors defined quantitively which measurements to include or exclude.

Response: This is for measurements outside the ±3 sigma range. Information now added.

P14 line 17: Since the second summary is not going to show here, please delete this paragraph.

Response: Deleted.

P16 Line15: Why are the author evaluating different retrievals for OMPS? If they are going to do that, they should include all the retrievals from MIPAS and OSIRIS, etc.

Response: We include them here as there is not yet enough information from independent validation exercises available that we could use to exclude one or the other retrieval.

P16 line 23: Wang et al is published already. And it shows how to get rid of the altitude registration problem. Such correction should be implemented for SAGEIII/ISS. As well as for SAGEII since that product is also affected.

Response: We have updated the reference. We have now also updated the SAGE III/ISS and SAGE II data by correcting the altitude registration problem.

P17 line 27: I thought kernels were not considered at all through the whole study. Further I do not see what retrieving in log space has to do with the kernels. The averaging kernels are concentration dependent disregarding if they are retrieved in log space or not. Retrieving in log space only ensures that the retrieval will be positive.

Response: Yes, the reviewer is correct. None of the evaluations take the AKs into account. We have now reworded this passage: Note, it is expected that the application of averaging kernels would likely improve these comparisons (which should be tested in future work).

P19 line 1: instead of the current figure 5 why did the authors do not show the figure 5 and 6 equivalent for $CH_4$. It will presumably show the same results as Figure 7 but I

will be consistent with the previous to sections.

Response: We see the reviewer's point but prefer to provide a selection of the evaluations we show in the SPARC Data Initiative report (2017) as outlined in the introduction of the manuscript.

Figure 7 what is the shaded region is this the excellent agreement region or the very good please specify in the caption.

Response: We have now added an explanation in the caption.

Figure 8: the color bar units are missing. Is the level in the differences really 2 or 2.5.

Response: Thank you for catching this. It should indeed have been 2.5. Now amended.

P20 line11: Not "climatologies", zonal monthly mean from SMR, MIPAS, . . .

Response: Changed.

P20 line18: Again, be consistent in Figure 5 and 6 you only show the MIM, not the individual instruments zonal means. Now, looking at those, why are the authors including SMR into the MIM when there is a clear high bias around 20hPa in the tropics and a clear artifact around 60N.

Response: We understand the reviewer's wish to see consistency across the evaluations. To find this information for all (most) trace gas species, the reviewer is referred to the SPARC Data Initiative report (SPARC, 2017). Here, we would like to showcase a range of evaluations to highlight how one can get different pieces of information on the different datasets when doing so. As also argued for Figure 13, the value of having the zonal mean fields of each instrument included in this figure is to show obvious, non-physical features in some of the instruments (e.g. Aura-MLS, SMR), which are not obvious from just looking at the difference fields.

P21L16: Were some instruments not included for O3, H2O in figure 14 due to differences in time period, Why is this important for CO but not for H2O, O3, etc?

Response: The reason we left out SMR and MIPAS(1) in this figure is to remain comparable to the SPARC Data Initiative report. Now added as explanation.

P22 line 3: Why was ozone at higher altitudes not corrected using a chemical box model? There is a strong diurnal variation in the US-LM affecting those concentrations.

Response: Diurnal ozone variations are of ~10% below 1ĽhPa and grow with increasing altitude up to more than 100% for upper mesospheric levels [e.g., Wang et al., 1996; Schneider et al., 2005]. In addition, the impact of temperature uncertainties on the conversion from altitude to pressure during the climatology production may cause additional errors that are particularly pronounced in the LM. Therefore, we decided to not correct the mesospheric ozone climatologies, but instead only evaluated the climatologies up to 1 hPa. This information is now added to the oznoe section.

P22 line 25: Why does figure 11 does not include the MIM? Also, this figure should include panels showing the difference versus the MIM to be consistent. How are the authors evaluating the good agreement (P22 line27) or the agrees well (P22 line31) or the very good agreement without it.

Response: The current version of Figure 11 has 15 panels. For readability and clarity of the figure, we have decided to keep the figure in its current version and to not include another 15 panels. The figure focuses on the consistency of the comparisons across the different nitrogen species, which benefits from showing comparisons for all species in one row. This would not be possible any more, if the differences to the MIM would be included. However, we have added the line for the MIM to each of the panels to guide the reader's eye.

What does the "s" stands for in ACE-FTS s10am etc?

Response: See next answer.

Figure 12: What is the "s" or the "ss"

Response: "s" stands for "scaled to 10 pm" and "ss" for "sunset measurements". Now

added to the caption.

P24 line 1: sorry if this is obvious and I am just not understanding it. How come if you compute anomalies as monthly – MYM (multy year mean) might display a diurnal cycle while doing (monthly – MYM) / MYM *100. does not?

Response: In a first approximation, the difference due to the diurnal variation can be described by a multiplicative factor not changing with the time (as the measurement time is fixed), this factor cancels out when calculating the multiplicative anomalies.

P24 line 11: "possibly due to its higher vertical resolution" This statement could be proven by applying for example the MIPAS averaging kernels to HIRDLS and repeating the comparison.

Response: We agree with the reviewer that this suggestion should be tested and added in the following note '(which should be tested in future work)'

P24 line 17: Again, please be consistent, in previous sections there has always been mention of the previous evaluation.

Response: We added to the manuscript the following statement: In comparison to earlier evaluations (SPARC, 2017), the updated nitrogen data sets show a slightly improved agreement. In particular the scaled ACE-FTS data sets agree better with the other time series in terms of absolute bias and seasonal cycle.

P24 line32 SMR data is not used at all in Figure 13, i.e., the only comparison shown in the manuscript.

Response: We have clarified in the text why SMR is not shown here.

P25 line 1: please include the equivalent of Figure 5 and 6 for HO2. Showing the MIM and the difference versus the MIM for SMILES, MLS and SMR before discussing the November 2009 and February comparison.

Response: See next comment.

Figure 13: are you comparing the MIM for 2 instruments. If you are, then just shown the MIM and then the differences. Were the SMILES selected according to their local time or all local times were included in this comparison?

Response: We have chosen this format for this figure since the SMILES-MIM/MIM differences look exactly opposite of that from Aura-MLS, and there is more information in looking at the Aura-MLS and SMILES zonal mean fields instead. We have added a note in the caption to highlight this. We have selected all available SMILES daytime data (LT) as now stated in the caption.

P25 Summary evaluations. Is the reader supposed to be using the definitions on Table 3 with Figure 14, 15 and 16? If it is, the color bar should reflect the values determined in such table 3.

Response: Based on the color scheme the 2.5%, 5%, 10% and 20% values can be clearly distinguished. We prefer to keep the current color scheme in order to stay consistent with the evaluations of earlier data versions published in the SPARC report. We added the following statement to the manuscript to provide the reader though with some more guidance: Note, we adopt the same vocabulary (Table 3) for the summary comparisons (based on relative standard deviations) as used earlier for instrument specific evaluations (based on relative differences).

P26 line1: Why is there no HO2 1-sigma multi-instrument summary as for the other molecules? There are other molecules with only 2 or 3 instruments as for SF6, HF, NO, HCl.

Response: We do not provide an assessment of the uncertainty in the mean field of HO2 because of the very limited time coverage of SMILES data which does not provide a meaningful annual mean distribution.

Why is there no BrO 1-sigma summary? Response: We do not provide an assessment of the uncertainty in the mean field of data products if they are not available at the

same local solar time. This is the case for BrO, and also N2O5, ClONO2, and HNO4. Figure 15 is not consistent with Figure 14, I understand that the authors are redoing the Figure from the SPARC DI but for this paper it will be much better is they kept a simple layout as in Figure 14. That is, remove the boxes and the chemistry explanation, etc.

Response: Figure is now simplified and necessary information added to the caption.

P26 line 11 The acronyms (CSA, ESA, JAXA, etc) are not are not defined.

Response: We delete the different agencies' names here and write instead "from various space agencies".

P26 line 16 These manuscript only updated the trace evaluation, as written it implies that it also evaluated the aerosols observations.

Response: We now clarify the situation by adding the following text: Note, aerosol evaluations and monthly zonal mean timeseries data will be presented in a follow-on study).

P26 line 26: The doi zenovo link those not exist. As in, google cannot find it (neither the zenovo webpage). You have to go into the webpage and search for Hegglin to get the dataset. Please clarify

Response: The reviewer seemed to have misspelled the word zenodo. Our google search did provide a result that links directly to the zenodo archive.

P27 line 5: Why generally? It always produces larger sample sizes.

Response: We now deleted the word 'generally'.

P27 line 9: Toohey et al 2013 only shows the sampling biases for Ozone and water vapor. As mention here it seems that it investigated the sampling biases for all molecules included in the SPARC data initiative.

Response: This has been clarified in the text.

Table 1- References column: Why are some years in brackets and some don't? Cloud top is not an appropriate vertical range, please provide the lowest possible measured altitude in clear sky and then either on the text or as a tablenote specify that in the presence of clouds is from the cloud top. Why do HALOE, UARS and SMR have pressure ranges as well as altitude ranges. Please be consistent.

Response: Thank you. Amended. For the pressure ranges. Different instruments use different native vertical grids. We had added (although inconsistently) the information in km where this native grid was pressure. We now do this consistently across the different instruments.

Table 2: Why does MIPAS have two vertical resolutions? Due to the change in spectral resolution? Please clarify.

Response: We have clarified this issue in the table caption: Note, the low (high) vertical resolution along with the high (low) data density entries in MIPAS refer to the two different measurement modes MIPAS had been measuring in before and after 2004, referred to as MIPAS(1) and MIPAS(2) in later tables, respectively.

What happened to the ascending LT of meas for SCIAMACHY?

Response: The ascending part of the SCIAMACHY orbit lies mostly in the darkness. Generally, there are no measurements available in this part of the orbit. In the middle of the summer, however, there are some measurements at high latitudes during the ascending part of the orbit. These measurements are not included in the gridded data set to avoid averaging of observations performed at very different local times. We now clarify this in the caption: Also note, the ascending part of the SCIAMACHY orbit lies mostly in darkness, resulting in only few measurements which are not included in the SPARC Data Initiative gridded datasets (thus the ascending measurement LT is not listed).

How come TES has a LT of meas but not MLS and HIRDLS.

Response: We now have removed information on TES. The LT of MLS and HIRDLS are changing with latitude.

For SAGEIII/ISS - Wang et al has already been published.

Response: Thank you, reference is now updated.

Table A1 I think this should be part of the main text. Also, I think the variable type is wrong: $N_2O$, N2O_NR and N2O_STD are 3d, [lat, pre, months] AVE-dom, ave-lat, lst's are 2D. [lat, months] But all are described as GEO2D.

Response: The reviewer is correct, amended.

Further, looking at some files, for example the OMPS-SASK or OSIRIS have other variables:

OZONE_CONCENTRATION_STANDARD_ERROR, OZONE_VMR_STANDARD_ERROR OZONE_COUNT_IN_BIN

Response: Yes, we are aware on some files showing deviations from the specified formatting. We had acknowledged this in the table caption.

Table A2. Why do some molecules used different versions? For example, MIPAS (1) uses v21 v20

Response: This is because for each gas a dedicated retrieval setup was developed. For gases with the lowest data versions the original retrieval setup was found good enough, while for others, data were reprocessed with a refined retrieval setup.

Table A3. Why is v4.0 in SAGEIII/M3M in brackets?

Response: A mistake. Now amended.

Table A4. Occultation needs to be bold in the Stellar or lunar occultation section.

Response: Thank you, amended.

Table A5. LIMS reference, is there no published reference. This dataset is from 1979? Perhaps this was meant to be Remsberg 2009?

Response: These references now have been updated.

MIPAS additional comments: Meas mode switched in 2004 from high spectral to low spectral resolution. Not from high to high

Response: Note, it is correct as written since it refers to a tradeoff between spatial and spectral resolution. It says from high spectral to high vertical resolution.

SAGEIII/ISS reference: Wang et al has been published.

Response: This has been updated.

HALOE: the vertical range states up to 80km, please provide the minimum altitude

Response: Minimum provided.

Table A6. MIPAS: Meas mode switched in 2004 from high spectral to low spectral resolution.

Response: Changed. See comments above/below.

SAGEIII/ISS cite 10.1029/2020JD033803.

Response: Reference added.

Is there an upper vertical range limit for UARS MLS ?

Response: Yes, we have changed this to “50-80 km”.

Table A7 The additional comment for MIPAS in the previous tables is that the Meas mode switched in 2004 from high to low . . ., in here it says Change in spectral resolution in 2005. Please be consistent. And also the years do not change.

Response: Now made consistent across all tables.

Table A8 should include https://doi.org/10.1029/2007JD008723 for MLS.

Response: Reference now added.

Table A9: In the previous tables the authors have given the overall period for MIPAS. I Actually prefer the format in A9. But be consistent.

Response: Changed according to the reviewer's suggestion.

Table A10. No references for SAGEIII and SCIAMACHY, at least provide the one describing the instrument.

Response: References now added.

The day/night in MIPAS imply that you are taking the day night difference? If it those, this detail should be in the additional comments.

Response: As discussed in section 4.5, "day/night" refers to the useful height range specified in the line above (12-50/70 km). We now write 12-50 (12-70) km for day (night) meas. which hopefully clarifies the issue.

Table A11: I do not understand the additional comment for UARS MLS, please clarify

Response: We now write "Data with significant (1-3 ppbv) low bias at pressures <15 hPa and high bias below the VMR peak" and hope this clarifies the entry.

Please use the vertical range MIPAS layout for all cloud top's in these tables.

Response: Format made consistent across the tables.

Table A12: Why is there a (UTLS) in the AURA MLS row. The resolution according to the v3 quality document varies from 4 at 10hPa to 10 at 0.046 hPa. Please clarify.

Response: We have now deleted the 'UTLS', thank you for spotting this error.

---

## Author Comment (AC2) · 15 Mar 2021

This manuscript describes the results of the SPARC initiative to identify and compare nearly all existing remote sensing data of the atmosphere from limb sounders, along with updates since the initial report. This was a long and careful effort, and the results are extremely impressive. Anyone using limb sounding measurements in their work (or interested in the chemical composition of the stratosphere and mesosphere) will
benefit from this project - to identify data availability, compare different instrumental data sets, or just to understand composition and chemistry in the region from about 200 hPa to 100 km altitude.

The authors used a "top-down" approach, in which all measurements are averaged into altitude-latitude bins and compared. For a project of this scope and completeness, this is probably the only feasible way to accomplish the goals and show the results in a finite amount of space. It is also a nice complement to more traditional methods of evaluating remote sensing data by comparing coincident profiles or measurements. Only some of the highlights are included here, but all data are available on a separate website. For many readers, the multi-instrument mean (MIM) will be most useful, but there are also carefully analyzed data on differences between each data set and the MIM, as well as details about sampling, and a brief description of the general level of understanding about each constituent. With a few minor revisions, this paper will be an excellent contribution to the literature.

We thank the reviewer for her/his positive assessment of the work we conducted within the SPARC Data Initiative and her/his valuable comments on our manuscript. Please find below our answers in blue.

P.17, SAGE II improvements - I am always cautious when data sets are reanalyzed to show better agreement. It is natural for the reanalysis to more closely approach the consensus value, but that is not necessarily the "true" value. The explanation is quite reasonable though; no changes needed.

We fully agree with the reviewer's cautioning remark! To express this more neutrally, we now say 'shows large changes when compared to' instead of 'has much improved over'. We still speak of an overall improvement between data versions at the end of the paragraph.

Section 4.5 and following - Peroxyacetyl nitrate (PAN) is not mentioned in the paper, but it does contribute to NOy at the lower end of the altitude range shown (Figure 15).

From in situ data from the NASA ATom mission, the fraction of PAN to NOy can be 1/3 to 1/2 in the tropics near 200 mbar. In the stratosphere (higher latitudes at 200 mbar), PAN is usually 10midlatitudes, probably more likely in the summer when the tropopause is higher. This will likely only affect the very bottom of the NOy plots, but it should probably be at least mentioned briefly in the text.

We have added the following statements in the nitrogen family section:

"While HNO3 and NOx constitute 80-100% of all possible species of NOy in the LS, PAN can constitute as much as 20-50% in the tropical UT and extratropical UTLS (i.e., altitudes below 200 hPa) (Kendo et al., 1997; Fadnavis et al., 2014)."

"It should also be noted that although available from both MIPAS and ACE-FTS, none of the NOy climatologies presented here includes PAN, which can be a significant contribution to NOy at the lower end of the altitude range shown as mentioned above."

P.27, l.7 Besides the non-uniformity of sampling, another factor can be the long-term trends in various gases. For example, CFC11 and 12 reached a peak and are now decreasing. But for a gas like water, the long-term trend is not so obvious (and quite important for climate forcing). The "true" (or measured) MIM could be changing over time. This is beyond the scope of this paper, but could be (briefly) mentioned as a possibility in the section on H2O. My understanding of the MIM here is that it is the average over the duration of the measurements, and does not change with time. If that is not correct, then I missed it in the text.

Thank you for this comment. A sentence on trends being a potential issue has been added in the new Section 3.2.1, which discusses the methodology. We also clarify that the MIM is the average over all the measurements and all the years/months that are presented in a specific evaluation (and not over the full timeseries available) by adding the following statement: *The MIM is thereby calculated over the same years (or months) and instruments as presented in a given evaluation (and not over the full timeseries available).* The MIM is thus changing with each evaluation done for different

time periods, and depending on the different sets (instruments) considered.

Some further, more specific comments:

P.1, l.8 It is difficult to separate "long-lived trace gases" and "transport tracers". There is considerable overlap. Just an observation; no changes needed in the text.

We agree that our categorisation didn't make much sense and changed them to: . . . the stratospheric trace gases of primary interest, O3 and H2O, major long-lived trace gases (SF6, N2O, HF, CCl3F, CCl2F2, and NOy), trace gases with intermediate lifetimes HCl, CH4, CO, HNO3), and shorter-lived trace gases. . .

P.2, l.2 "nitrogens"? Maybe "nitrogen-containing species"

Thank you. Changed according to the reviewer's suggestion.

l.7 add comma after "climatologies" (if "which" refers back to "approach").

Corrected.

l.12 "intended summary"

Corrected to "as an intended summary. . .".

P.3, l. 25, I was initially confused why TES did not appear here, but was on the list on P.2, l.6. It is explained on P.5, and is fine; no changes needed.

We now have removed most (except initial) TES-references and explain that we will not treat it in this document.

l.27 At some point NOx and NOy should probably be defined, although almost everyone knows what they are. I was curious how NOy would be handled, since it's hard to measure all components of NOy with limb sounding. Defining them later in the paper is OK; see comment above about components of NOy in the troposphere.

We have added a more succinct definition of NOy and the following statement to the first paragraph of Section 4.5, which in concert with addition above hopefully answers

the request of the reviewer:

While HNO3 and NOx constitute 80-100% of all possible species of NOy in the LS above 200 hPa, PAN can constitute as much as 20-50% in the tropical UT and (extratropical) UTLS (Kendo et al., 1997; Fadnavis et al., 2014).

l.33, "data are"

Corrected.

P.4, l.29 "its" instead of "the above" (since I don't see any disadvantages listed above)

Corrected.

P.5, l.4, "JGR - Atmospheres"

Corrected.

P.6, l.5 vs. l.17, dates include S3/ISS or not?

Dates corrected.

P.8, l.12, "98.55 S"? Maybe drop the "S"?

Corrected.

l.22, "on board Envisat"

Corrected.

P.10, l.29 perhaps something like "demonstration of ultrasensitive sub-mm limb emission observations..." And I don't think 4 K should be hyphenated.

Suggestion adopted.

P.13, l.8, not sure why "roughly-uniform" is hyphenated. Or why "generally" and "roughly" are both used together. Perhaps "is roughly uniform with respect to longitude".

Suggestion adopted.

l.17 "can be found"

Corrected.

P.14,l.17 Where can this second summary be found? In SPARC 2017?

This paragraph has been removed to answer a similar comment from another reviewer.

P.15,l.25 "spectroscopic"

Corrected.

l.30, I don't think that MS and US have been defined.

Now corrected, but see also Table 4.

P.16,l.28, What is the "LM"? OK, it's all in Table 4 (I missed the reference to it on P.12,

We now define the abbreviations in addition at their first occurrence (except in the abstract).

l.7). But would it be worthwhile to include the stratopause in a figure somewhere too (like the top panel of Figure 4, where it won't get in the way too much)? Maybe not. It was not obvious to me whether the boundary between the US and LM is simply taken as an altitude or pressure level, or whether it has latitudinal structure (or varies with month). Not an important point, and I may have missed a description of this - fine as long as it is clear to an interested reader. (If needed, you could probably put any explanations in the Table 4 caption.)

The following sentence has been added in Table 4 caption: "The transition between the stratosphere and the mesosphere (the stratopause) is here defined uniformly across all latitudes as the 1 hPa pressure level."

l.33, "carbon dioxide and other anthropogenically emitted greenhouse gases" or something like that. And maybe combine the next two sentences as "H2O is also a key

constituent in atmospheric chemistry as a source gas of the hydroxyl radical..."

This sentence now reads: "...a positive feedback to climate change driven by anthropogenic emissions of carbon dioxide and other greenhouse gases." We have also adopted the reviewer's second suggestion.

P.18, l.15 Does USLM mean "upper stratosphere and lower mesosphere"?

Yes, the explanation of such composites is described in Table 8

P.19, l.27 Need to edit this sentence - something like "a somewhat patchier difference field, however, it provides supporting evidence..." or "a somewhat patchier difference field, however, which provides supporting evidence..." or "a somewhat patchier difference field, however, providing supporting evidence..." I'm not sure I follow the logic in this sentence.

We have rewritten this indeed rather complicated sentence to: "MIPAS(2), despite exhibiting a somewhat patchier difference field, provides supporting evidence for a high bias in MIPAS(1) at this pressure level." and hope this clarifies the content.

l.33 I assume that "overlap year" describes when all three instruments were reporting data. Also, "confirms the results described here"

We have rewritten this sentence to: "It is important to note that CH4 showed only small trends in the troposphere over the time period 1998-2008, thus a trend in this trace gas is not expected to contribute significantly to the inter-instrument differences. An evaluation limited to the year 2005 (during which all instruments were reporting data) mostly confirms the results described here (not shown)."

P.20, l.1 I would not hyphenate "mean state".

Corrected.

P.21, l.16 "for which they provided data."

Corrected.

l.18 OK, here are the definitions of NOy and NOx (fine with me). In the midtroposphere, peroxyacetyl nitrate (PAN) is one of the major components of NOy. This is mostly below the altitudes shown here, but see comment above.

See answer above, we have now added a more detailed definition and highlighted that the instruments don't measure all of the species contributing to NOy.

l.25 "mechanisms"

Corrected.

l.29 "conversion" instead of "exchange"?

Corrected.

l.32 "is taken into account"

Corrected.

P.22, l.25, "latitude bands"?

Corrected.

P.23, l.8, "consistent with" rather than "confirming"?

Corrected.

P.58 I did not notice the reference to table 3 while reading the text. That is a fine way to make all the descriptions quantitative.

Yes, chapter 3 includes a reference to table 3.

Table A2, perhaps "added recently" or at least "newly added"

Corrected, thank you.

Figure 8 - missing units on the color bars. I am guessing they should be ppm and

percent, but the lower one could be ppb. Also, some of the subscript "4" look a little too small and too close to the "CH" (like for HALOE).

Corrected, thanks.

Figure 10 - x-axis labels are a little confusing. Is there room to write "Jan 2006" etc.? If not, then having the year under the month should work. Similar comment for Figure 12.

Changed to year numbers only.

Figure A1 - About the units on the color bars: In the middle figures, do the grid boxes represent the number of samples per year, and in the lower figures, the boxes represent the number of samples per month in each latitude bin for each month? I suppose so, but maybe that can be included in the figure (like in the color bar) or in the caption.

Included according to the reviewer's suggestion.

―――――――――――――――

---

## Author Comment (AC3) · 15 Mar 2021

This paper is an update to an ambitious effort to assess currently available satellite limb measurements of stratospheric trace gases. The authors are to be applauded for their comprehensive assessment of a number of different data sets produced by different institutions and spanning multiple decades. It is encouraging to see that for several of the species, the use of updated retrievals results in better agreement than in the earlier version of the SPARC DI data set. This data set will no doubt be useful to

the observational and modeling communities for studies of stratospheric composition. I have only a few minor comments and recommendations before it is accepted to ESSD.

We thank the reviewer, Sean Davis, for his assessment of our manuscript and his valuable comments. Please find below our answers in blue.

Page 3, lines 15-20 – This data also contributed to several of the S-RIP chapters/papers, and I think that is worth mentioning here somewhere.

We now have included a reference to S-RIP and cited the relevant publication (Davis et al. ACP 2017; Fujiwara et al., ACP 2017).

Page 9, Lines 1-3 – As I understand it there are multiple MIPAS retrievals from different groups. Could the authors please provide some justification for why they choose the IMK retrieval, and/or provide any information and references concerning known differences between the retrievals?

Yes, the reviewer is correct, there are several MIPAS retrievals available, however, they were not contributed to the SPARC Data Initiative. We added the following sentence to highlight this and to provide a reference instead:

*"Several other MIPAS retrieval products are available (see Lossow et al., 2019), however, were not contributed to the SPARC Data Initiative in the required climatological format. Note, the IMK-processor also provides more species than these other processors."*

Page 5, line 22 – the reference to appendix table A4 seems quite out of order. Additionally, I don't understand the distinction between the figures and tables in the "appendix" versus the main text. Content-wise, it seems like the material in the appendix belongs in the paper itself and is not really an appendix.

Thank you for catching that the tables were not numbered in chronological order! We have now corrected the problem and moved all appendix tables and figures into the main manuscript.

Page 10, section 2.14 – It would be helpful if the authors mentioned the end date and reason for the end of HIRDLS data.

We added the following information: *"HIRDLS stopped acquiring data on 17 March 2008 due to a chopper failure."*

Page 11, section 2.16 – It looks like the authors are using two different versions of OMPS (based on table A5). Which is the primary one they are considering? Reference to/discussion of the version they are using here would be helpful. Also, I believe there is yet another OMPS-LP retrieval that is not included here (Kramarova et al., 2014). As with the MIPAS discussion it would be helpful to have some insight into the choices the authors have made and justifications for excluding certain products, and what the known major differences are between the retrievals.

Kramarova, N. A., Nash, E. R., Newman, P. A., Bhartia, P. K., McPeters, R. D., Rault, D. F., Seftor, C. J., Xu, P. Q. and Labow, G. J.: Measuring the Antarctic ozone hole with the new Ozone Mapping and Profiler Suite (OMPS), Atmospheric Chemistry and Physics, 14(5), 2353–2361, doi:10.5194/acp-14-2353-2014, 2014.

Indeed, we use OMPS data based on two different retrieval algorithms. We added the following information: *"It should be noted that the OMPS-LP ozone datasets used in the SPARC Data Initiative are based on two different retrieval algorithms, IUP-OMPS (Arosio et al., 2018) and USask-OMPS (Zawada et al., 2018). The main difference between these two products is that the USask is retrieved using a 2D tomographic algorithm and the IUP uses a standard 1D algorithm. Furthermore, the spectral information and associated tangent height ranges are used differently. NASA also produces a stratospheric ozone product from OMPS-LP (Rault and Loughman, 2013) which is not included in the SPARC Data Initiative."*

Rault, D. F., and R. P. Loughman (2013), The OMPS Limb Profiler Environmental Data Record Algorithm Theoretical Basis Document and expected performance, IEEE Trans. Geosci. Rem. Sens., 51, 2505-2527.

Page 11, lines 26-28 – I think the term "climatology" is a confusing term to use to describe this data set. As the authors acknowledge here, a climatology typically refers to some long term mean state. But in this paper, "climatology" is being used to describe a time series. The authors also use the term "climatology" (e.g., "climatological approach") as a stand in for "gridded data set" when contrasting their approach to profile-to-profile coincident comparisons (e.g., sentence starting line 28). I also find this terminology confusing. The data set the authors have produced is a gridded time series data set, and I think it is more accurate to describe it as such.

We agree that the term "climatology" for the monthly zonal mean timeseries can be confusing. We now change this notation throughout the manuscript to "gridded datasets", "mean fields" or "timeseries of monthly zonal mean fields". However, we kept the "climatological validation approach" terminology in order to highlight its difference to the coincident validation approach, since this evaluation approach is based on comparing multi-annual means of the zonal monthly mean fields.

Page 11, starting line 28 – It seems as though one of the main advantages of the approach used here (comparing gridded data sets) is that all data from each sensor are used in the comparison, as opposed to profile-profile comparisons where some profiles simply don't meet the chosen coincidence criteria. I believe this is the reduction in random error the authors are referring to here. However, this benefit must be weighed against the sampling bias (e.g., as addressed in Toohey et al 2013) that is introduced when one grids data. It's not totally obvious how these two factors compete, and some acknowledgement of this balance would be appreciated.

Some more discussion of the influence of the sampling bias has been added in the new Section 2.3.1.

Page 12, line 16 – What do the authors mean by hybrid log-linear here? Do you mean interpolating the log VMR linearly in altitude, or interpolating the VMR linearly in log pressure? I'm guessing the latter, but please clarify.

"Hybrid log-linear" refers to interpolating VMR linearly in log pressure, as correctly guessed by the reviewer. We now added this explanation for clarification.

Page 12 lines 16-20 – It appears as though the authors are using the most convenient method for converting to VMR on a pressure grid for each individual data set. I don't mean to belittle this approach because it would be a rather Herculean task to use a common data source for all the different instruments. And even then some of the retrievals may use p/T in their retrieval "upstream" of what is available to the public. Nevertheless, I think it is important to recognize that this grid conversion using different ancillary data as a possible source of uncertainty. I am not aware of any work that has attempted to quantify this source of uncertainty, but any additional discussion or references related to this issue would be very helpful.

This is a valid comment and we have added the following text to the manuscript:

*It should be noted, that using different ancillary data for the grid- and unit-conversions will introduce an additional source of uncertainty, which has not been quantified here. Any known problems in the ancillary temperature/pressure data that were used to convert measured species from their native to VMR/pressure grids have been fixed by an updated retrieval algorithm or minimized with empirical corrections. For example, problems in the older SAGE II (v6.2) temperature/pressure auxiliary files, mainly in the tropics above 2 hPa, were empirically corrected (Froidevaux et al., 2015) before being incorporated in the original SPARC Data Initiative (see SPARC, 2017). The anomalous temperature problem in SAGE II (v6.2) has been fixed in the latest V7 retrieval, which is used in the updated SPARC Data Initiative dataset and this manuscript. Both SAGE III/ISS (v5.1) and SAGE II (v7) data were also updated to remove/minimize the effects of altitude registration errors in the auxiliary temperature profiles (Wang et al., 2020).*

Page 14, paragraph line 17 – 22 – This paragraph doesn't make any sense and should probably be removed. It is addressing some evaluation that is not shown in the paper, and doesn't really even explain what the result is from this evaluation.

We now have removed this paragraph and rewritten the previous paragraph accordingly.

Page 15, line 25 – spectroscopical -> spectroscopic

Corrected, thank you.

Page 15, line 30 – considerable -> considerably

Corrected.

Page 16, line 22 – The Wang et al paper is now published

Reference is now updated.

Page 16, lines 22-24 – The altitude registration problem is easily corrected, as outlined in the appendix of Wang et al 2020. The authors should implement this correction.

We have now updated both SAGE III/ISS (V5.1) and SAGE II (v7.0) data versions to address the altitude registration problem in the auxiliary temperature/pressure data.

Page 17, line 31 – "also slightly" -> "also has slightly"

Corrected.

Page 21, line 25 – "mechanism" -> "mechanisms"

Corrected.

Page 22, line 5 – I think you mean "time" here instead of "date"

Deleted.

Page 24, line 19 – this paragraph ends abruptly. Can you say something about how this compares to SPARC 2017, as is done for the other species?

We added the following text:*'In comparison to earlier evaluations (SPARC, 2017), the updated nitrogen data sets show a slightly improved agreement. In particular the*

*scaled ACE-FTS data sets agree better with the other time series in terms of absolute bias and seasonal cycle.'*

Page 23, line 23-25 – This is a run on sentence.

We changed this sentence as follows:

*In general, we expect increasing NOy values during the dynamically quiescent spring and summer, and this is observed by ACE-FTS and MIPAS. In the NH, the NOy maximum is observed in boreal autumn by all three instruments. In the SH spring, Odin shows a secondary maximum that is less pronounced than in the NH, but this provides for a better agreement with the other two datasets. For ACE-FTS, the too low NOx values in the SH and NH boreal winter cancel out with the too high HNO3 values, resulting in an overall good NOy agreement with MIPAS.*

Table A5 – as previously mentioned, Wang et al. paper has been published now.

Corrected.

Table A6 – should cite Davis et al. for the SAGE III/ISS water vapor

Davis, S. M., Damadeo, R., Flittner, D., Rosenlof, K. H., Park, M., Randel, W. J., et al. (2020). Validation of SAGE III/ISS solar water vapor data with correlative satellite and balloon-borne measurements. Journal of Geophysical Research: Atmospheres, 125, e2020JD033803. https://doi.org/10.1029/2020JD033803

Thank you, added.

Data versions questions:

In general, it is preferable to use the newest data set from each satellite. There is a new Aura MLS version 5 data set, which I assume will become the widely adopted version of the data to use. Could this be included in the data set? Similarly, there is a new ACE-FTS version (4.1) that is the recommended version. Also, which version of MAESTRO data is being used here? It says "31" in the table, which I assume refers to

v3.1. But there are several sub-versions of 3.1 (eg, 3.11, 3.12, : : :). The latest version is 3.13 – is that what is being used?

We agree with the reviewer that it would be preferable to use the latest data versions. However, we rely on the expert advice of the instrument PIs and use the data versions that they are most comfortable sharing. This is why we use version 3.6 for ACE-FTS and version 31 for ACE-MAESTRO (note this is the official versioning number for $H_2O$, which is different from ozone). These data versions have generally undergone considerable validation efforts. Note that Aura-MLS version 5, while available to the public, is just finishing its reprocessing and has not yet been re-evaluated by the MLS team, or rushed to processing into the format of the SPARC Data Initiative. At some point, the data versions have to be "frozen", and a considerable amount of work is needed to redo all the comparisons shown here; moreover, most comparisons will not be affected significantly.